# DHG-Bench: A Comprehensive Benchmark for Deep Hypergraph Learning

**Fan Li[1], Xiaoyang Wang[1]\*, Wenjie Zhang[1], Ying Zhang[2], Xuemin Lin[3],**
[1]The University of New South Wales, [2]Laboratory for Statistical Monitoring and
Intelligent Governance of Common Prosperity, Zhejiang Gongshang University,
[3]Shanghai Jiao Tong University
{fan.li8,xiaoyang.wang1,wenjie.zhang}@unsw.edu.au
ying.zhang@zjgsu.edu.cn, xuemin.lin@sjtu.edu.cn

## Abstract

Deep graph models have achieved great success in network representation learning. However, their focus on pairwise relationships restricts their ability to learn pervasive higher-order interactions in real-world systems, which can be naturally modeled as hypergraphs. To tackle this issue, Hypergraph Neural Networks (HNNs) have garnered substantial attention in recent years. Despite the proposal of numerous HNNs, the absence of consistent experimental protocols and multi-dimensional empirical analysis impedes deeper understanding and further development of HNN research. While several toolkits for deep hypergraph learning (DHGL) have been introduced to facilitate algorithm evaluation, they provide only limited quantitative evaluation results and insufficient coverage of advanced algorithms, datasets, and benchmark tasks. To fill the gap, we introduce DHG-Bench, the first comprehensive benchmark for HNNs. Specifically, DHG-Bench systematically investigates the characteristics of HNNs in terms of four dimensions: effectiveness, efficiency, robustness, and fairness. We comprehensively evaluate 17 state-of-the-art HNN algorithms on 22 diverse datasets spanning node-, edge-, and graph-level tasks, under unified experimental settings. Extensive experiments reveal both the strengths and limitations of existing algorithms, offering valuable insights and directions for future research. Furthermore, to facilitate reproducible research, we have developed an easy-to-use library for training and evaluating different HNN methods. The DHG-Bench library is available at: https://github.com/Coco-Hut/DHG-Bench.

## 1 Introduction

Graph-structured data has become a ubiquitous tool for modeling the complex relational dependencies among entities in various domains, such as social analysis (Tan et al., 2025a), e-commerce (Liu et al., 2021), and finance (Li et al., 2024b). Graph Neural Networks (GNNs) have emerged as the dominant approach for learning on such data, owing to their exceptional ability to leverage both the graph topology and node attributes. However, many real-world systems involve multi-way or group-wise interactions beyond the pairwise connections of graphs. For instance, multiple authors co-write a paper in co-authorship networks (Yang et al., 2022), and groups of proteins interact collectively in biological systems (Kim et al., 2024b). These higher-order interactions can be naturally modeled by hypergraphs, where each hyperedge connects an arbitrary number of nodes. As hypergraphs become increasingly prevalent, there is a growing demand for predictive tasks on them, such as estimating node properties or identifying missing hyperedges (Kim et al., 2024b). However, directly applying GNNs to such tasks inevitably collapses higher-order interactions into pairwise relations, resulting in significant information loss and thus sub-optimal performance (Chien et al., 2022).

To mitigate the aforementioned problem, Hypergraph Neural Networks (HNNs) (Yadati et al., 2019; Chien et al., 2022; Wang et al., 2023b; Tang et al., 2025) have become the prevailing paradigm for deep hypergraph learning (DHGL), attracting considerable research interest in recent years. These

---

\*Corresponding author.

methods employ neural architectures to transform nodes, hyperedges, and their associated features into vector representations (i.e., embeddings) that effectively preserve higher-order semantics. HNNs have demonstrated state-of-the-art performance across diverse industrial and scientific applications, including product recommendation (Khan et al., 2025), 3D object detection (Fixelle, 2025), and disease diagnosis (Han et al., 2025).

Despite the emerging studies of HNN algorithms, the comprehensive benchmark for evaluating these methods remains absent, bringing out the following problems: *(i)* Existing works utilize different datasets, compared baselines, and experimental setups (e.g., data splitting strategies and parameter settings), which makes it challenging to achieve a fair comparison. *(ii)* Existing works primarily focus on the effectiveness evaluation of HNN algorithms, while lacking empirical understanding of their efficiency and trustworthiness (e.g., robustness and fairness), both of which are essential for real-world deployment. This prevents practitioners from understanding the advantages and limitations of HNN algorithms from multiple perspectives and makes it difficult to select appropriate methods for different application scenarios. Hence, there is an urgent necessity within the community to develop a standardized and comprehensive benchmark for HNNs.

In recent years, several open-sourced toolkits, including HyFER (Hwang et al., 2021), DHG (Gao et al., 2022), and TopoX (Hajij et al., 2024), have been proposed to facilitate benchmarkable deep hypergraph learning. However, these works provide only limited or even no quantitative performance comparisons, which thus compromises their practical value for practitioners. Furthermore, they fail to incorporate many state-of-the-art HNN algorithms and provide insufficient coverage of benchmark datasets and evaluation tasks. Specifically, HyFER supports only the implementation of three HNN models, while the other two libraries include only HNNs proposed before 2023. Moreover, none of these toolkits integrate heterophilic hypergraph datasets, which represent a particularly challenging setting (Li et al., 2025c), nor do they support graph-level tasks (e.g., hypergraph classification). These limitations significantly restrict the reproducibility and comprehensive evaluation of advanced HNNs.

To bridge the gap, we propose DHG-Bench, which serves as the first open-sourced and comprehensive benchmark for HNNs. Our benchmark encompasses 17 representative HNN methods and 22 diverse hypergraph datasets covering node-level, edge-level, and graph-level tasks. We employ standardized computational operators and APIs, along with consistent data splitting and processing strategies, to ensure fair comparison. Beyond effectiveness, our benchmark supports multi-faceted analysis, allowing researchers to investigate the efficiency, robustness, and fairness of current HNN algorithms. Through extensive experiments, we derive the following key insights: *(i)* Existing HNNs exhibit substantial performance variability across datasets and tasks, reflecting their limited generalization ability. *(ii)* Most HNN methods struggle to strike a satisfactory balance between predictive performance and computational efficiency. *(iii)* The performance of HNN algorithms is affected by different types of data perturbations, with feature-level and supervision-level perturbations causing particularly adverse impacts. *(iv)* HNN algorithms tend to result in more severe fairness issues than deep models without higher-order message passing, such as MLPs. Our main contributions are as follows.

- **Comprehensive Benchmark.** DHG-Bench enables a fair and unified comparison among 17 state-of-the-art HNN methods by standardizing the experimental settings across 22 widely used hypergraph datasets of diverse characteristics. To the best of our knowledge, this is the first comprehensive benchmark for deep hypergraph learning.

- **Multi-dimensional Evaluation and Analysis.** We conduct a systematic analysis of existing HNN methods from various dimensions, encompassing effectiveness, efficiency, robustness, and fairness. Extensive experiments uncover the potential strengths and limitations of existing HNN algorithms, offering valuable insights to inform and inspire future research in this field.

- **Open-sourced Benchmark Library.** We release DHG-Bench, an easy-to-use open-sourced benchmark library to support future HNN research. With our toolkit, users can evaluate their algorithms or datasets with less effort.

## 2 PRELIMINARY

Let $\mathcal{G}(\mathcal{V}, \mathcal{E}, \mathbf{X})$ represent a hypergraph with vertex set $\mathcal{V} = \{v_i\}_{i=1}^{|\mathcal{V}|}$ and hyperedge set $\mathcal{E} = \{e_j\}_{j=1}^{|\mathcal{E}|}$. $\mathbf{X} \in \mathbb{R}^{|\mathcal{V}| \times F}$ is the node feature matrix with $F$-dimension. In this benchmark, we focus on three supervised learning tasks, covering node-, edge-, and graph-level prediction.

**Node Classification.** Given the labeled node set $\mathcal{V}_L \subset \mathcal{V}$ with labels $\mathbf{Y}_L \in \mathbb{R}^C$, where each node $v_i$ is associated with a label $y_i$ from one of the $C$ classes, the goal of node classification is to train a classifier $f_\theta : v \mapsto \mathbb{R}^C$ to predict labels $\mathbf{Y}_U$ of the remaining unlabeled nodes $\mathcal{V}_U = \mathcal{V} \setminus \mathcal{V}_L$.

**Hyperedge Prediction.** Given a hypergraph $\mathcal{G}(\mathcal{V}, \mathcal{E}, \mathbf{X})$, we denote $\mathcal{E}' \subset 2^{\mathcal{V}} \setminus \mathcal{E}$ as the **target set** which typically consists of (a) unobserved hyperedges or (b) new hyperedges that will arrive in the near future. Each element in $2^{\mathcal{V}} \setminus \mathcal{E}$ is referred to as a **hyperedge candidate**, denoted by $c$, as it may belong to $\mathcal{E}'$. The hyperedge prediction task aims to train a hyperedge classifier $f'_\theta : e \mapsto \{0, 1\}$ to predict whether a candidate $c$ belongs to the target set $\mathcal{E}'$ or not.

**Hypergraph Classification.** Let $\mathcal{H}$ as the hypergraph set. Given the labeled hypergraph set $\mathcal{H}_L$ and their labels $\mathbf{Y}_L \in \mathbb{R}^C$, where each hypergraph $\mathcal{G}_i$ is assigned a label $y_i$. The hypergraph classification task aims to train a hypergraph classifier $f''_\theta : \mathcal{G} \mapsto \mathbb{R}^C$ to predict labels $\mathbf{Y}_U$ of the unlabeled hypergraphs $\mathcal{H}_U = \mathcal{H} \setminus \mathcal{H}_L$.

## 3 BENCHMARK DESIGN

In this section, we introduce the DHG-Bench in terms of datasets (Section 3.1), algorithms (Section 3.2), and research questions (Section 3.3) that guide the benchmark study.

### 3.1 BENCHMARK DATASETS

To comprehensively evaluate HNNs, we integrate 22 benchmark datasets from various domains spanning node-, edge-, and graph-level tasks. In this section, we introduce each dataset category and the corresponding data splitting strategy. Detailed descriptions are provided in Appendix A.1.

**Node-level Classification Datasets.** For the node classification task, we select 13 hypergraph datasets that cover diverse domains and characteristics. Specifically, we include 8 homophilic datasets: two co-citation networks (Cora and Pubmed (Yadati et al., 2019)); two co-authorship networks (Cora-CA and DBLP (Yadati et al., 2019)); two graphics datasets (NTU2012 and ModelNet40 (Feng et al., 2019)); and two hypergraphs that capture user interactions, namely Walmart for co-purchasing (Chien et al., 2022) and Trivago for co-clicking (Kim et al., 2023). In addition, we consider 5 heterophilic datasets, including two information networks (Actor (Li et al., 2025c) and Yelp (Chien et al., 2022)), an e-commerce network (Amazon-ratings (Li et al., 2025c)), and two social networks (Twitch-gamers and Pokec (Li et al., 2025c)). Moreover, to investigate algorithmic fairness, we include three fairness-sensitive datasets (German, Bail, and Credit (Wei et al., 2022)), which contain sensitive node attributes such as gender, race, and age. Following (Feng et al., 2019; Chien et al., 2022; Tang et al., 2025), we adopt a split of 50%/25%/25% for training, validation, and testing in the node classification task.

**Hyperedge-level Prediction Datasets.** For the hyperedge prediction task, we use 6 datasets: four widely adopted homophilic academic networks (Cora, Pubmed, Cora-CA, and DBLP-CA) (Hwang et al., 2022; Ko et al., 2025) and two newly introduced heterophilic datasets, Actor and Pokec (Li et al., 2025c), which enable a more comprehensive evaluation due to their low hyperedge homophily. Following (Hwang et al., 2022; Ko et al., 2025; Yu et al., 2025), we randomly split the hyperedges (i.e., positive samples) into training (60%), validation (20%), and test (20%) sets. In addition, we adopt negative sampling (NS) (Yadati et al., 2020; Hwang et al., 2022), which is devised to enhance the distinguishing ability of the model by introducing non-existing hyperedges as contrastive information for model training. Specifically, for each training, validation, and test set, we sample an equal number of negative examples as the positive ones. Following (Ko et al., 2025), we employ a mixed NS strategy that integrates three common heuristic methods, namely sized NS (SNS), motif NS (MNS), and clique NS (CNS) (Patil et al., 2020), to increase the diversity of negative samples.

**Hypergraph-level Classification Datasets.** For the hypergraph classification task, we consider 6 benchmark datasets introduced in (Feng et al., 2024). RHG-10 and RHG-3 are two synthetic datasets consisting of distinct high-order structural patterns (e.g., Hyper Pyramid, Hyper Flower, and Hyper Wheel). IMDB-Dir-Form and IMDB-Dir-Genre are two datasets constructed by the co-director relationship from the original IMDB dataset [1]. Steam-Player is a player-based dataset, where each

---

[1] https://www.imdb.com/

hypergraph captures tag co-occurrence relationships among games played by a user. Twitter-Friend is a social media dataset where each hypergraph represents the friendship network of a specific Twitter user. For hypergraph classification, following (Feng et al., 2024), we adopt an 80%/10%/10% train/validation/test data split.

## 3.2 Benchmark Algorithms

We integrate 17 state-of-the-art HNN algorithms across three mainstream categories: spectral-based, spatial-based, and tensor-based methods. In addition, we include MLP and two GNN-based methods, CEGCN and CEGAT (Chien et al., 2022), as baselines. Detailed descriptions are provided in Appendix A.2. We rigorously reproduce all methods according to their papers and source codes.

**Spectral-based HNNs.** Spectral-based HNNs perform message propagation and feature transformation by applying spectral convolution defined through Laplacian operators of hypergraphs (Wang et al., 2024). We implement 10 representative algorithms including HGNN (Feng et al., 2019), HyperGCN (Yadati et al., 2019), HCHA (Bai et al., 2021), LEGCN (Yang et al., 2022), HyperND (Prokopchik et al., 2022), PhenomNN (Wang et al., 2023b), SheafHyperGNN (Duta et al., 2023), HJRL (Yan et al., 2024), DPHGNN (Saxena et al., 2024), and TF-HNN (Tang et al., 2025).

**Spatial-based HNNs.** Unlike spectral methods, spatial-based HNNs focus on local connectivity without entering the spectral domain, typically learning representations through two-stage neighborhood aggregation: updating hyperedges from incident nodes and updating nodes from incident hyperedges. We incorporate 5 typical algorithms including HNHN (Dong et al., 2020), UniGNN (Huang & Yang, 2021), AllSetTransformer (Chien et al., 2022), ED-HNN (Wang et al., 2023a), and HyperGT (Liu et al., 2024). For UniGNN with multiple variants (e.g., UniGAT, UniGIN, and UniGCNII), we report only UniGCNII, the most competitive variant identified in the original paper, while our open-sourced library also supports the implementations of other variants.

**Tensor-based HNNs.** Tensor-based methods leverage tensor operations that provide a structured and effective means of capturing the complexity of hypergraph interactions (Wang et al., 2025). We select two representative algorithms: EHNN (Kim et al., 2022) and T-HyperGNN (Wang et al., 2024).

## 3.3 Research Questions

We systematically design the DHG-Bench to comprehensively evaluate the existing HNN algorithms and inspire future research. In particular, we aim to investigate the following research questions.

**RQ1: How much progress has been made by existing HNN methods?**

**Motivation and Experiment Design.** Previous research on HNNs has been limited by inconsistent experimental settings and insufficient coverage of datasets, algorithms, and tasks, thereby hindering fair and comprehensive evaluation of different methods. Given the standardized experimental environment provided by DHG-Bench, the first question is to revisit the progress of existing HNN methods and identify potential directions for enhancement. A high-quality HNN method is expected to perform consistently well across different datasets and application scenarios. To answer this question, we evaluate the performance of HNN methods on diverse, widely used hypergraph datasets across three benchmark tasks: node classification, hyperedge prediction, and hypergraph classification. Detailed experimental settings can be found in Appendix B.1.

**RQ2: How efficient are these HNN methods in terms of time and space?**

**Motivation and Experiment Design.** Training the message-passing module of HNNs makes loss computation interdependent for connected nodes, resulting in intensive computational demands and substantial memory constraints. However, the efficiency and scalability of HNN algorithms have been largely overlooked. A thorough understanding of the trade-off between computational cost and predictive performance is essential for assessing their suitability for real-time and large-scale applications. To answer this question, we perform node classification, the most widely used benchmark task, on datasets of varying scales (Cora, DBLP-CA, Yelp, and Trivago), reporting the training time to reach the best validation performance and the peak GPU memory consumption.

Table 1: Evaluation results of node classification: mean accuracy (%) ± standard deviation. The best results are shown in **bold** and the runner-ups are underlined. OOM denotes the out-of-memory issue.

| Method | Cora | Pubmed | Cora-CA | DBLP-CA | Walmart | Trivago | Actor | Gamers | Pokec | Yelp |
|---|---|---|---|---|---|---|---|---|---|---|
| MLP | 75.33±0.88 | 86.62±0.26 | 75.57±1.08 | 85.54±0.15 | 63.21±0.12 | 36.76±0.66 | 86.06±0.36 | **52.57±0.49** | 59.64±0.48 | 31.84±0.45 |
| CEGCN | 76.90±0.75 | 86.03±0.39 | 78.40±1.25 | 89.75±0.33 | 70.40±0.18 | 47.24±1.09 | 67.41±0.29 | 51.02±0.53 | 57.37±0.38 | OOM |
| CEGAT | 77.22±1.03 | 86.09±0.51 | 78.02±1.24 | 89.61±0.22 | 65.83±0.92 | OOM | 73.87±0.83 | 51.05±0.61 | 57.34±0.52 | OOM |
| HGNN | 77.90±1.17 | 86.17±0.52 | 82.84±0.46 | 91.00±0.27 | 77.12±0.12 | 57.67±1.61 | 77.83±0.37 | 52.38±0.56 | 57.87±0.76 | 33.71±0.24 |
| HyperGCN | 78.38±1.63 | 87.42±0.42 | 81.65±1.58 | 89.51±0.18 | 68.75±0.56 | 42.39±1.25 | 81.82±0.39 | 51.32±0.72 | 57.51±0.54 | 29.29±0.55 |
| HCHA | 77.84±1.23 | 86.33±0.54 | 83.01±0.58 | 91.18±0.30 | 77.66±0.18 | 52.50±3.43 | 78.30±0.47 | 52.35±0.71 | 58.19±0.45 | 33.13±0.23 |
| LEGCN | 74.36±1.03 | 87.52±0.50 | 74.59±1.04 | 85.16±0.14 | 62.98±0.09 | 33.45±1.45 | 85.34±0.45 | 51.31±0.65 | **59.66±0.63** | OOM |
| HyperND | 79.23±0.63 | 86.73±0.56 | 83.19±0.71 | 91.34±0.19 | 75.10±0.54 | 87.19±1.89 | 83.19±0.92 | 52.39±0.60 | 57.65±1.08 | OOM |
| PhenomNN | 78.97±1.41 | 87.81±0.12 | 84.05±1.05 | **91.83±0.25** | OOM | OOM | 83.14±0.49 | 51.80±0.73 | 58.43±0.92 | OOM |
| SheafHyperGNN | 79.03±0.90 | 87.10±0.47 | 84.08±0.50 | 91.09±0.31 | OOM | OOM | 85.00±0.32 | 52.07±0.53 | 59.06±0.37 | OOM |
| HJRL | 78.67±1.47 | **87.98±0.49** | 83.72±0.74 | OOM | OOM | OOM | 71.54±0.64 | 51.62±0.61 | 57.57±0.47 | OOM |
| DPHGNN | 76.40±1.36 | 86.72±0.33 | 82.13±1.13 | OOM | OOM | OOM | 83.65±0.59 | 52.36±0.59 | 58.20±0.58 | OOM |
| TF-HNN | 79.47±1.31 | 87.90±0.37 | **84.19±0.89** | 91.38±0.24 | 77.04±0.12 | **90.79±0.79** | 85.96±0.41 | 52.34±0.53 | 59.17±0.52 | **35.16±0.54** |
| HNHN | 75.24±1.38 | 85.66±1.28 | 76.51±1.34 | 85.84±0.07 | 65.21±0.28 | 53.75±1.43 | 81.20±0.36 | 51.12±0.65 | 58.55±0.93 | 25.86±0.63 |
| UniGNN | 79.41±1.24 | 87.57±0.54 | 83.49±1.58 | 91.71±0.20 | 76.26±0.58 | 36.15±0.56 | 84.61±0.46 | 52.50±0.57 | 58.56±0.73 | 31.09±0.61 |
| AllSetTransformer | 78.02±1.43 | 87.79±0.30 | 82.95±0.62 | 91.51±0.22 | **78.61±0.13** | 59.92±4.02 | 85.66±0.41 | 51.74±0.75 | 58.55±0.56 | 33.18±0.88 |
| ED-HNN | 78.58±0.52 | 87.65±0.23 | 82.98±0.93 | 91.55±0.19 | 77.90±0.21 | 75.99±2.60 | 85.77±0.46 | 50.54±0.23 | 58.68±0.40 | 34.84±0.93 |
| HyperGT | 75.57±1.11 | 86.06±0.54 | 75.42±0.62 | 84.53±0.30 | OOM | OOM | 84.43±0.47 | 51.19±0.57 | 57.73±0.76 | OOM |
| EHNN | 76.51±1.52 | 87.12±0.31 | 81.68±0.81 | 90.47±0.43 | 77.95±0.14 | OOM | **86.21±0.49** | 52.14±0.76 | 58.23±1.07 | 34.09±3.19 |
| T-HyperGNN | 74.20±1.37 | 86.28±0.62 | 75.01±1.44 | 85.44±0.14 | 73.48±0.33 | OOM | 85.32±0.48 | 51.82±0.38 | 58.82±0.49 | OOM |

**RQ3: Are existing HNN methods robust to different types of data perturbations?**

**Motivation and Experiment Design.** Real-world hypergraph data inevitably contains noise, task-irrelevant information, or even mistakes (Cai et al., 2022). A reliable HNN should maintain stable performance when exposed to such noisy data, particularly in high-stakes domains such as healthcare and finance (Cai et al., 2025), where inaccurate decisions can adversely affect individual lives or broader societal systems. Evaluating the robustness of HNNs not only reveals potential vulnerabilities in existing methods but also guides the development of more resilient models. To answer this question, we simulate realistic data perturbations from three perspectives: structure, feature, and supervision signals. For each perturbation type, we vary the noise intensity and subsequently train and test HNNs on the corresponding modified hypergraph. Detailed experimental settings are in Appendix B.4.

**RQ4: Do existing HNN methods yield unbiased predictions across demographic groups?**

**Motivation and Experiment Design.** Fairness has recently emerged as a critical concern in graph machine learning (GML) (Dong et al., 2023). Prior studies have shown that representations learned by GNNs can result in biased predictions, often favoring certain demographic groups defined by sensitive attributes (e.g., gender and race) (Ling et al., 2023; Zhu et al., 2024; Yang et al., 2024). Such bias hinders the deployment of GML models in high-stakes applications such as crime prediction (Suresh & Guttag, 2019) and credit evaluation (Yeh & Lien, 2009). Despite its importance, fairness in deep hypergraph learning has received little attention. To the best of our knowledge, this work presents the first benchmark evaluation of fairness in this context, which is crucial for developing ethically sound and trustworthy HNN models. To answer this question, we conduct node classification on three fairness-sensitive datasets (German, Bail, and Credit (Wei et al., 2022)), each of which contains demographic-sensitive attributes. We assess algorithmic fairness using two widely adopted group fairness metrics: demographic parity ($\Delta_{DP}$) (Dwork et al., 2012), and equalized odds ($\Delta_{EO}$) (Hardt et al., 2016). The detailed descriptions of the two metrics can be found in Appendix B.5.

## 4 EXPERIMENT RESULTS AND ANALYSIS

### 4.1 EFFECTIVENESS EVALUATION (**RQ1**)

To investigate the effectiveness of existing HNNs, we compare their performance across benchmark tasks at the node, edge, and graph levels. Due to space constraints, additional node classification results on NTU2012, ModelNet40, and Ratings (Table A5), as well as the complete results of hyperedge prediction (Table A6) and hypergraph classification (Table A7), are available in Appendix C.1.

#### 4.1.1 EFFECTIVENESS ON NODE CLASSIFICATION TASK

**Results** (Table 1 and Table A5). ❶ Across diverse datasets, HNNs generally outperform both CEGCN and CEGAT, suggesting that naively extending GNNs to hypergraphs via clique expansion

disrupts high-order structures and degrades predictive performance. This highlights the necessity of designing neural architectures with dedicated high-order message passing. ❷ HNNs achieve notable improvements over MLP on homophilic datasets, but on heterophilic datasets, most HNNs even underperform MLP, which only leverages node features. This reveals the adverse impact of heterophilic connections on hypergraph representation learning and underscores the need to rethink HNN design in such settings. ❸ TF-HNN consistently ranks among the top-performing methods across diverse datasets, achieving optimal or near-optimal results. Moreover, unlike other advanced HNNs (e.g., PheomNN, DPHGNN, and HyperGT) that fail on large-scale datasets due to out-of-memory issues, TF-HNN remains scalable. These findings underscore the promise of its decoupled architecture for enhanced generalization and scalability.

### 4.1.2 EFFECTIVENESS ON HYPEREDGE PREDICTION TASK

**Results** (Table A6). ❶ Advanced HNN methods that generally achieve superior performance on node classification fail to maintain the same level of competitiveness in hyperedge prediction. Specifically, the two earliest methods, HGNN and HyperGCN, along with the tensor-based EHNN introduced in 2022, collectively achieve all the best results and the majority of second-best results across the six hyperedge prediction datasets. In contrast, recent HNNs (e.g., ED-HNN, HJRL, DPHGNN, TF-HNN) often show a notable performance gap compared to the above three. For example, on DBLP-CA, TF-HNN achieves an AUROC of 75.70% and an AP of 74.97%, which are 13.76% and 16.70% lower than those of the best-performing model, HyperGCN. ❷ Across hyperedge prediction benchmarks, HNN algorithms display considerable performance divergence depending on the dataset, and none consistently deliver the best results. For instance, while EHNN achieves state-of-the-art performance on Cora and Pubmed, it obtains only 77.83% AUROC on Cora-CA, ranking 11th among 17 HNNs and 14.90% lower than the top-performing HyperGCN.

### 4.1.3 EFFECTIVENESS ON HYPERGRAPH CLASSIFICATION TASK

**Results** (Table A7). ❶ HNN algorithms perform markedly better on synthetic datasets than on real-world ones. On RHG-10, most models achieve over 90% accuracy and Macro-F1, and on RHG-3, many even exceed 98%. In contrast, on real-world datasets, accuracies rarely surpass 70%, reflecting the structural complexity of real hypergraphs. This gap underscores the need for more realistic and challenging benchmarks to rigorously evaluate hypergraph classification. ❷ HNN methods generally outperform GNN-based approaches built on clique expansion, as the latter often distorts global hypergraph structures, whereas higher-order message passing in HNNs preserves these dependencies and enhances discriminative power. ❸ HNNs' performance varies considerably across datasets, with no method demonstrating consistent superiority. For instance, while DPHGNN achieves the best accuracy on IMDB-Dir-Form, it falls to 11th on IMDB-Dir-Genre and 14th on Steam-Player across all evaluated HNNs, underscoring the substantial impact of dataset characteristics on model performance. ❹ Many HNN methods fail to achieve a desirable trade-off between accuracy and Macro-F1. For example, on the Twitter dataset, HNHN achieves 58.47% accuracy (third highest among all HNN models) but only 39.40% Macro-F1, the lowest overall.

---

**Key Insights for RQ1:** HNN algorithms display varying levels of effectiveness across predictive tasks. While advanced HNNs achieve strong results on node-level tasks, they often fail to deliver superior performance on edge- and graph-level tasks. Moreover, the predictive capability of HNNs is highly sensitive to dataset characteristics, with data heterophily substantially impairing learning on hypergraphs. These findings highlight the need for future research to enhance the generalization and adaptability of hypergraph models across diverse tasks and datasets.

---

### 4.2 EFFICIENCY AND SCALABILITY EVALUATION (RQ2)

**Results** (Figure 1). ❶ CEGCN and CEGAT face scalability challenges on large datasets (e.g., Yelp and Trivago), where clique expansion produces dense edges and leads to significant training memory overhead. ❷ Most advanced HNN methods struggle to achieve a satisfactory balance between model utility and efficiency. For example, on the Yelp dataset, ED-HNN and EHNN provide only marginal accuracy gains over the simple HGNN, yet their training times are over 9× and 23× longer, respectively, reflecting a substantial rise in computational cost. In addition, many HNNs suffer from memory bottlenecks on large-scale datasets. Specifically, on Yelp, 8 out of 17 methods encounter

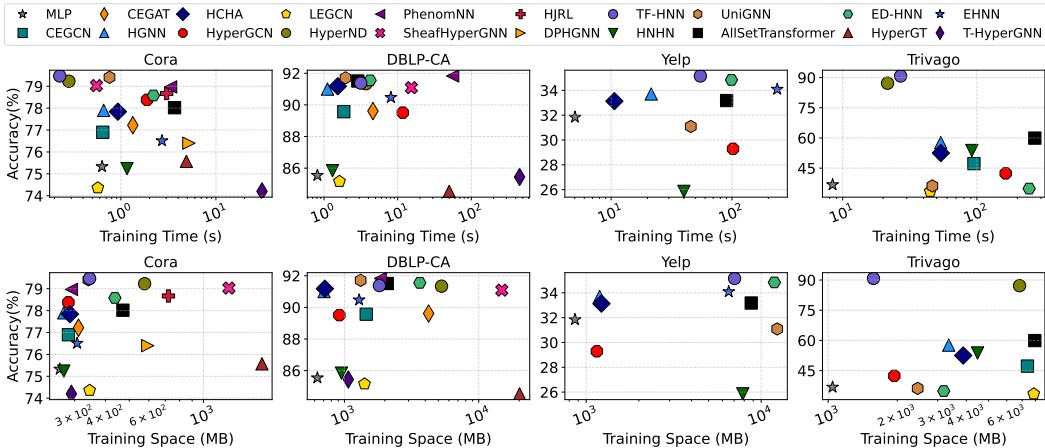

Figure 1: Training time and space analysis on Cora, DBLP-CA, Yelp, and Trivago.

out-of-memory (OOM) issues. On Trivago, although 10 HNNs remain computationally scalable, most fail to deliver satisfactory predictive performance. Only TF-HNN (90.79%) and HyperND (87.19%) achieve accuracy above 60%. This may result from the intricate patterns of large-scale graphs. ❸ Tensor-based approaches exhibit more pronounced efficiency and scalability limitations than the other two kinds of methods. T-HyperGNN can only scale to the medium-sized DBLP-CA dataset, where it runs approximately 406 times slower than the fastest method, HGNN. Moreover, on Yelp, EHNN incurs the longest training time and fails to scale to the large-scale Trivago dataset. ❹ Among all evaluated methods, TF-HNN generally achieves a superior trade-off between utility and both time and space efficiency. For example, on the large-scale Trivago dataset, it achieves the best predictive performance with no more than 1.6 GB of memory and under 30 seconds of runtime, ranking first in memory efficiency and second in training time among all HNN methods.

> **Key Insights for RQ2:** Most existing HNN algorithms, when applied to large-scale datasets, either suffer from efficiency and scalability issues or fail to deliver satisfactory utility. Investigating decoupled architectures that separate high-order information propagation from training modules presents a promising avenue for achieving efficient, scalable, and high-performing HNNs.

## 4.3 ROBUSTNESS EVALUATION (RQ3)

In this section, we assess HNN robustness by simulating structural, feature, and supervision perturbations, as detailed in Appendix B.4. While our experiments primarily focus on the node classification task due to space limits, DHG-Bench supports flexible extension to other tasks. We evaluate 10 representative models on four datasets (Cora, Pubmed, Actor, and Pokec). The results on Pubmed and Pokec (Figures A2, A3, and A4) are provided in Appendix C.2.

### 4.3.1 ROBUSTNESS ANALYSIS WITH RESPECT TO STRUCTURE PERTURBATIONS

**Results** (Figure 2 and Figure A2). ❶ Most HNN algorithms exhibit strong robustness against random structural noise, experiencing only marginal performance drops or even remaining nearly unaffected under high perturbation rates. For example, when 90% of hyperlinks are randomly removed from Cora, 7 out of 10 methods degrade by less than 7%. Similarly, when 90% of random hyperlinks are injected into Actor, only 2 models show a noticeable decline in performance. ❷ Spectral-based approaches are generally more vulnerable to structural perturbations. On Pubmed, for instance, increasing the ratio of noisy hyperlinks results in a pronounced performance decline across four spectral-based methods (HGNN, PhenomNN, DPHGNN, and TF-HNN), whereas most other methods remain stable. This may be because spectral methods rely on the hypergraph's global eigenstructure, which is highly sensitive to topological noise. ❸ The robustness of HNN algorithms varies with both the type of structural perturbation (deletion vs. addition) and the choice of dataset. For example, on the Actor dataset, SheafHyperGNN suffers substantial performance degradation under hyperlink

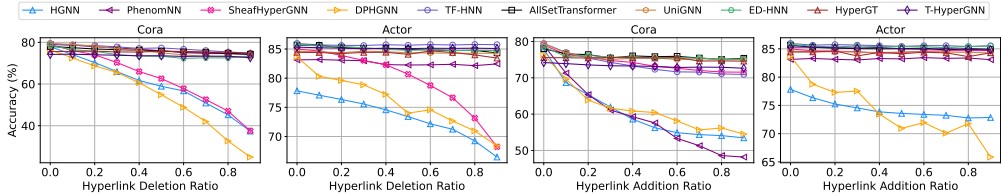

Figure 2: Structure robustness analysis on Cora and Actor.

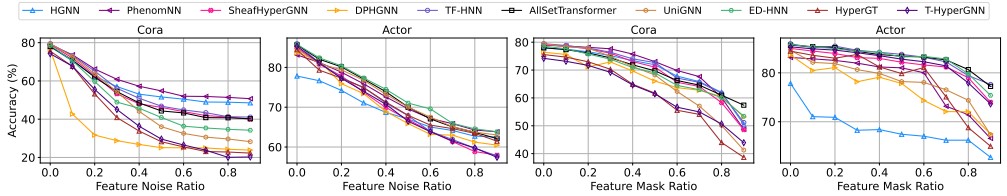

Figure 3: Feature robustness analysis on Cora and Actor.

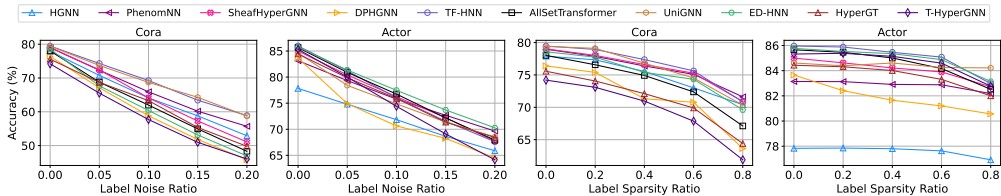

Figure 4: Supervision robustness analysis on Cora and Actor.

deletion but demonstrates strong robustness under hyperlink addition. In another case, PhenomNN exhibits strong robustness on Cora in the addition scenario while showing the opposite trend on Actor.

### 4.3.2 ROBUSTNESS ANALYSIS WITH RESPECT TO FEATURE PERTURBATIONS

**Results** (Figure 3 and Figure A3). ❶ Feature perturbations under equal noise or sparsity levels result in greater performance degradation than structural ones, indicating a more critical role of node features in model prediction. ❷ With increasing noise intensity, model accuracy decreases sharply at the beginning and then stabilizes, as highly corrupted features approximate randomness and lose predictive utility. ❸ As the feature masking rate increases, model performance degrades progressively faster, with a slow decline at low ratios and a sharp drop under high sparsity. ❹ Compared to feature sparsity, feature noise poses a greater challenge for HNN algorithms, with equivalent levels of noise typically resulting in lower predictive accuracy across different datasets.

### 4.3.3 ROBUSTNESS ANALYSIS WITH RESPECT TO SUPERVISION PERTURBATIONS

**Results** (Figure 4 and Figure A4). ❶ As noise intensity increases or supervision becomes sparser, all models show a clear downward trend in performance, with label noise exerting a more pronounced impact. ❷ Increasing label noise generally causes a rapid yet steady decline in performance, which appears approximately linear in most cases. ❸ The impact of supervision sparsity is modest at lower levels but intensifies at higher ratios, resulting in an accelerating decline in model performance. This trend highlights the challenges faced by current HNNs in low-label scenarios. ❹ Label noise and sparsity tend to degrade performance more substantially on homophilic datasets than on heterophilic ones, reflecting the reliance of model predictions on data homophily.

> **Key Insights for RQ3:** Most HNN algorithms demonstrated remarkable robustness to random structural noise, but are considerably more vulnerable to feature perturbations. In addition, at the label level, even simple small-scale poisoning attacks can substantially degrade predictive

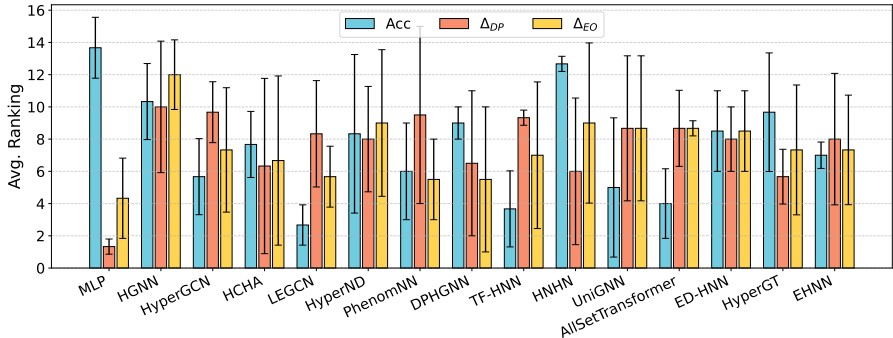

Figure 5: Average rankings on Acc, $\Delta_{DP}$, $\Delta_{EO}$ across the German, Bail, and Credit datasets, where lower values indicate better ranks (ascending order).

performance, and HNNs face significant challenges under extreme label sparsity. These findings underscore the need for designing robust HNN architectures or training techniques capable of providing strong defenses against diverse forms of noisy data.

## 4.4 FAIRNESS EVALUATION (RQ4)

In this section, we analyze algorithmic fairness and report full quantitative results in terms of accuracy (Acc), $\Delta_{DP}$, and $\Delta_{EO}$ in Table A8 of Appendix C.3. To better illustrate the strengths and limitations of each algorithm, we present Figure 5, which shows their average rankings across the three metrics on datasets where they can run, considering only HNNs executable on at least two datasets.

**Results** (Figure 5 and Table A8). ❶ While HNN algorithms achieve higher predictive performance, they generally suffer from more severe fairness issues compared to MLP, which is free from message passing. Figure 5 shows that MLP ranks best on the two fairness metrics but worst on accuracy. For example, on the Credit dataset, MLP achieves lower $\Delta_{DP}$ and $\Delta_{EO}$ values than HCHA, the fairest among the evaluated HNN models, as shown in Table A8. ❷ The fairness performance of HNN algorithms varies considerably across datasets, with no method achieving consistently superior performance on all benchmarks. For instance, Table A8 illustrates that while HCHA achieves the best fairness performance on the Credit dataset across both metrics, its $\Delta_{DP}$ and $\Delta_{EO}$ rank as the second- and third-worst, respectively, on the German dataset. Moreover, Figure 5 shows that most algorithms exhibit substantial variance in their rankings, further highlighting the instability of fairness across datasets. ❸ HNN algorithms show inconsistent behavior across fairness metrics, and strong performance on one does not guarantee superiority on another. For example, on the Bail dataset, although HNHN achieves the lowest $\Delta_{DP}$ among all HNN methods, its $\Delta_{EO}$ ranks as the third worst among the 17 HNN models.

**Key Insights for RQ4:** Existing HNN algorithms tend to produce more biased predictions than MLPs, indicating that high-order information propagation may exacerbate the amplification of biases from sensitive information. Moreover, fairness performance varies substantially across datasets and metrics. These findings highlight the need for developing debiased algorithms that can achieve stronger fairness across diverse high-stakes real-world applications.

## 5 A GUIDE FOR PRACTITIONERS

Based on the comprehensive benchmarking results and analyses, we provide practical guidance for selecting suitable HNN models for new tasks, organized by task type.

**Node-level prediction tasks.** We recommend TF-HNN as the first-choice model. Across a wide range of datasets, TF-HNN consistently achieves top-ranked node classification performance, demonstrating its strong ability to learn highly discriminative node representations. Moreover, its training-free message-passing architecture offers substantial efficiency and scalability benefits, making it well-

suited for large-scale or resource-constrained applications. Importantly, our experiments show that, compared with other HNNs, TF-HNN does not exhibit pronounced weaknesses in robustness or fairness, making it a reliable choice for most node-level scenarios.

**Edge-level prediction tasks.** We suggest starting with EHNN, HGNN, and HyperGCN. Together, these models account for most of the best and second-best results on hyperedge prediction benchmarks. Their performance, however, varies across homophilic and heterophilic settings: on homophilic datasets, EHNN and HyperGCN generally perform better; on heterophilic datasets, HGNN and EHNN tend to yield stronger results. Our robustness analysis further indicates that HGNN is more sensitive to structural perturbations, and may therefore be less dependable under distribution shifts or noisy hypergraph structures. As a result, EHNN and HyperGCN are generally safer and more robust defaults, while HGNN should be chosen with awareness of dataset stability.

**Graph-level prediction tasks.** No single architecture consistently outperforms all others across datasets and evaluation metrics in hypergraph classification. Nonetheless, HJRL, DPHGNN, and AllSetTransformer frequently appear among the top-performing models, reflecting their strong ability to capture and discriminate global structural patterns that drive hypergraph-level prediction. However, our robustness experiments reveal that DPHGNN can be sensitive to structural and feature perturbations, and practitioners are therefore advised to carefully assess its stability before deployment. Among these models, AllSetTransformer often provides a more favorable utility–efficiency trade-off, making it particularly appealing in computationally constrained environments.

## 6 CONCLUSION AND FUTURE DIRECTIONS

This paper introduces DHG-Bench, the first comprehensive benchmark for deep hypergraph learning, which integrates and compares 17 representative HNNs across 22 hypergraph datasets encompassing various domains, sizes, and structural properties, under consistent experimental settings. We comprehensively evaluate the effectiveness, efficiency, robustness, and fairness of HNN algorithms, and our analysis reveals the strengths and weaknesses of different HNNs in a wide range of scenarios, offering valuable insights into their practical applicability and design trade-offs. Furthermore, we develop and release a package, DHG-Bench, that includes all experimental protocols, baseline algorithms, datasets, and reproducibility scripts to facilitate future research. Drawing upon our empirical analyses, we point out some promising future directions for the deep hypergraph learning community.

- **Developing adaptive HNN methods for diverse datasets and tasks.** Our experiments in Section 4.1 reveal that existing HNN architectures show substantial performance disparities across datasets and tasks, limiting their applicability in diverse scenarios. Future research should focus on designing adaptive HNN architectures and training techniques that can better accommodate the unique characteristics of datasets from different domains and varying task granularities, thereby enhancing the generalization ability of HNNs.

- **Improving the efficiency of HNN methods.** Observations in Section 4.2 indicate that many advanced HNN methods fail to balance efficiency and predictive performance, and often run out of memory on large-scale datasets. As the size of hypergraphs continues to grow exponentially, a key area of future research is the reduction of memory and computational complexity in HNN algorithms while maintaining satisfactory model utility. Inspired by the favorable efficiency–effectiveness trade-off achieved by TF-HNN, it would be promising to devise more powerful decoupled architectures specifically tailored for HNN.

- **Developing more robust HNN methods.** Our experimental results in Section 4.3 show that HNN algorithms are affected by different types of data perturbations and are particularly vulnerable to those at the feature and supervision levels. Future work should emphasize enhancing the robustness of HNNs to resist varying degrees of data noise and even adversarial attacks, thereby ensuring reliable performance in a wide range of industrial applications.

- **Developing fairness-aware HNN methods.** Empirical evidence in Section 4.4 suggests that HNNs are more prone to biased predictions than traditional MLPs. Future research should investigate the theoretical mechanisms through which high-order message passing exacerbates fairness issues and then develop fairness-aware HNN methods that mitigate such discriminatory behavior. Progress in this direction is essential to ensure the safe adoption of HNNs in high-stakes real-world applications such as crime prediction and credit evaluation.

## ACKNOWLEDGEMENTS

Xiaoyang Wang is supported by ARC DP240101322 and ARC DP230101445. Wenjie Zhang is supported by Australian Research Council Centre of Excellence for Mathematical Modelling of Cellular Systems CE230100001.

## ETHICS STATEMENT

This work does not raise any specific ethical concerns. All datasets used in our experiments are publicly available and have been released for academic purposes. None of the datasets contains personally identifiable information or offensive content.

## REPRODUCIBILITY STATEMENT

We describe our data splitting strategy in Section 3.1, the experiment design for multi-dimensional analysis in Section 3.3, and detailed experimental setups in Appendix B. All datasets, algorithm implementations, and hyperparameter configurations are publicly available at https://github.com/Coco-Hut/DHG-Bench.

- The datasets are provided in the repository as a compressed file, data.zip, and data loading and preprocessing are handled by the code in the lib_dataset folder.
- The implementation of the training and evaluation pipeline for algorithms is available in the lib_utils folder in the repository.
- Additional instructions for reproducing experiments are included in the README.md.

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

# APPENDIX

## A    DATASETS AND ALGORITHMS

### A.1    BENCHMARK DATASETS

Table A1: Statistics of the standard node-level datasets: $|e|$ denotes the hyperedge size, while $\mathcal{H}_{\text{edge}}$ indicates the hyperedge homophily ratio introduced in (Li et al., 2025c). $I_{node}$, $P_{node}$, and $H_{node}$ indicate the isolated nodes, the nodes involved only in pairwise interactions, and the nodes participating in higher-order interactions, respectively.

| Dataset | # Nodes | # Edges | # Features | Avg. $|e|$ | $\mathcal{H}_{\text{edge}}$ | # $I_{node}$ | # $P_{node}$ | # $H_{node}$ | # Classes |
|---|---|---|---|---|---|---|---|---|---|
| Cora | 2,708 | 1,579 | 1,433 | 3.03 | 0.75 | 1,274 | 205 | 1,229 | 7 |
| Pubmed | 19,717 | 7,963 | 500 | 4.35 | 0.78 | 15,877 | 201 | 3,639 | 3 |
| Cora-CA | 2,708 | 1,072 | 1,433 | 4.28 | 0.78 | 320 | 278 | 2,110 | 7 |
| DBLP-CA | 41,302 | 22,363 | 1,425 | 4.45 | 0.87 | 0 | 3,876 | 37,426 | 6 |
| NTU2012 | 2,012 | 2,012 | 100 | 5.00 | 0.79 | 0 | 0 | 2,012 | 67 |
| ModelNet40 | 12,311 | 12,311 | 100 | 5.00 | 0.87 | 0 | 0 | 12,311 | 40 |
| Walmart | 88,860 | 69,906 | 100 | 6.59 | 0.60 | 0 | 3,295 | 85,565 | 11 |
| Trivago | 172,738 | 233,202 | 300 | 3.12 | 0.98 | 0 | 25,532 | 147,206 | 160 |
| Actor | 16,255 | 10,164 | 50 | 5.25 | 0.46 | 563 | 600 | 15,092 | 3 |
| Ratings | 22,299 | 2,090 | 111 | 3.10 | 0.37 | 19,175 | 176 | 2,948 | 5 |
| Gamers | 16,812 | 2,627 | 7 | 6.23 | 0.49 | 456 | 624 | 15732 | 2 |
| Pokec | 14,998 | 2,406 | 65 | 2.29 | 0.45 | 11,798 | 1,948 | 1,252 | 2 |
| Yelp | 50,758 | 679,302 | 1,862 | 6.66 | 0.29 | 0 | 19 | 50,739 | 9 |

Table A2: Statistics of fairness-sensitive datasets. **Sens** denotes the sensitive attribute.

| Dataset | # Nodes | # Edges | # Features | Sens | Label |
|---|---|---|---|---|---|
| German | 1,000 | 1,000 | 27 | Gender | Credit status |
| Bail | 18,876 | 18,876 | 18 | Race | Bail decision |
| Credit | 30,000 | 30,000 | 13 | Age | Future default |

Table A3: Statistics of graph-level datasets. Avg. $|\mathcal{V}|$, $|\mathcal{E}|$, and $|e|$ represent the average number of nodes, hyperedges, and hyperedge sizes, respectively.

| Dataset | # Hypergraphs | Avg. $|\mathcal{V}|$ | Avg. $|\mathcal{E}|$ | Avg. $|e|$ | # Classes |
|---|---|---|---|---|---|
| RHG-10 | 2,000 | 31.3 | 29.8 | 5.2 | 10 |
| RHG-3 | 1,500 | 35.5 | 17.9 | 6.9 | 3 |
| IMDB-Dir-Form | 1,869 | 15.7 | 39.2 | 3.7 | 3 |
| IMDB-Dir-Genre | 3,393 | 17.3 | 36.4 | 3.8 | 3 |
| Steam-Player | 2,048 | 13.8 | 46.4 | 4.5 | 2 |
| Twitter-Friend | 1,310 | 21.6 | 84.3 | 4.3 | 2 |

We adopt 22 publicly available benchmark datasets to comprehensively evaluate HNN algorithms. The statistics of node-level datasets, fairness-sensitive datasets, and graph-level datasets are reported in Tables A1, A2, and A3, respectively. Detailed descriptions of these datasets are provided below.

- **Cora/Pubmed/Cora-CA/DBLP-CA** (Yadati et al., 2019): Cora and Pubmed are co-citation networks where nodes represent papers and hyperedges connect papers cited together. Cora-CA and DBLP-CA are co-authorship hypergraphs, with nodes as papers and hyperedges linking all papers co-authored by the same author. Node features are Bag-of-Words (BoW) (Zhang et al., 2010) representations of the documents, and labels indicate paper categories.

- **NTU2012/ModelNet40** (Feng et al., 2019): The ModelNet40 and the NTU2012 are two computer vision and graphics datasets. ModelNet40 contains 12,311 3D objects from 40 popular categories,

while NTU2012 consists of 2,012 3D shapes from 67 categories. For each object, features are extracted using both the Group-View Convolutional Neural Network (GVCNN)(Feng et al., 2018) and the Multi-View Convolutional Neural Network (MVCNN)(Su et al., 2015). Following (Feng et al., 2019), we construct hyperedges by aggregating the nearest neighbors of each node based on Euclidean distance.

- **Walmart** (Chien et al., 2022): The Walmart dataset models a hypergraph where nodes represent products and hyperedges capture sets of products purchased together. Node labels indicate product categories. Following (Chien et al., 2022), each node feature is a 100-dimensional vector obtained by adding Gaussian noise $\mathcal{N}(0, \sigma^2 I)$ with $\sigma = 0.6$ to one-hot encodings of the labels.

- **Trivago** (Kim et al., 2023): Trivago is a hotel-web search hypergraph where each node indicates a hotel, and each hyperedge corresponds to a user. If a user (hyperedge) has visited the website of a particular hotel (node), the corresponding node is added to the respective user hyperedge. Furthermore, each hotel's class is labeled based on the country in which it is located.

- **Actor** (Li et al., 2025c): The actor co-occurrence network is derived from a heterogeneous movie-actor-director-writer network [2], capturing intricate collaborations within films. Nodes represent individuals involved in film production (actors, directors, and writers), and hyperedges denote their joint participation in a single film. Node attributes are extracted from Wikipedia keywords, and labels indicate each individual's specific role.

- **Amazon-ratings (Ratings)** (Li et al., 2025c): This dataset, sourced from the Amazon co-purchasing network in the SNAP repository (Leskovec, 2014), includes products like books, music CDs, DVDs, and VHS tapes. Nodes represent individual products, and hyperedges link those frequently purchased together. The task is to predict each product's average user rating, classified into ten levels. Node features are extracted using the BoW technique applied to product descriptions (Juluru et al., 2021).

- **Twitch-gamers (Gamers)** (Li et al., 2025c): The Twitch-gamers dataset is a connected undirected hypergraph representing user interactions on the Twitch streaming platform. Nodes denote user accounts, and hyperedges are formed based on mutual follows within specific timeframes. Each node is associated with features such as view counts, timestamps, language preferences, activity duration, and inactivity status. The goal is to predict whether a channel hosts explicit content (binary classification).

- **Pokec** (Li et al., 2025c): The Pokec dataset is derived from Slovakia's largest online social networking platform and is used to model social relationships and attributes. Nodes represent individual users, and hyperedges correspond to each user's full set of friends. Node labels indicate user-reported gender, while node features are extracted from profile information, including age, hobbies, interests, education level, region, etc.

- **Yelp** (Chien et al., 2022): The Yelp dataset is a hypergraph where nodes represent restaurants and hyperedges link those visited by the same user. Node labels denote average star ratings (1.0–5.0 in 0.5 steps). Features include geographic coordinates, one-hot encodings of city/state, and BoW vectors from the top-1000 restaurant name tokens.

- **German** (Wei et al., 2022): The nodes in the dataset represent clients in the German Bank, and hyperedges are constructed by linking individuals with the most similar credit accounts to each person in the dataset. The task is to classify credit risk levels as high or low based on the sensitive attribute "gender" (Male/Female).

- **Bail** (Wei et al., 2022): The nodes in the datasets are defendants who got released on bail at the U.S state courts during 1990- 2009 (Jordan & Freiburger, 2015). Hyperedges are constructed based on the similarity of past criminal records among individuals. The task is to classify whether defendants are on bail or not with the sensitive attribute "race" (White/Black).

- **Credit** (Wei et al., 2022): The nodes in the dataset represent credit card users, and hyperedges are formed based on the similarity of users' spending and payment patterns. The task is to classify the default status with the sensitive attribute "age" ($<25$ / $>25$).

- **RHG-10/RHG-3** (Feng et al., 2024): RHG-10 dataset encompasses ten distinct synthetic factor hypergraph structures (i.e., Hyper Flower, Hyper Pyramid, Hyper Checked Table, Hyper Wheel, Hyper Lattice, Hyper Windmill, Hyper Firm Pyramid, Hyper RChecked Table, Hyper Cycle, and

---

[2]https://www.aminer.org/lab-datasets/soinf/

Figure A1: A timeline of the representative hypergraph neural networks.

Hyper Fern). To evaluate the algorithm's ability to recognize significant high-order structures, the RHG-3 dataset is constructed by randomly generating hypergraphs for three distinctively various hypergraph structures: Hyper Pyramid, Hyper Checked Table, and Hyper Wheel.

- **IMDB-Dir-Form/IMDB-Dir-Genre** (Feng et al., 2024): These two datasets contain hypergraphs constructed by the co-director relationship from the original IMDB dataset. The director of each movie is a hypergraph. "Form" included in the dataset's name indicates that the movie category is identified by its form, like animation, documentary, and drama. "Genre" denotes that the movie is classified by its genres, like adventure, crime, and family.

- **Steam-Player** (Feng et al., 2024): The Steam-Player dataset is a player dataset where each player is a hypergraph. The vertex is the games played by the player, and the hyperedge is constructed by linking the games with shared tags. The target of the dataset is to identify each user's preference: single-player game or multiplayer game.

- **Twitter-Friend** (Feng et al., 2024): The Twitter-Friend dataset is a social media dataset. Each hypergraph is the friends of a specified user. The hyperedge is constructed by linking the users who are friends. The label associated with the hypergraph is to identify whether the user posted the blog about "National Dog Day" or "Respect Tyler Joseph".

## A.2 BENCHMARK ALGORITHMS

Figure A1 illustrates 17 HNN algorithms integrated into our DHG-Bench, including 10 spectral-based, 5 spatial-based, and 2 tensor-based methods. We introduce these methods in detail below.

### A.2.1 SPECTRAL-BASED ALGORITHMS

- **HGNN** (Feng et al., 2019): HGNN is a framework for representation learning that extends spectral convolution to hypergraphs. By leveraging the hypergraph Laplacian and approximating spectral filters with truncated Chebyshev polynomials, it effectively captures high-order correlations inherent in complex data.

- **HyperGCN** (Yadati et al., 2019): HyperGCN approximates each hyperedge of the hypergraph by a set of pairwise edges connecting the vertices of the hyperedge, and treats the learning problem as a graph learning task on the approximated graph.

- **HCHA** (Bai et al., 2021): HCHA is a hypergraph neural network that introduces two end-to-end trainable operators: hypergraph convolution and hypergraph attention. Hypergraph convolution efficiently propagates information by leveraging high-order relationships and local clustering structures, with standard graph convolution shown as a special case. Hypergraph attention further enhances representation learning by dynamically adjusting hyperedge connections through an attention mechanism, enabling task-relevant information aggregation and yielding more discriminative node embeddings.

- **LEGCN** (Yang et al., 2022): LEGCN is a hypergraph learning model based on the Line Expansion (LE). By modeling vertex-hyperedge pairs, LEGCN bijectively transforms a hypergraph into a simple graph, preserving the symmetric co-occurrence structure and avoiding information loss. This enables existing graph learning algorithms to operate directly on hypergraphs.

- **HyperND** (Prokopchik et al., 2022): HyperND develops a nonlinear diffusion process on hypergraphs that propagates both features and labels along the hypergraph structure. The novel diffusion incorporates a broad class of nonlinearities to increase the modeling capability, and the limiting point serves as a node embedding from which we make predictions with a linear model.

- **PhenomNN** (Wang et al., 2023b): PhenomNN is a hypergraph learning framework grounded in a family of expressive, parameterized hypergraph-regularized energy functions. It formulates node embeddings as the minimizers of these energy functions, which are optimized jointly with a parameterized classifier through a supervised bilevel optimization process. This approach provides a principled way to model high-order relationships in hypergraphs while enabling end-to-end training.

- **SheafHyperGNN** (Duta et al., 2023): SheafHyperGNN introduces a cellular sheaf framework for hypergraphs, enabling the modeling of complex dynamics while preserving their higher-order connectivity. Then, it generalizes the two commonly used hypergraph Laplacians to incorporate the richer structure sheaves offer and constructs two powerful neural networks capable of inferring and processing hypergraph sheaf structure.

- **HJRL** (Yan et al., 2024): HJRL introduces a novel cross expansion method, which transforms both hypervertices and edges of a hypergraph to vertices in a standard graph. Then, a joint learning model is proposed to embed both hypervertices and hyperedges into a shared representation space. In addition, the algorithm employs a hypergraph reconstruction objective to preserve structural information in the model.

- **DPHGNN** (Saxena et al., 2024): DPHGNN is a hybrid framework designed for effective feature representation in resource-constrained hypergraph settings. It introduces equivariant operator learning to capture lower-order semantics by inducing topology-aware inductive biases. It employs a dual-layered feature update mechanism: a static update layer provides spectral biases and relational features, while a dynamic update layer fuses explicitly aggregated features from the underlying topology into the hypergraph message-passing process.

- **TF-HNN** (Tang et al., 2025): TF-HNN is the first model to decouple hypergraph structural processing from model training, substantially improving training efficiency. Specifically, it introduces a unified, training-free message-passing module (TF-MP-Module) by identifying feature aggregation as the core operation in HNNs. The TF-MP-Module removes learnable parameters and nonlinear activations, and compresses multi-layer propagation into a single step, offering a simplified and efficient alternative to existing architectures.

### A.2.2 SPATIAL-BASED ALGORITHMS

- **HNHN** (Dong et al., 2020): HNHN is a hypergraph convolution network with nonlinear activation functions applied to both hypernodes and hyperedges, combined with a normalization scheme that can flexibly adjust the importance of high-cardinality hyperedges and high-degree vertices depending on the dataset.

- **UniGNN** (Huang & Yang, 2021): UniGNN is a unified message-passing framework that generalizes standard GNNs to hypergraphs. It models the two-stage aggregation process by first computing hyperedge representations using a permutation-invariant function over the features of incident vertices, and then updating each vertex by aggregating its associated hyperedge representations. This formulation enables seamless adaptation of existing GNN architectures to hypergraph structures.

- **AllSetTransformer** (Chien et al., 2022): AllSetTransformer is a novel HNN paradigm that implements each layer as a composition of two multiset functions. By incorporating the Set Transformer (Lee et al., 2019) into its architecture, it achieves greater modeling flexibility and enhanced expressive power.

- **ED-HNN** (Wang et al., 2023a): ED-HNN is an architecture designed to approximate any continuous, permutation-equivariant hypergraph diffusion operator. The model is efficiently implemented by combining the star expansion (bipartite representation) of hypergraphs with standard message-passing neural networks, and supports scalable training via shared weights across layers.

- **HyperGT** (Liu et al., 2024): HyperGT is a Transformer-based HNN architecture designed to capture global correlations among nodes and hyperedges. To preserve local structural information, it incorporates incidence-matrix-based positional encoding and a structure regularization term. These designs enable comprehensive hypergraph representation learning by jointly modeling global interactions and local connectivity patterns.

### A.2.3 TENSOR-BASED ALGORITHMS

- **EHNN** (Kim et al., 2022): EHNN is the first framework to realize equivariant GNNs for general hypergraph learning. It establishes a connection between sparse hypergraphs and dense, fixed-order tensors, enabling the design of a maximally expressive equivariant linear layer. To ensure scalability and generalization to arbitrary hyperedge orders, EHNN further introduces hypernetwork-based parameter sharing.

- **T-HyperGNN** (Wang et al., 2024): T-HyperGNN is a general framework that integrates tensor hypergraph signal processing (t-HGSP) (Pena-Pena et al., 2023) to encode hypergraph structures using tensors. It models node interactions through multiplicative interaction tensors, elevating aggregation from traditional linear operations to higher-order polynomial mappings, thereby enhancing expressive power. To ensure scalability, T-HyperGNN introduces tensor-message-passing by exploiting tensor sparsity, enabling efficient processing of large hypergraphs with computational and memory costs comparable to matrix-based HNNs.

In addition, we include MLP and two GNN-based methods, CEGCN and CEGAT (Chien et al., 2022), as baselines in our comparative study. Both CEGCN and CEGAT are expansion-based approaches that transform a hypergraph into a pairwise graph via clique expansion (Zhou et al., 2006), where each hyperedge is converted into a clique over its incident nodes. Specifically, CEGCN applies GCN (Kipf & Welling, 2017) to the expanded graph, while CEGAT employs GAT (Veličković et al., 2018) to model node importance within the cliques.

## B DETAILS OF THE EXPERIMENTAL SETTINGS

### B.1 GENERAL EXPERIMENTAL SETTINGS

We strive to follow the original implementations of various HNN methods from their respective papers or source codes and integrate them into a unified training and evaluation framework. All parameters are randomly initialized. We use the cross-entropy loss function (Mao et al., 2023) for all three benchmark classification tasks. Adam optimizer (Kingma, 2014) is adopted with an appropriate learning rate and weight decay to achieve the best performance on the validation split. Detailed hyperparameter settings and experimental environments are provided in Appendix B.2 and Appendix B.3, respectively. For evaluation, we follow prior studies in choosing task-specific metrics: accuracy for node classification (Feng et al., 2019; Chien et al., 2022; Wang et al., 2023a); AUROC (area under the ROC curve) and AP (average precision) for hyperedge prediction (Hwang et al., 2022; Ko et al., 2025; Yu et al., 2025; Tang et al., 2025); and both accuracy and Macro-F1 score for hypergraph classification (Feng et al., 2024). Higher values of these metrics indicate better predictive performance. In addition, to assess algorithmic fairness, we adopt two commonly used group fairness metrics: demographic parity ($\Delta_{DP}$) (Dwork et al., 2012) and equalized odds ($\Delta_{EO}$) (Hardt et al., 2016), with detailed definitions provided in Appendix B.5. For each method and dataset, we record the mean results and the standard deviation across 5 runs.

### B.2 HYPERPARAMETER SETTING

We carefully tune hyperparameters to ensure a rigorous and unbiased evaluation of the integrated HNN methods. For algorithms without explicit hyperparameter guidelines in their original papers or source code, we perform a grid search with a reasonable budget across all datasets to identify optimal configurations. The search spaces are provided in Table A4. For detailed interpretations, please refer to the corresponding papers, and the complete hyperparameter configurations are available in our publicly released GitHub repository.

### B.3 EXPERIMENTAL ENVIRONMENT

All the experiments are conducted with the following computational resources and configurations:

- Operating system: Ubuntu 24.04 LTS.
- CPU information: Intel(R) Xeon(R) Silver 4208 CPU @ 2.10GHz with 128G Memory.
- GPU information: Quadro RTX 6000 with 24GB of Memory.

Table A4: Hyperparameter search space of different methods.

| Method | Hyperparameter | Search Space |
|---|---|---|
| General Settings | Epochs | 100, 200, 300, 400, 500, 800, 1000 |
| | Learning Rate | 0.1, 0.01, 0.001, 0.0001 |
| | Layers | 1, 2, 3, 4 |
| | Dropout Rate | 0, 0.1, 0.2, 0.3, 0.4, 0.5, 0.6, 0.7, 0.8 |
| | Weight Decay | 0, 0.0005 |
| | Hidden Units | 64, 128, 256, 512, 1024 |
| | Activation | LeakyReLU, ReLU, PReLU, Sigmoid, Softmax |
| | Hyperedge Pooling | max, mean, max-min |
| | Hypergraph Pooling | max, mean |
| HCHA | heads | 1, 2, 4, 8, 16 |
| HyperND | HyperND_ord | 1, 2, 3, 5, 10 |
| | HyperND_tol | 0.001, 0.0001, 0.00001, 0.000001 |
| | HyperND_steps | 50, 100, 150, 200 |
| | alpha | 0, 0.1, 0.2, 0.3, 0.4, 0.5, 0.6, 0.7, 0.8, 0.9 |
| HJRL | $\lambda_0$ | 0.001, 0.01, 0.1, 1, 10 |
| PhenomNN | $\lambda_0$ | 0, 0.1, 1, 10, 20, 50, 80, 100 |
| | $\lambda_1$ | 0, 0.1, 1, 10, 20, 50, 80, 100 |
| | prop_steps | 2, 4, 8, 16, 32, 64, 128 |
| | alpha | 0, 0.1, 0.2, 0.3, 0.4, 0.5, 0.6, 0.7, 0.8, 0.9 |
| SheafHyperGNN | init_hedge | rand, avg |
| | sheaf_pred_block | MLP_var1, MLP_var2, MLP_var3, cp_decomp |
| | sheaf_transformer_head | 1, 2, 4, 8, 16 |
| | stalk_dim | 1, 2, 4, 8 |
| TF-HNN | mlp_hidden_size | 64, 128, 256, 512, 1024 |
| | # layers of classifier | 1, 2, 3, 4 |
| HNHN | alpha | -3.0, -2.5, -2.0, -1.5, -1.0, -0.5, 0.0, 0.5 |
| | beta | -2.5, -2.0, -1.5, -1.0, -0.5, 0.0, 0.5, 1.0 |
| UniGNN | alpha | 0, 0.1, 0.2, 0.3, 0.4, 0.5, 0.8, 0.9 |
| | beta | 0, 0.1, 0.2, 0.3, 0.4, 0.5, 0.8, 0.9 |
| AllSetTransformer | attention_heads | 1, 2, 4, 8, 16 |
| ED-HNN | alpha | 0, 0.1, 0.2, 0.3, 0.4, 0.5, 0.6, 0.7, 0.8, 0.9 |
| | # layers of $\hat{\phi}$ | 0, 1, 2, 3 |
| | # layers of $\hat{\rho}$ | 0, 1, 2, 3 |
| | # layers of $\hat{\varphi}$ | 0, 1, 2, 3 |
| DPHGNN | attention_heads | 1, 2, 4, 8, 16 |
| | # layers of TAA module | 1, 2, 3, 4 |
| | # layers of SIB module | 1, 2 |
| | # layers of DFF module | 1, 2 |
| HyperGT | attention_heads | 1, 2, 4, 8, 16 |
| EHNN | ehnn_qk_channels | 64, 128, 256 |
| | ehnn_n_heads | 1, 2, 4, 8, 16 |
| | ehnn_pe_dim | 64, 128 |
| | ehnn_inner_channel | 64, 128, 256 |
| | ehnn_hidden_channel | 64, 128, 256 |
| T-HyperGNN | M: maximum cardinality | 1, 2, 3, 4, 5 |
| | combine | concat, sum |

- Software: CUDA 12.1, Python 3.9.21, Pytorch (Paszke et al., 2019) 2.2.2, Pytorch Geometric (Fey & Lenssen, 2019) 2.6.1.

### B.4 ROBUSTNESS EVALUATION SETTINGS

In our robustness study, we simulate data perturbation scenarios from three perspectives: structure, feature, and supervision signal. Each perturbation setting is repeated 5 times with different random seeds to account for randomness, and we report the average results. Our experiments primarily focus on the node classification task. The detailed experimental setups are as follows.

**Structure-level Robustness Evaluation Setting.** To analyze structure-level robustness, following (Cai et al., 2022), we randomly remove or add a proportion of node–hyperedge connections (i.e., hyperlinks) in the original hypergraph and then train and evaluate HNN algorithms on the perturbed structures. The modification ratio ranges from 0 to 0.9 to simulate varying levels of noise intensity.

**Feature-level Robustness Evaluation Setting.** To study feature-level robustness, we simulate two realistic types of feature perturbations: feature noise and feature sparsity. For feature noise, following (Wu et al., 2020), we add independent Gaussian noise to each feature dimension of all nodes with gradually increasing amplitude. Specifically, we use the mean of the maximum feature value of each node as the reference amplitude $r$, and add Gaussian noise $\lambda \cdot r \cdot \epsilon$ to each feature dimension, where $\epsilon \sim \mathcal{N}(0, 1)$ and $\lambda$ denotes the feature noise ratio. We evaluate model performance as $\lambda$ varies from 0 to 0.9 with a step size of 0.1. For feature sparsity, following (Li et al., 2023), we randomly mask a certain proportion of node features by filling them with zeros, with the sparsity ratio ranging from 0 to 0.9 at an interval of 0.1.

**Supervision-level Robustness Evaluation Setting.** We study supervision-level robustness by simulating realistic noise and sparsity scenarios. For label noise, following (Dai et al., 2021), a certain proportion of training samples are randomly assigned incorrect labels by uniformly flipping them to one of the other classes. The noise ratio varies from 0 to 0.2 in increments of 0.05. Sparsity is introduced by reducing the ratio of training nodes, with the sparsity rate ranging from 0 to 0.8 with a step size of 0.2.

### B.5 FAIRNESS EVALUATION METRICS

For fairness evaluation, we adopt two widely used group fairness metrics: demographic parity (DP) (Dwork et al., 2012), and equalized odds (EO) (Hardt et al., 2016). We focus on a binary classification task, with target label $y \in \{0, 1\}$ and binary sensitive attribute $s \in \{0, 1\}$.

**Demographic Parity.** If the predicted result $\hat{y}$ is independent of sensitive attributes $s$, i.e., $\hat{y} \perp s$, then we can consider demographic parity is achieved. Formally, this criterion can be expressed as:

$$P(\hat{y} = 1 \mid s = 0) = P(\hat{y} = 1 \mid s = 1). \tag{1}$$

If a model satisfies demographic parity, the acceptance rate of different protected groups is the same. The deviation measure $\Delta_{DP}$ in the quantitative evaluation is given by:

$$\Delta_{DP} = |P(\hat{y} = 1 \mid s = 0) - P(\hat{y} = 1 \mid s = 1)|, \tag{2}$$

where a smaller value indicates a fairer prediction distribution across groups.

**Equalized Odds.** If the predicted outcome $\hat{y}$ and the sensitive attribute $s$ are conditionally independent given the ground-truth label $y$, i.e., $\hat{y} \perp s \mid y$, then we consider equalized odds is achieved. The formula for this criterion is as follows:

$$P(\hat{y} = 1 \mid s = 1, y = 1) = P(\hat{y} = 1 \mid s = 0, y = 1). \tag{3}$$

If a model achieves equalized odds, the True Positive Rate (TPR) and False Positive Rate (FPR) are equal across different protected groups. The deviation measure $\Delta_{EO}$ is calculated as:

$$\Delta_{EO} = |P(\hat{y} = 1 \mid s = 1, y = 1) - P(\hat{y} = 1 \mid s = 0, y = 1)|, \tag{4}$$

where a smaller value reflects more equitable predictive behavior across sensitive groups under the same ground-truth condition.

### B.6 Discussion on Robustness and Fairness Evaluation

In this section, for the newly introduced robustness and fairness metrics, we discuss how an ideal HNN model is expected to behave during evaluation.

#### B.6.1 Discussion on Robustness Metrics

**Structure Robustness.** (1) In homophilous settings, meaningful higher-order relations benefit classification. Under drop perturbations, a desirable HNN should maintain accuracy that is no lower than that of structure-agnostic baselines (e.g., MLPs), and ideally remain as stable as possible. This indicates that when higher-order structure exists, the model is indeed able to effectively leverage it. Under addition perturbations, which introduce noisy or spurious links, an ideal HNN is expected to identify and down-weight these noisy edges during message passing. Consequently, the model should also maintain stable performance and stay close to the clean-hypergraph accuracy, demonstrating resilience to the adverse effects of structural noise. (2) In heterophilous settings, many higher-order connections are not helpful and may even be harmful. In this case, as the perturbation ratio increases, a robust HNN is expected to show a performance trend that remains stable or even improves. Such a trend indicates that disrupting harmful heterophilous links enables the model to better capture the remaining homophilous patterns, reflecting stronger robustness to misleading structural signals.

**Feature Robustness.** For feature robustness evaluation, an ideal HNN is one whose predictive performance degrades slowly as feature noise increases or feature sparsity becomes more severe. Under our benchmark setting, we expect a good HNN to maintain an average performance clearly above the baseline obtained when all features are replaced with random noise, indicating that the model can effectively exploit meaningful feature signals. Likewise, as the feature sparsity ratio increases, the model's performance should decline gradually while remaining above the extreme case where only a single feature dimension is preserved and, within this feasible range, stay as close as possible to the clean-hypergraph performance. Such behavior reflects the model's ability to utilize informative features even under highly degraded or partially missing feature conditions.

**Label Robustness.** For label robustness evaluation, we regard an ideal HNN as one whose predictive performance remains insensitive to different levels of label noise and label sparsity. Under our benchmark setting, a strong HNN should retain test accuracy close to its clean-data performance, showing either minimal degradation or no noticeable drop as the proportion of noisy labels increases or as the fraction of labeled training nodes decreases.

#### B.6.2 Discussion on Fairness Metrics

For fairness evaluation, an ideal HNN maintains strong predictive performance while exhibiting no algorithmic bias across different sensitive demographic groups. Specifically, under our benchmark setting, a good HNN should achieve high node classification accuracy while simultaneously attaining low values on the fairness metrics demographic parity ($\Delta_{DP}$) and equalized odds ($\Delta_{EO}$).

### B.7 Discussion on Memory Mitigation Strategies

In our DHG-Bench, our primary mitigation strategy for handling memory-intensive settings is the unified support for sparse-matrix storage and training. Sparse operations are broadly compatible with all HNN models and effectively reduce memory overhead without affecting training dynamics, making them a practical and reliable choice. Below, we detail this strategy and explain why certain other techniques were not adopted.

**Support for Sparse Matrix.** DHG-Bench implements full sparse support for all HNNs, including sparse incidence matrices and sparse matrix computations during message passing. Representing the incidence matrix in a sparse format substantially reduces memory consumption, particularly for large-scale datasets. Sparse tensor operations also eliminate the need to materialize dense intermediate matrices during aggregation, which lowers peak memory usage in both the forward and backward passes. This design allows DHG-Bench to scale to larger hypergraphs than would be feasible with dense representations and serves as our main approach to preventing the OOM issue.

**Why Mini-batching is not Used.** Following the standard practice in most related HNN studies, DHG-Bench employs full-batch training for all models. Hypergraphs differ fundamentally from

Table A5: Additional node classification results on NTU2012, ModelNet40, and Ratings.

| Method | NTU2012 | ModelNet40 | Ratings |
|---|---|---|---|
| MLP | $88.59_{\pm 1.27}$ | $96.88_{\pm 0.23}$ | $28.47_{\pm 0.76}$ |
| CEGCN | $84.93_{\pm 1.12}$ | $92.34_{\pm 0.24}$ | $26.65_{\pm 1.61}$ |
| CEGAT | $84.14_{\pm 1.77}$ | $92.02_{\pm 0.26}$ | $28.23_{\pm 0.50}$ |
| HGNN | $90.13_{\pm 0.89}$ | $97.43_{\pm 0.20}$ | $28.05_{\pm 0.28}$ |
| HyperGCN | $75.78_{\pm 4.82}$ | $91.15_{\pm 3.88}$ | $27.34_{\pm 0.72}$ |
| HCHA | $90.53_{\pm 1.00}$ | $97.68_{\pm 0.16}$ | $28.33_{\pm 0.34}$ |
| LEGCN | $89.82_{\pm 0.91}$ | $96.82_{\pm 0.24}$ | $28.21_{\pm 0.50}$ |
| HyperND | $88.98_{\pm 1.56}$ | $97.18_{\pm 0.58}$ | $28.32_{\pm 0.38}$ |
| PhenomNN | $88.78_{\pm 0.67}$ | $98.28_{\pm 0.18}$ | $28.49_{\pm 0.41}$ |
| SheafHyperGNN | $90.81_{\pm 0.58}$ | $98.30_{\pm 0.19}$ | $28.35_{\pm 0.57}$ |
| HJRL | $88.15_{\pm 1.18}$ | $96.33_{\pm 0.30}$ | $26.90_{\pm 0.55}$ |
| DPHGNN | $84.77_{\pm 1.06}$ | $97.19_{\pm 0.17}$ | $28.57_{\pm 1.07}$ |
| TF-HNN | $\mathbf{91.69_{\pm 0.75}}$ | $98.38_{\pm 0.11}$ | $28.56_{\pm 0.68}$ |
| HNHN | $87.27_{\pm 1.53}$ | $97.30_{\pm 0.27}$ | $27.29_{\pm 0.70}$ |
| UniGNN | $89.86_{\pm 0.44}$ | $98.42_{\pm 0.08}$ | $28.39_{\pm 0.64}$ |
| AllSetTransformer | $90.17_{\pm 1.03}$ | $98.07_{\pm 0.21}$ | $27.32_{\pm 1.11}$ |
| ED-HNN | $\underline{91.45_{\pm 0.70}}$ | $\mathbf{98.51_{\pm 0.15}}$ | $28.38_{\pm 0.31}$ |
| HyperGT | $86.00_{\pm 2.05}$ | $96.83_{\pm 0.17}$ | $26.58_{\pm 0.33}$ |
| EHNN | $87.99_{\pm 0.39}$ | $97.97_{\pm 0.17}$ | $\mathbf{28.95_{\pm 0.81}}$ |
| T-HyperGNN | $89.15_{\pm 1.09}$ | $97.76_{\pm 0.34}$ | $24.63_{\pm 1.22}$ |

graphs because hyperedges connect multiple nodes simultaneously. However, there is currently no widely adopted, hypergraph-specific mini-batch sampling strategy that preserves hyperedge integrity or provides unbiased training signals. Existing sampling methods designed for graphs do not directly transfer to hypergraphs, as they often break hyperedge structures or distort higher-order relationships. DHG-Bench therefore follows the full-batch protocol to ensure comparability with prior works. Developing principled mini-batch sampling strategies for hypergraphs is an important direction, and we plan to explore this in future extensions of DHG-Bench.

**Why Mixed-Precision is not Used.** Mixed-precision training can reduce memory usage in some deep learning models. However, many HNNs rely on sparse operations and irregular message-passing kernels, and while PyTorch technically allows FP16 sparse tensors, most sparse operators either lack full FP16 support or exhibit numerical instability in half-precision settings. To keep the evaluation fair and consistent across all models, we choose not to include the mixed precision strategy.

## C  SUPPLEMENTARY EXPERIMENTAL RESULTS

### C.1  EXPERIMENTAL RESULTS ON EFFECTIVENESS EVALUATION

Table A5 shows the node classification results of all HNN algorithms on three datasets: NTU2012, ModelNet, and Ratings.

Table A6, A7 reports the full result of hyperedge prediction and hypergraph classification, respectively. Tensor-based methods are not considered in the hypergraph classification task, as they lack flexibility in supporting multi-graph training.

### C.2  EXPERIMENTAL RESULTS ON ROBUSTNESS EVALUATION

Figures A2, A3, and A4 show the robustness evaluation results at the structure, feature, and supervision levels on the Pubmed and Pokec datasets, respectively.

### C.3  EXPERIMENTAL RESULTS ON FAIRNESS EVALUATION

Table A8 presents the full experimental results of fairness evaluation in terms of three metrics: accuracy (Acc), demographic parity ($\Delta_{DP}$), and equalized odds ($\Delta_{EO}$).

Table A6: Evaluation results of hyperedge prediction.

| Method | Cora | | PubMed | | Cora-CA | | DBLP-CA | | Actor | | Pokec | |
|---|---|---|---|---|---|---|---|---|---|---|---|---|
| | AUROC | AP | AUROC | AP | AUROC | AP | AUROC | AP | AUROC | AP | AUROC | AP |
| MLP | 68.01±1.23 | 71.32±1.13 | 66.00±0.44 | 69.21±0.61 | 71.15±1.73 | 72.80±1.27 | 69.19±0.19 | 70.66±0.36 | 54.75±2.29 | 53.63±1.56 | 69.69±2.56 | 69.07±3.03 |
| CEGCN | 66.10±2.43 | 65.50±2.85 | 60.14±3.97 | 60.25±3.60 | 67.27±3.78 | 71.23±2.69 | 64.06±1.11 | 65.07±1.69 | 50.02±0.01 | 50.05±0.02 | 73.03±2.76 | 70.08±2.95 |
| CEGAT | 72.48±0.52 | 71.02±0.64 | 62.20±6.25 | 61.63±5.99 | 69.81±1.13 | 70.38±1.33 | 66.50±8.80 | 65.29±8.21 | 56.34±5.33 | 56.36±5.07 | 81.01±0.43 | 79.61±1.60 |
| HGNN | 73.70±1.19 | 71.73±1.57 | 66.08±9.84 | 63.67±9.02 | 89.16±1.11 | 89.85±0.82 | 75.44±3.01 | 73.96±4.91 | **72.42±1.96** | **68.79±1.83** | 86.09±0.92 | 84.32±0.95 |
| HyperGCN | 77.34±1.30 | 77.15±0.33 | 66.46±8.87 | 64.84±7.98 | **92.73±0.95** | **93.42±0.89** | **89.46±0.18** | **91.39±0.34** | 55.01±8.76 | 56.29±7.44 | **91.45±0.70** | 90.76±0.70 |
| HCHA | 73.57±1.08 | 72.24±1.80 | 63.35±1.61 | 63.13±1.47 | 85.85±3.27 | 84.77±5.66 | 73.30±3.72 | 72.09±3.07 | 69.86±0.95 | 66.72±0.74 | 88.81±0.28 | 88.25±0.41 |
| LEGCN | 67.16±2.85 | 68.76±4.89 | 56.39±3.28 | 54.33±2.41 | 74.29±0.59 | 75.95±0.62 | 50.70±1.40 | 50.47±0.94 | 48.25±3.00 | 49.76±1.29 | 74.94±1.44 | 73.89±1.04 |
| HyperND | 69.10±1.28 | 72.71±1.48 | 72.12±0.78 | 73.53±0.63 | 84.01±0.61 | 84.98±1.07 | 78.63±0.71 | 79.42±0.94 | 53.12±2.56 | 52.64±2.33 | 75.77±1.56 | 73.51±1.62 |
| PhenomNN | 75.71±0.91 | 75.22±1.42 | 74.29±0.85 | 72.93±1.27 | 80.27±1.62 | 79.59±1.11 | 75.86±0.86 | 75.54±0.88 | 56.65±3.04 | 55.75±2.87 | 70.83±2.52 | 70.17±2.36 |
| SheafHyperGNN | 70.53±5.28 | 70.93±4.04 | 68.26±1.92 | 68.07±1.18 | 79.21±4.53 | 75.42±6.73 | 76.30±1.91 | 75.41±1.76 | 59.83±6.77 | 59.84±5.73 | 83.44±2.49 | 85.11±1.80 |
| HJRL | 58.48±2.52 | 61.02±2.60 | 59.28±0.84 | 58.63±1.50 | 82.41±1.90 | 85.67±1.11 | OOM | OOM | 48.26±0.77 | 50.00±0.31 | 84.88±3.30 | 86.18±2.61 |
| DPHGNN | 66.48±5.82 | 67.23±5.11 | 60.37±7.77 | 59.86±7.70 | 82.89±2.28 | 83.78±2.50 | OOM | OOM | 42.44±5.81 | 46.60±3.03 | 73.35±4.59 | 73.28±3.74 |
| TF-HNN | 76.94±0.86 | 76.57±0.71 | 73.75±0.73 | 75.54±0.72 | 74.97±1.85 | 71.13±1.65 | 75.70±2.77 | 74.69±2.26 | 54.03±1.71 | 54.06±1.57 | 68.00±0.97 | 67.41±1.20 |
| HNHN | 70.13±1.67 | 68.84±1.09 | 55.67±0.39 | 53.52±0.31 | 84.33±1.40 | 83.49±1.00 | 82.85±0.70 | 82.13±0.58 | 69.89±0.98 | 66.45±0.74 | 82.25±1.34 | 81.72±1.53 |
| UniGNN | 73.51±0.87 | 75.23±1.51 | 74.20±0.82 | 71.76±1.16 | 80.59±0.98 | 82.37±1.11 | 81.08±0.79 | 79.39±0.46 | 50.24±1.26 | 50.01±0.56 | 85.64±1.20 | 84.36±1.48 |
| AllSetTransformer | 72.55±2.95 | 74.86±1.85 | 71.09±2.99 | 73.15±2.49 | 76.13±7.70 | 75.02±8.68 | 75.12±4.14 | 77.12±4.22 | 55.84±5.99 | 58.73±4.39 | 83.65±4.34 | 83.36±4.72 |
| ED-HNN | 67.24±1.91 | 69.89±2.24 | 70.09±0.43 | 72.61±0.48 | 74.58±1.37 | 72.94±1.32 | 81.86±0.67 | 84.75±0.50 | 51.74±2.79 | 52.27±2.54 | 85.27±1.48 | 84.95±1.43 |
| HyperGT | 60.68±4.46 | 63.02±4.00 | 64.38±0.58 | 67.79±0.59 | 65.99±2.48 | 69.66±2.20 | 72.90±0.24 | 72.90±0.17 | 65.18±1.60 | 63.24±0.53 | 81.37±5.38 | 82.73±5.83 |
| EHNN | **78.99±0.99** | **79.54±0.93** | **76.50±0.62** | **75.94±0.70** | 77.83±3.01 | 78.29±3.72 | 87.96±0.98 | 89.00±0.64 | 65.69±0.46 | 65.37±0.35 | 88.63±1.58 | **91.31±0.88** |
| T-HyperGNN | 58.91±1.23 | 62.17±1.58 | 58.35±4.43 | 55.81±3.71 | 66.87±0.88 | 69.65±0.53 | 67.17±5.79 | 68.45±3.85 | 49.16±0.22 | 50.20±0.41 | 65.21±1.21 | 66.90±1.56 |

Table A7: Evaluation results of hypergraph classification. Acc and F1_ma denote the accuracy and Macro-F1, respectively. Tensor-based methods are omitted as they cannot be applied to this task.

| Method | RHG-10 | | RHG-3 | | IMDB-Dir-Form | | IMDB-Dir-Genre | | Steam-Player | | Twitter-Friend | |
|---|---|---|---|---|---|---|---|---|---|---|---|---|
| | Acc | F1_ma | Acc | F1_ma | Acc | F1_ma | Acc | F1_ma | Acc | F1_ma | Acc | F1_ma |
| MLP | 91.70±1.02 | 91.43±1.09 | 95.73±1.86 | 95.72±1.84 | 63.62±1.69 | 56.98±3.93 | 75.12±0.70 | 71.10±0.74 | 52.34±0.55 | 51.60±0.68 | 57.25±1.81 | 52.88±4.57 |
| CEGCN | 91.50±1.55 | 90.48±1.42 | 98.63±0.73 | 98.65±0.77 | 62.66±1.82 | 55.31±3.58 | 75.06±0.76 | 68.98±1.67 | 48.16±3.87 | 47.03±3.79 | 54.66±5.66 | 42.16±2.71 |
| CEGAT | 88.70±1.71 | 88.43±1.72 | 98.80±0.61 | 98.83±0.59 | 63.51±1.54 | 56.97±4.83 | 74.12±2.69 | 68.61±4.73 | 49.51±4.71 | 46.85±4.93 | 57.32±2.59 | 38.22±2.54 |
| HGNN | 94.60±1.66 | 94.47±1.84 | 98.93±0.68 | 98.97±0.65 | 63.72±0.62 | **57.92±2.24** | 76.76±2.66 | 72.02±4.37 | 51.65±2.51 | 50.91±2.92 | 56.95±4.17 | 46.81±4.27 |
| HyperGCN | 85.50±1.10 | 95.42±1.09 | 99.47±0.50 | 99.48±0.49 | 62.87±0.40 | 57.20±2.46 | 77.53±0.99 | 72.97±1.08 | 51.17±3.32 | 50.48±3.12 | 56.95±4.17 | 50.12±5.88 |
| HCHA | 96.60±1.02 | 96.48±1.09 | 99.33±0.42 | 99.37±0.38 | 61.60±2.16 | 55.37±2.17 | **78.12±1.96** | 73.20±3.00 | 51.42±2.30 | 51.77±2.52 | 58.17±2.34 | 49.57±6.84 |
| LEGCN | 92.40±1.16 | 92.06±1.19 | 96.80±0.98 | 96.78±0.29 | 61.81±1.32 | 56.05±3.75 | 76.38±1.68 | 72.03±1.54 | 53.11±1.58 | 52.70±1.87 | 56.64±3.72 | **53.38±4.92** |
| HyperND | 91.00±0.95 | 90.74±1.04 | 92.80±1.95 | 92.75±1.90 | 60.74±3.25 | 55.02±4.95 | 75.65±0.51 | 71.37±1.10 | 53.88±2.15 | 49.71±2.05 | 55.27±3.79 | 43.61±6.51 |
| PhenomNN | 91.10±0.73 | 90.77±0.77 | 93.47±1.90 | 93.45±1.90 | 61.28±1.97 | 53.71±3.13 | 74.59±0.61 | 70.15±0.88 | 51.65±3.06 | 48.94±4.55 | 57.40±3.84 | 48.26±4.66 |
| SheafHyperGNN | 96.00±1.38 | 95.96±1.32 | 99.73±0.33 | 99.74±0.30 | 62.34±2.06 | 56.47±3.49 | 77.00±1.14 | 72.78±1.17 | 53.11±2.39 | 52.56±2.74 | 56.49±2.51 | 51.43±4.42 |
| HJRL | 96.10±0.80 | 95.98±0.85 | 99.60±0.53 | 99.57±0.52 | 63.09±2.83 | 56.54±3.62 | 77.82±1.47 | **73.73±1.92** | 51.84±3.52 | 51.13±3.29 | 57.10±2.79 | 44.19±7.19 |
| DPHGNN | 96.80±0.68 | 96.71±0.71 | 99.49±0.65 | 99.61±0.64 | **64.04±2.70** | 57.41±3.96 | 76.18±1.30 | 71.59±1.82 | 51.36±1.72 | 49.03±3.63 | 59.24±2.88 | 46.12±8.49 |
| TF-HNN | 95.90±0.80 | 95.88±0.78 | 98.80±0.65 | 98.84±0.61 | 62.34±1.76 | 55.32±3.81 | 76.41±1.31 | 71.89±1.45 | **54.85±1.82** | **52.72±2.54** | 56.18±3.53 | 44.17±8.95 |
| HNHN | 94.00±1.90 | 94.08±1.88 | **99.92±0.02** | **99.95±0.02** | 62.34±2.98 | 55.24±3.88 | 73.65±1.47 | 69.68±1.18 | 52.82±1.61 | 52.68±1.69 | 58.47±4.65 | 39.40±3.14 |
| UniGNN | 95.50±1.38 | 95.40±1.44 | 98.80±0.27 | 98.83±0.27 | 61.06±2.88 | 55.75±4.01 | 77.12±0.88 | 72.93±1.43 | 51.46±2.48 | 48.85±2.59 | 55.88±4.14 | 46.48±4.90 |
| AllSetTransformer | **97.30±0.98** | **97.26±1.04** | 98.80±0.27 | 98.81±0.26 | 62.23±1.01 | 56.26±2.93 | 76.47±1.38 | 72.26±1.12 | 53.01±2.77 | 48.21±7.02 | **60.15±1.75** | 51.52±7.00 |
| ED-HNN | 96.50±0.77 | 96.41±0.78 | 99.07±0.53 | 99.10±0.51 | 62.13±2.36 | 57.00±4.71 | 77.12±1.11 | 72.87±0.44 | 52.82±2.65 | 48.73±2.36 | 57.40±2.66 | 42.57±5.09 |
| HyperGT | 91.60±1.42 | 91.29±1.53 | 96.27±1.93 | 96.28±1.88 | 61.49±4.32 | 55.14±6.02 | 73.82±1.27 | 69.36±1.45 | 54.47±1.33 | 51.55±1.08 | 54.35±2.72 | 47.49±5.11 |

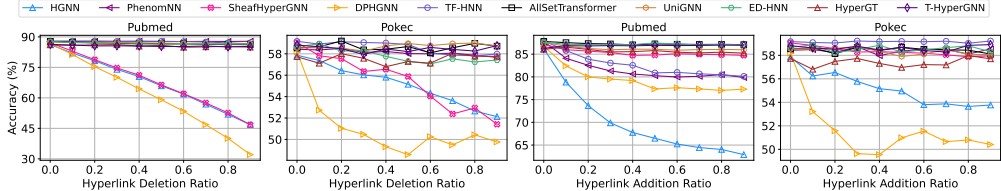

Figure A2: Structure robustness analysis on Pubmed and Pokec.

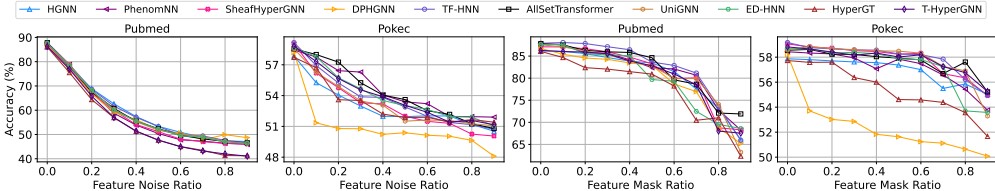

Figure A3: Feature robustness analysis on Pubmed and Pokec.

## C.4 NODE CLASSIFICATION IN LABEL-SCARCE SCENARIOS

In this section, we analyze the HNNs in more label-scarce scenarios to provide additional insights into the effectiveness of the HNN algorithms, particularly in understanding their applicability in real-world settings where labeled data is limited.

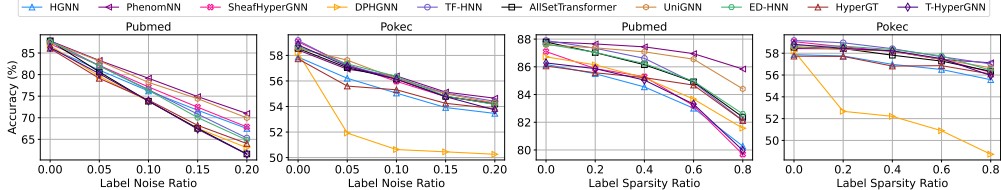

Figure A4: Supervision robustness analysis on Pubmed and Pokec.

Table A8: Fairness Evaluation.

| Method | German | | | Bail | | | Credit | | |
|---|---|---|---|---|---|---|---|---|---|
| | Acc ↑ | $\Delta_{DP}$ ↓ | $\Delta_{EO}$ ↓ | Acc ↑ | $\Delta_{DP}$ ↓ | $\Delta_{EO}$ ↓ | Acc ↑ | $\Delta_{DP}$ ↓ | $\Delta_{EO}$ ↓ |
| MLP | 67.68±3.46 | **1.78±1.30** | 2.59±0.99 | 89.40±1.76 | 6.16±0.93 | 1.79±0.84 | 79.69±0.85 | **3.43±0.83** | **2.07±0.43** |
| CEGCN | 69.60±2.78 | 6.19±4.59 | 6.46±5.48 | OOM | OOM | OOM | OOM | OOM | OOM |
| CEGAT | 69.12±2.44 | 9.00±4.77 | 8.52±3.82 | OOM | OOM | OOM | OOM | OOM | OOM |
| HGNN | 69.76±2.50 | 9.59±3.51 | 6.90±3.82 | 91.02±0.54 | 7.83±0.80 | 2.60±1.05 | 80.21±0.41 | 5.04±2.07 | 3.46±1.05 |
| HyperGCN | 70.40±3.23 | 6.39±2.87 | 3.57±1.61 | 94.72±0.79 | 7.90±0.95 | 1.23±0.52 | 80.42±0.34 | 5.38±2.61 | 3.89±1.60 |
| HCHA | 70.56±2.51 | 9.37±3.74 | 6.72±3.05 | 91.52±0.92 | 7.62±0.95 | 1.40±0.80 | 80.08±0.43 | 3.58±1.87 | 2.47±0.78 |
| LEGCN | 70.88±3.22 | 7.53±2.54 | 3.25±2.00 | 95.02±0.40 | 7.87±0.62 | 1.28±0.52 | **80.48±0.37** | 4.31±2.24 | 3.15±1.12 |
| HyperND | 71.04±2.61 | 7.37±4.70 | 3.67±3.10 | 89.75±2.41 | 7.92±1.52 | 3.19±2.22 | 80.02±0.49 | 4.14±2.22 | 2.50±0.72 |
| PhenomNN | 70.96±2.85 | 3.54±3.07 | 1.60±1.94 | 91.71±1.13 | 10.83±1.64 | 1.94±0.40 | OOM | OOM | OOM |
| SheafHyperGNN | 70.64±3.29 | 8.14±3.25 | 5.23±2.05 | OOM | OOM | OOM | OOM | OOM | OOM |
| HJRL | 69.92±3.46 | 3.52±2.70 | 3.05±1.82 | OOM | OOM | OOM | OOM | OOM | OOM |
| DPHGNN | 70.24±3.25 | 2.25±0.49 | **1.38±0.77** | 93.41±0.93 | 8.07±1.20 | 2.05±1.22 | OOM | OOM | OOM |
| TF-HNN | 70.48±3.14 | 5.27±3.09 | 4.19±2.23 | 95.33±0.25 | 7.96±0.65 | **1.03±0.67** | 80.46±0.36 | 4.93±2.44 | 3.43±1.43 |
| HNHN | 69.52±3.62 | 4.01±2.76 | 1.59±1.60 | 90.76±1.30 | **6.03±1.43** | 3.04±1.34 | 78.00±0.23 | 5.70±3.11 | 4.67±3.10 |
| UniGNN | 71.07±2.70 | 5.08±3.03 | 2.80±1.32 | 91.30±1.47 | 9.42±1.68 | 3.94±2.18 | 80.44±0.37 | 3.90±2.40 | 2.85±1.33 |
| AllSetTransformer | 70.48±3.11 | 4.47±3.39 | 3.50±3.38 | **96.26±1.83** | 8.36±0.85 | 1.95±1.10 | 80.40±0.44 | 4.46±2.96 | 3.44±1.60 |
| ED-HNN | 70.16±3.15 | 4.06±3.05 | 4.07±2.75 | 94.26±0.77 | 8.05±0.64 | 1.51±0.26 | OOM | OOM | OOM |
| HyperGT | 68.88±2.01 | 5.05±2.88 | 4.36±2.59 | 94.33±0.62 | 7.68±1.13 | 1.64±1.37 | 79.83±0.39 | 4.17±2.50 | 2.69±1.97 |
| EHNN | 70.40±3.07 | 2.87±5.73 | 2.34±4.69 | 93.62±1.75 | 9.29±1.60 | 2.88±1.23 | 80.34±0.47 | 4.51±2.77 | 3.13±1.75 |
| T-HyperGNN | **71.20±1.82** | 8.99±6.52 | 6.80±5.02 | OOM | OOM | OOM | OOM | OOM | OOM |

**Experiment Settings.** We first split the node labels into 20%/20%/60% for the train/validation/test sets. The validation and test sets are then kept fixed, and different levels of label scarcity are simulated by masking a portion of the training labels. Specifically, we adjust the masking ratio so that the visible training labels constitute 20%, 15%, 10%, 5%, and 1% of all nodes. This design allows us to systematically examine how HNNs behave as labeled data becomes increasingly limited. We evaluate 8 representative HNN algorithms spanning three major categories (spectral-based, spatial-based, and tensor-based) on the Cora and Actor datasets, and report model performance in terms of accuracy.

Table A9: Label-scarce node classification on Cora.

| Method | 20% | 15% | 10% | 5% | 1% |
|---|---|---|---|---|---|
| HGNN | 74.84 | 73.24 | 70.09 | 64.75 | 42.30 |
| PhenomNN | 75.35 | 74.07 | 71.96 | 67.55 | 44.96 |
| SheafHyperGCN | 76.06 | 74.66 | 71.29 | 66.37 | 43.67 |
| TF-HNN | 76.31 | 75.07 | 71.77 | 64.48 | 39.29 |
| UniGNN | 76.08 | 74.04 | 70.89 | 64.85 | 43.18 |
| AllSetTransformer | 73.48 | 72.33 | 68.46 | 61.70 | 40.44 |
| ED-HNN | 74.20 | 72.41 | 69.93 | 63.65 | 42.79 |
| T-HyperGNN | 69.02 | 66.99 | 62.50 | 52.89 | 36.60 |

**Results and Analysis.** From Tables A9 and A10, we derive the following key observations: (1) As label scarcity increases, all HNN models exhibit a clear degradation in performance, with the decline becoming more significant under extremely low-label settings; notably, all methods experience substantial drops when the labeled ratio decreases from 5% to 1%. (2) Across both datasets, PhenomNN consistently shows the strongest robustness under highly label-scarce conditions (1% and 5%). In contrast, TF-HNN, although it achieves SOTA performance in label-abundant scenarios (see Table 1 in the original manuscript), suffers a severe accuracy collapse when supervision is limited and ranks as the second-worst method on Cora at the 1% label ratio. (3) The performance degradation on the homophilous Cora dataset is more pronounced than on the heterophilous Actor

Table A10: Label-scarce node classification on Actor.

| Method | 20% | 15% | 10% | 5% | 1% |
|---|---|---|---|---|---|
| HGNN | 77.90 | 77.79 | 77.52 | 76.39 | 70.12 |
| PhenomNN | 82.99 | 82.89 | 82.56 | 81.77 | 76.89 |
| SheafHyperGCN | 84.16 | 83.67 | 82.95 | 80.48 | 72.27 |
| TF-HNN | 85.34 | 84.74 | 83.95 | 81.74 | 74.88 |
| UniGNN | 82.85 | 82.65 | 82.64 | 81.69 | 76.34 |
| AllSetTransformer | 84.06 | 83.46 | 82.25 | 79.55 | 75.47 |
| ED-HNN | 84.74 | 84.30 | 83.38 | 81.14 | 75.53 |
| T-HyperGNN | 84.87 | 84.39 | 83.48 | 81.29 | 73.63 |

dataset. This may be because heterophilous links introduce misleading feature mixing, which reduces the usefulness of label information during training and makes Actor less sensitive to label scarcity.

## C.5 ADDITIONAL RESULTS FOR DIRECTION-AWARE GNNS

In this section, we additionally include two widely used direction-aware GNNs, MagNet (Zhang et al., 2021) and DirGNN (Rossi et al., 2024), as supplementary baselines. Both models are evaluated on node-level, edge-level, and graph-level tasks, with the corresponding results reported in Table A11, Table A12, and Table A13, respectively.

Table A11: Node classification performance of direction-aware GNNs.

| Method | Cora | Pubmed | DBLP-CA | Walmart | Actor | Pokec |
|---|---|---|---|---|---|---|
| MagNet | $77.10_{\pm 1.35}$ | $86.12_{\pm 0.16}$ | $89.99_{\pm 0.31}$ | $71.81_{\pm 0.27}$ | $67.62_{\pm 0.56}$ | $57.01_{\pm 0.69}$ |
| DirGNN | $78.17_{\pm 0.81}$ | $86.50_{\pm 0.46}$ | $90.75_{\pm 0.28}$ | $73.78_{\pm 0.09}$ | $84.92_{\pm 0.49}$ | $58.47_{\pm 0.87}$ |

Table A12: Hyperedge prediction performance of direction-aware GNNs.

| Method | Cora | | Pubmed | | Actor | | Pokec | |
|---|---|---|---|---|---|---|---|---|
| | AUROC | AP | AUROC | AP | AUROC | AP | AUROC | AP |
| MagNet | $56.45_{\pm 0.02}$ | $55.18_{\pm 0.01}$ | $53.64_{\pm 0.02}$ | $54.79_{\pm 0.01}$ | $50.76_{\pm 0.02}$ | $50.21_{\pm 0.02}$ | $79.95_{\pm 0.01}$ | $80.78_{\pm 0.01}$ |
| DirGNN | $63.02_{\pm 0.02}$ | $61.38_{\pm 0.03}$ | $55.03_{\pm 0.02}$ | $55.28_{\pm 0.02}$ | $51.72_{\pm 0.02}$ | $51.33_{\pm 0.02}$ | $80.14_{\pm 0.01}$ | $79.65_{\pm 0.01}$ |

Table A13: Hypergraph classification performance of direction-aware GNNs.

| Method | RHG-10 | | IMDB-Dir-Genre | | Steam-Player | | Twitter-Friend | |
|---|---|---|---|---|---|---|---|---|
| | Acc | F1_ma | Acc | F1_ma | Acc | F1_ma | Acc | F1_ma |
| MagNet | $93.20_{\pm 0.02}$ | $92.95_{\pm 0.02}$ | $75.94_{\pm 0.01}$ | $71.45_{\pm 0.00}$ | $51.75_{\pm 0.02}$ | $51.12_{\pm 0.03}$ | $55.11_{\pm 0.02}$ | $46.64_{\pm 0.03}$ |
| DirGNN | $94.80_{\pm 0.01}$ | $94.68_{\pm 0.01}$ | $76.53_{\pm 0.02}$ | $72.59_{\pm 0.02}$ | $52.33_{\pm 0.01}$ | $52.24_{\pm 0.01}$ | $54.81_{\pm 0.04}$ | $46.02_{\pm 0.03}$ |

**Results and Analysis.** From the results shown in the tables above, we derive the following key findings: (1) In node classification, the two newly added direction-aware GNNs generally fall short of most HNN methods across the six datasets, reflecting the advantage of HNN architectures in modeling higher-order structures. We also observe that DirGNN achieves competitive performance on heterophilous datasets such as Actor and Pokec, likely because its separation mechanism in neighbor aggregation helps mitigate the adverse feature mixing effects induced by heterophily. (2) In hyperedge prediction, direction-aware GNNs perform notably worse than HNNs and, in many cases, even underperform traditional MLPs. A key reason is that their directional aggregation mechanism, which separates incoming and outgoing neighbors, reinforces a pairwise and asymmetric view of interactions. This asymmetry limits the model's ability to form coherent representations of multi-node groups and makes it difficult to capture the joint, order-invariant dependencies required for accurate hyperedge prediction. (3) In hypergraph classification, direction-aware GNNs remain less competitive than state-of-the-art HNNs, which benefit from explicit modeling of higher-order interactions that are crucial for capturing complex hypergraph structures.

C.6 ANALYZING PERFORMANCE DEGRADATION ON HETEROPHILOUS DATASETS

In this section, we investigate the underlying causes of performance degradation on heterophilous hypergraphs and test two key hypotheses: oversmoothing and feature collapse.

At the first step, we evaluate how the accuracy of four representative HNN architectures changes as the number of layers increases on two heterophilous datasets, Actor and Pokec, with the goal of examining whether oversmoothing occurs. According to Tables A14 and A15, although increasing depth generally causes a gradual performance decline in HNNs (i.e., oversmoothing), all HNN variants already underperform the MLP baseline under the 1-layer message passing. This suggests that depth is not the primary factor behind the performance gap.

Table A14: Node classification on Actor with varying layer depths.

| Method | 1 | 2 | 3 | 4 | 5 |
|---|---|---|---|---|---|
| MLP | $81.23 \pm 0.39$ | $86.06 \pm 0.36$ | $84.90 \pm 0.41$ | $84.55 \pm 0.54$ | $84.28 \pm 0.69$ |
| HGNN | $77.63 \pm 0.74$ | $73.84 \pm 0.37$ | $70.82 \pm 0.70$ | $68.59 \pm 0.68$ | $67.33 \pm 0.45$ |
| SheafHyperGNN | $85.00 \pm 0.32$ | $84.71 \pm 0.43$ | $83.61 \pm 0.48$ | $82.88 \pm 0.41$ | $82.15 \pm 0.63$ |
| AllSetTransformer | $85.79 \pm 0.77$ | $85.63 \pm 0.35$ | $85.68 \pm 0.55$ | $85.38 \pm 0.35$ | $85.49 \pm 0.21$ |
| ED-HNN | $85.69 \pm 0.45$ | $85.82 \pm 0.28$ | $85.53 \pm 0.37$ | $84.93 \pm 0.47$ | $82.60 \pm 9.96$ |

Table A15: Node classification on Pokec with varying layer depths.

| Method | 1 | 2 | 3 | 4 | 5 |
|---|---|---|---|---|---|
| MLP | $57.91 \pm 0.61$ | $59.64 \pm 0.48$ | $58.81 \pm 0.58$ | $58.52 \pm 0.85$ | $58.94 \pm 0.87$ |
| HGNN | $57.43 \pm 0.67$ | $57.48 \pm 0.82$ | $57.26 \pm 0.78$ | $56.88 \pm 1.24$ | $56.79 \pm 0.68$ |
| SheafHyperGNN | $59.02 \pm 0.42$ | $58.94 \pm 0.67$ | $58.26 \pm 0.61$ | $58.03 \pm 0.83$ | $57.93 \pm 0.73$ |
| AllSetTransformer | $58.75 \pm 0.48$ | $58.58 \pm 0.55$ | $58.50 \pm 0.85$ | $58.54 \pm 0.58$ | $58.35 \pm 0.34$ |
| ED-HNN | $58.52 \pm 0.32$ | $58.71 \pm 0.30$ | $58.74 \pm 0.50$ | $58.24 \pm 0.50$ | $58.11 \pm 0.58$ |

To further examine the underlying factors, we first compute the Mean Average Distance (MAD) (Chen et al., 2020), a widely adopted metric for measuring the smoothness (i.e., similarity) of graph representations. Specifically, we report the MAD values for both the raw input features and the representations obtained after the first layer. Next, to assess the extent of feature mixing under heterophily, we measure the similarity between each node and its heterophilous neighbors using the cosine distance. Formally, the heterophilous similarity is defined as:

$$\text{Sim}^{\text{diff}} = \text{avg}_{(i,j):\, j \in \mathcal{N}^{\text{diff}}(i)} \cos\left(h_i^{(l)}, h_j^{(l)}\right) \tag{5}$$

where $\mathcal{N}^{\text{diff}}(i) = \{j \in \mathcal{N}(i) \mid y_j \neq y_i\}$ denotes the set of heterophilous neighbors whose labels differ from that of node $i$. The results are reported in Tables A16 and A17.

Table A16: MAD and $\text{Sim}^{\text{diff}}$ values on Actor.

| Layer | HGNN | | SheafHyperGNN | | AllSetTransformer | | ED-HNN | | MLP | |
|---|---|---|---|---|---|---|---|---|---|---|
| | MAD | $\text{Sim}^{\text{diff}}$ | MAD | $\text{Sim}^{\text{diff}}$ | MAD | $\text{Sim}^{\text{diff}}$ | MAD | $\text{Sim}^{\text{diff}}$ | MAD | $\text{Sim}^{\text{diff}}$ |
| 0 | 0.8114 | 0.0584 | 0.8114 | 0.0584 | 0.8114 | 0.0584 | 0.8114 | 0.0584 | 0.8114 | 0.0584 |
| 1 | 0.4700 | 0.4013 | 0.3976 | 0.4274 | 0.2379 | 0.4892 | 0.5540 | 0.0515 | 0.7456 | -0.0829 |

Table A17: MAD and $\text{Sim}^{\text{diff}}$ values on Pokec.

| Layer | HGNN | | SheafHyperGNN | | AllSetTransformer | | ED-HNN | | MLP | |
|---|---|---|---|---|---|---|---|---|---|---|
| | MAD | $\text{Sim}^{\text{diff}}$ | MAD | $\text{Sim}^{\text{diff}}$ | MAD | $\text{Sim}^{\text{diff}}$ | MAD | $\text{Sim}^{\text{diff}}$ | MAD | $\text{Sim}^{\text{diff}}$ |
| 0 | 0.2697 | 0.0775 | 0.2697 | 0.0775 | 0.2697 | 0.0775 | 0.2697 | 0.0775 | 0.2697 | 0.0775 |
| 1 | 0.1054 | 0.6364 | 0.1490 | 0.7573 | 0.0058 | 0.9796 | 0.0592 | 0.7818 | 0.2034 | 0.2990 |

Our empirical analysis reveals two key observations: (1) After only 1-layer hypergraph message passing, MAD decreases sharply compared to the raw input features, indicating that node representations rapidly become more homogeneous. This demonstrates that HNN message passing

introduces representation smoothness at a very early stage. (2) The similarity between nodes and their heterophilous neighbors increases substantially, suggesting that heterophilous links cause strong cross-class feature mixing and pull representations of different classes closer together. Such mixing reduces class separability and ultimately impairs the effectiveness of HNN-based classifiers in heterophilous settings.

These observations align closely with prior theoretical and empirical findings on heterophilic GNNs. Existing studies (e.g., (Zhu et al., 2020; Luan et al., 2022; Yan et al., 2022)) suggest that heterophily may negatively affect message-passing architectures, because features of nodes from different classes are falsely mixed, leading to feature collapse and making nodes increasingly indistinguishable. Our results directly validate this hypothesis in the hypergraph setting: the sharp MAD reduction and pronounced cross-class similarity we observe mirror the failure patterns reported in these works.

## C.7 HYPEREDGE PREDICTION UNDER DIFFERENT DATA SPLITS

In this section, we conduct hyperedge prediction experiments under temporal and inductive split settings to account for potential temporal and inductive drift, thereby enabling more realistic evaluation scenarios.

### C.7.1 TEMPORAL SPLITS EVALUATION

**Experiment Settings.** Since the datasets in our current benchmark are static hypergraphs and therefore do not support temporal splits, we introduce two widely used temporal hypergraph datasets: the email network Email-Enron and the drug network NDC-Classes (Benson et al., 2018). Their detailed statistics are reported in Table A18. Based on timestamp information, we sort all hyperedges in ascending temporal order and let $T$ denote the maximum timestamp. Hyperedges with timestamps $\leq 0.6T$ are used for training, those within $(0.6T, 0.8T]$ form the validation set, and those with timestamps $> 0.8T$ constitute the test set, resulting in a 60%/20%/20% temporal split.

Table A18: Statistics of the two temporal hypergraphs.

| Dataset | # Nodes | # Edges | # Timestamps |
|---|---|---|---|
| Email-Enron | 1,161 | 49,724 | 5,891 |
| NDC-Classes | 143 | 10,883 | 10,788 |

Table A19: Hyperedge prediction performance under temporal splits.

| Method | Email-Enron | | NDC-Classes | |
|---|---|---|---|---|
| | AUROC | AP | AUROC | AP |
| HGNN | **87.30**$_{\pm 0.10}$ | **86.42**$_{\pm 0.21}$ | **94.22**$_{\pm 0.25}$ | **93.75**$_{\pm 0.49}$ |
| SheafHyperGNN | 80.17$_{\pm 0.52}$ | 80.85$_{\pm 0.85}$ | 91.97$_{\pm 0.17}$ | 92.03$_{\pm 0.06}$ |
| TF-HNN | 78.87$_{\pm 0.99}$ | 79.07$_{\pm 0.53}$ | 87.28$_{\pm 0.32}$ | 88.77$_{\pm 0.92}$ |
| UniGNN | 82.52$_{\pm 0.59}$ | 82.02$_{\pm 0.92}$ | 92.17$_{\pm 0.05}$ | 90.36$_{\pm 0.46}$ |
| ED-HNN | 76.97$_{\pm 1.13}$ | 76.29$_{\pm 0.23}$ | 75.93$_{\pm 2.11}$ | 76.04$_{\pm 0.74}$ |
| EHNN | 80.58$_{\pm 0.63}$ | 79.19$_{\pm 0.10}$ | 86.06$_{\pm 6.05}$ | 88.47$_{\pm 3.84}$ |

**Results and Analysis.** As shown in Table A19, HGNN outperforms all other HNN architectures on both temporal hypergraphs, suggesting a stronger capability to capture group-level temporal interaction patterns, making it more suitable for real-world higher-order relational prediction. In contrast, ED-HNN consistently achieves substantially lower predictive performance across both datasets. Moreover, all HNN models exhibit noticeably lower accuracy on Email-Enron compared to NDC-Classes, which may be attributed to the increased temporal complexity introduced by its larger number of nodes and hyperedges, thereby making inductive prediction more challenging.

### C.7.2 INDUCTIVE SPLITS EVALUATION

**Experiment Settings.** In the inductive setting, we divide the nodes of each dataset into three disjoint subsets for training, validation, and testing. Hyperedges in each split are constrained to include only nodes within the corresponding subset, ensuring a strictly disjoint node–hyperedge partition. In our experiments, we adopt a 40%/20%/40% split for the training, validation, and testing node sets, respectively.

Table A20: Hyperedge prediction performance under inductive splits.

| Method | Cora | | Pubmed | | Actor | |
|---|---|---|---|---|---|---|
| | AUROC | AP | AUROC | AP | AUROC | AP |
| HGNN | $74.07_{\pm8.50}$ | $76.81_{\pm9.75}$ | $65.18_{\pm9.41}$ | $63.09_{\pm9.67}$ | **$71.22_{\pm5.37}$** | $70.37_{\pm4.71}$ |
| SheafHyperGNN | $60.13_{\pm4.90}$ | $65.06_{\pm3.79}$ | $65.59_{\pm1.25}$ | $66.84_{\pm0.11}$ | $67.04_{\pm2.63}$ | **$71.44_{\pm3.54}$** |
| TF-HNN | **$80.81_{\pm4.68}$** | **$84.19_{\pm5.95}$** | $71.74_{\pm0.67}$ | $73.37_{\pm1.09}$ | $70.98_{\pm1.91}$ | $71.41_{\pm2.17}$ |
| UniGNN | $67.63_{\pm5.91}$ | $72.69_{\pm3.01}$ | $59.28_{\pm3.11}$ | $61.62_{\pm0.46}$ | $57.26_{\pm2.50}$ | $60.60_{\pm2.49}$ |
| ED-HNN | $54.33_{\pm3.93}$ | $59.44_{\pm1.23}$ | **$75.22_{\pm1.89}$** | **$76.74_{\pm2.02}$** | $67.89_{\pm3.64}$ | $68.33_{\pm3.65}$ |
| EHNN | $68.85_{\pm1.17}$ | $71.02_{\pm2.96}$ | $64.66_{\pm10.14}$ | $63.92_{\pm9.83}$ | $64.73_{\pm4.65}$ | $63.60_{\pm5.11}$ |

**Results and Analysis.** As shown in Table A20, TF-HNN typically ranks first or second across inductive hyperedge prediction datasets, indicating strong generalization to inductive distribution shift. In contrast, UniGNN performs noticeably worse in the inductive setting, particularly on Pubmed and Actor, suggesting that it is more sensitive to inductive drift. Moreover, our results suggest that inductive robustness may vary across datasets, as the same architecture does not always perform consistently on different hypergraphs. For example, ED-HNN achieves the best performance on Pubmed but the lowest on Cora. These observations collectively demonstrate that inductive hyperedge prediction remains a non-trivial challenge for current HNNs, and model behavior can vary substantially across datasets.

### C.8 BENCHMARKING HNNS IN SELF-SUPERVISED SETTINGS

In this section, we evaluate HNN models under self-supervised learning settings, incorporating pretraining–fine-tuning tracks into the benchmark to better reflect modern training practices.

**Experiment Settings.** We adopt two recently proposed hypergraph self-supervised learning methods, TriCL (Lee & Shin, 2023) and SE-HSSL (Li et al., 2024a), to pretrain different HNN architectures. The pretrained models are then fine-tuned on both node classification and hyperedge prediction tasks. For node classification, following (Lee & Shin, 2023; Li et al., 2024a), we use a 10%/10%/80% split of labeled nodes for training, validation, and testing, and report accuracy. For hyperedge prediction, we follow (Kim et al., 2024a) and adopt a 60%/20%/20% split of hyperedges, evaluating performance with AUROC and Average Precision (AP).

**Results and Analysis.** Based on the results reported in Tables A21 and A22, we observe that: (1) Different self-supervised training frameworks lead to noticeable variations in HNN backbone performance. Overall, models pretrained with SE-HSSL and subsequently fine-tuned achieve stronger and more consistent downstream performance than those trained under TriCL in most cases. (3) Even under the same SSL framework, HNNs may exhibit divergent performance across downstream tasks. For example, within TriCL, EHNN performs relatively worse on node classification but achieves top-ranked performance on hyperedge prediction. (3) Across both SSL frameworks, HNN architectures obtain substantially lower hyperedge prediction accuracy on the heterophilous Actor dataset. This suggests that existing self-supervised objectives may struggle to effectively capture higher-order relationships in strongly heterophilous hypergraphs.

### C.9 PERFORMANCE SENSITIVITY TO HYPEREDGE SIZE DISTRIBUTIONS

In this section, we empirically analyze the sensitivity of different HNN models to datasets containing a few very large hyperedges versus many small ones.

**Experiment Settings.** We construct modified datasets to systematically evaluate model sensitivity. Specifically, we define super-large hyperedges as those containing at least 10% of all nodes in the

Table A21: Node classification performance under self-supervised learning.

| Strategy | Method | Cora | Pubmed | Actor |
|---|---|---|---|---|
| TriCL | HGNN | $68.74_{\pm 2.42}$ | $80.74_{\pm 1.02}$ | $73.28_{\pm 2.13}$ |
| | SheafHyperGNN | $62.13_{\pm 4.34}$ | $77.14_{\pm 1.57}$ | $81.17_{\pm 0.26}$ |
| | TF-HNN | $64.79_{\pm 2.33}$ | $80.48_{\pm 1.23}$ | $78.60_{\pm 1.46}$ |
| | UniGNN | $67.55_{\pm 3.38}$ | $81.48_{\pm 1.83}$ | $78.92_{\pm 0.55}$ |
| | ED-HNN | $64.54_{\pm 3.20}$ | $80.17_{\pm 0.78}$ | $81.76_{\pm 0.92}$ |
| | EHNN | $62.37_{\pm 4.28}$ | $80.37_{\pm 0.73}$ | $78.03_{\pm 3.73}$ |
| SE-HSSL | HGNN | $\mathbf{72.79_{\pm 0.43}}$ | $82.67_{\pm 0.24}$ | $81.12_{\pm 0.67}$ |
| | SheafHyperGNN | $67.65_{\pm 1.57}$ | $83.11_{\pm 1.11}$ | $80.45_{\pm 1.04}$ |
| | TF-HNN | $68.00_{\pm 1.19}$ | $81.81_{\pm 0.68}$ | $79.88_{\pm 0.50}$ |
| | UniGNN | $70.51_{\pm 0.75}$ | $\mathbf{85.27_{\pm 0.10}}$ | $82.30_{\pm 0.79}$ |
| | ED-HNN | $\underline{70.95_{\pm 1.76}}$ | $83.71_{\pm 0.16}$ | $\mathbf{83.01_{\pm 0.93}}$ |
| | EHNN | $69.85_{\pm 2.72}$ | $82.03_{\pm 1.73}$ | $81.39_{\pm 1.24}$ |

Table A22: Hyperedge prediction performance under self-supervised learning.

| Strategy | Method | Cora | | Pubmed | | Actor | |
|---|---|---|---|---|---|---|---|
| | | AUROC | AP | AUROC | AP | AUROC | AP |
| TriCL | HGNN | $81.25_{\pm 6.13}$ | $81.64_{\pm 6.17}$ | $66.80_{\pm 5.44}$ | $65.45_{\pm 4.02}$ | $52.73_{\pm 5.37}$ | $53.43_{\pm 5.31}$ |
| | SheafHyperGNN | $69.87_{\pm 9.72}$ | $70.96_{\pm 9.37}$ | $51.21_{\pm 4.47}$ | $52.45_{\pm 3.87}$ | $50.45_{\pm 1.52}$ | $50.55_{\pm 1.01}$ |
| | TF-HNN | $79.53_{\pm 6.74}$ | $79.75_{\pm 6.77}$ | $71.42_{\pm 1.30}$ | $\underline{72.14_{\pm 1.10}}$ | $48.67_{\pm 2.33}$ | $49.64_{\pm 1.34}$ |
| | UniGNN | $77.50_{\pm 6.75}$ | $77.41_{\pm 7.05}$ | $68.97_{\pm 0.59}$ | $68.28_{\pm 0.26}$ | $45.43_{\pm 3.15}$ | $48.74_{\pm 1.68}$ |
| | ED-HNN | $78.82_{\pm 6.78}$ | $80.24_{\pm 6.63}$ | $67.74_{\pm 1.06}$ | $68.89_{\pm 1.45}$ | $51.39_{\pm 3.60}$ | $52.82_{\pm 2.73}$ |
| | EHNN | $81.25_{\pm 4.53}$ | $81.28_{\pm 4.90}$ | $71.01_{\pm 1.95}$ | $67.87_{\pm 3.16}$ | $\underline{53.27_{\pm 4.63}}$ | $51.99_{\pm 2.81}$ |
| SE-HSSL | HGNN | $\mathbf{85.31_{\pm 4.68}}$ | $\mathbf{85.22_{\pm 4.92}}$ | $\mathbf{73.18_{\pm 0.76}}$ | $70.07_{\pm 0.72}$ | $\mathbf{62.20_{\pm 4.31}}$ | $\mathbf{60.29_{\pm 2.61}}$ |
| | SheafHyperGNN | $55.18_{\pm 5.24}$ | $57.51_{\pm 3.75}$ | $56.42_{\pm 3.69}$ | $55.30_{\pm 2.71}$ | $42.16_{\pm 3.67}$ | $47.11_{\pm 1.59}$ |
| | TF-HNN | $\underline{84.74_{\pm 4.99}}$ | $\underline{84.29_{\pm 5.36}}$ | $\underline{72.24_{\pm 0.70}}$ | $\mathbf{73.83_{\pm 1.02}}$ | $50.97_{\pm 3.99}$ | $\underline{54.05_{\pm 2.89}}$ |
| | UniGNN | $82.20_{\pm 5.62}$ | $81.61_{\pm 5.96}$ | $69.38_{\pm 3.58}$ | $69.52_{\pm 2.97}$ | $47.41_{\pm 0.67}$ | $50.11_{\pm 0.39}$ |
| | ED-HNN | $78.33_{\pm 7.22}$ | $76.64_{\pm 4.53}$ | $68.52_{\pm 0.39}$ | $69.74_{\pm 0.49}$ | $52.97_{\pm 0.67}$ | $52.19_{\pm 1.10}$ |
| | EHNN | $67.94_{\pm 6.35}$ | $67.72_{\pm 6.27}$ | $69.21_{\pm 4.05}$ | $66.16_{\pm 5.33}$ | $50.03_{\pm 0.04}$ | $50.02_{\pm 0.03}$ |

Table A23: Hyperedge size sensitivity analysis on Cora.

| Method | 0 | 2 | 4 | 6 | 8 | 10 |
|---|---|---|---|---|---|---|
| HGNN | 77.90 | 77.22 | 75.66 | 74.74 | 72.35 | 67.86 |
| TF-HNN | 79.47 | 79.20 | 78.49 | 77.93 | 76.63 | 76.04 |
| AllSetTransformer | 78.02 | 77.87 | 77.34 | 76.45 | 76.10 | 75.24 |
| ED-HNN | 78.58 | 77.93 | 77.25 | 76.69 | 75.78 | 75.10 |
| EHNN | 76.51 | 76.01 | 75.98 | 75.98 | 76.13 | 76.04 |

Table A24: Hyperedge size sensitivity analysis on DBLP-CA.

| Method | 0 | 2 | 4 | 6 | 8 | 10 |
|---|---|---|---|---|---|---|
| HGNN | 91.00 | 90.42 | 89.81 | 88.93 | 88.32 | 87.30 |
| TF-HNN | 91.38 | 90.28 | 89.96 | 89.44 | 89.03 | 88.57 |
| AllSetTransformer | 91.51 | 90.95 | 90.34 | 89.48 | 88.31 | 87.29 |
| ED-HNN | 91.55 | 91.09 | 90.72 | 89.98 | 89.43 | 88.84 |
| EHNN | 90.47 | 90.47 | 90.44 | 90.48 | 90.50 | 90.51 |

hypergraph. We sort all hyperedges in descending order by size and iteratively merge them; once the merged hyperedge exceeds the super-large threshold, we restart the merging process for the next one. By controlling the number of constructed super-large hyperedges (0, 2, 4, 6, 8, and 10), where

Table A25: Hyperedge size sensitivity analysis on Actor.

| Method | 0 | 2 | 4 | 6 | 8 | 10 |
|---|---|---|---|---|---|---|
| HGNN | 77.83 | 77.91 | 77.94 | 77.93 | 77.90 | 77.72 |
| TF-HNN | 85.96 | 85.96 | 85.68 | 85.22 | 85.57 | 85.61 |
| AllSetTransformer | 85.66 | 85.69 | 85.68 | 85.63 | 85.84 | 85.70 |
| ED-HNN | 85.77 | 85.79 | 85.80 | 85.76 | 85.74 | 85.77 |
| EHNN | 86.21 | 86.05 | 86.19 | 86.18 | 86.07 | 85.93 |

Table A26: Hyperedge size sensitivity analysis on Pokec.

| Method | 0 | 2 | 4 | 6 | 8 | 10 |
|---|---|---|---|---|---|---|
| HGNN | 57.87 | 58.98 | 58.57 | 57.87 | 57.11 | 57.78 |
| TF-HNN | 59.17 | 59.13 | 59.18 | 59.06 | 59.07 | 59.22 |
| AllSetTransformer | 58.55 | 58.90 | 58.76 | 59.02 | 58.67 | 58.74 |
| ED-HNN | 58.68 | 58.71 | 59.05 | 58.82 | 58.91 | 58.98 |
| EHNN | 58.23 | 58.23 | 58.20 | 58.06 | 58.11 | 58.02 |

0 corresponds to the original dataset, we obtain variants that introduce only a few extremely large hyperedges while keeping all remaining ones small.

**Results and Analysis.** From Tables A23 to A26, we observe that: (1) On homophilic datasets, introducing only a few extremely large hyperedges while keeping the rest small consistently degrades model performance. As the proportion of these super-large hyperedges increases, performance generally continues to decline. This is likely because a small number of oversized hyperedges disrupt fine-grained local structure, causing the models to lose the class-consistent neighborhood signals that homophilic settings rely on. (2) On heterophilic datasets, increasing the proportion of super-large hyperedges generally maintains stable performance and may even yield slight improvements. A plausible explanation is that, in heterophilic settings, the presence of a small number of oversized hyperedges further weakens the influence of the original heterophilic connections during message passing, thereby reducing the impact of noisy or label-inconsistent neighbors. (3) Among all evaluated architectures, the tensor-based EHNN demonstrates the strongest robustness to extreme hyperedge-size skew: its performance remains stable across all constructed settings on both homophilic and heterophilic datasets.

## C.10 ANALYZING HNN BEHAVIOR ON EXTREME-DEGREE NODES

Table A27: Performance on very high-degree vs. very low-degree nodes ($p = 1\%$).

| Method | Cora | | DBLP-CA | | Actor | | Pokec | |
|---|---|---|---|---|---|---|---|---|
| | Very High | Very Low | Very High | Very Low | Very High | Very Low | Very High | Very Low |
| HGNN | 85.58 | 72.05 | 93.61 | 88.04 | 73.37 | 64.93 | 59.96 | 58.23 |
| PhenomNN | 85.59 | 73.47 | 93.97 | 89.70 | 93.18 | 66.51 | 64.60 | 57.42 |
| SheafHyperGNN | 85.48 | 74.38 | 94.16 | 88.22 | 82.42 | 74.10 | 64.29 | 58.41 |
| TF-HNN | 85.55 | 75.11 | 94.42 | 87.44 | 93.71 | 73.97 | 67.03 | 57.80 |
| UniGNN | 86.22 | 74.12 | 94.22 | 89.49 | 93.73 | 65.35 | 67.34 | 57.61 |
| AllSetTransformer | 85.36 | 73.25 | 95.37 | 89.12 | 94.18 | 72.55 | 68.24 | 57.11 |
| ED-HNN | 85.36 | 73.21 | 94.73 | 88.88 | 95.42 | 70.80 | 67.63 | 57.22 |
| EHNN | 83.58 | 69.93 | 95.24 | 86.73 | 95.84 | 75.66 | 63.99 | 57.62 |

In this section, we conduct experiments to compare the behavior of different HNNs on nodes with very high versus very low degrees.

**Experiment Settings.** To investigate this question, we design an experiment that explicitly contrasts model behavior on nodes with substantially different degrees. Specifically, we define very high–degree nodes as those whose degrees fall within the top-$p\%$ of the dataset, and very low–degree nodes as those in the bottom-$p\%$. To ensure robustness, we consider two thresholds, $p = 1$ and $p = 5$. Our study evaluates 8 representative HNN architectures spanning three major categories across four

Table A28: Performance on very high-degree vs. very low-degree nodes ($p = 5\%$).

| Method | Cora | | DBLP-CA | | Actor | | Pokec | |
|---|---|---|---|---|---|---|---|---|
| | Very High | Very Low | Very High | Very Low | Very High | Very Low | Very High | Very Low |
| HGNN | 81.82 | 72.05 | 93.38 | 88.13 | 76.93 | 76.39 | 58.39 | 57.80 |
| PhenomNN | 80.74 | 73.47 | 93.69 | 89.64 | 90.24 | 80.22 | 64.74 | 57.98 |
| SheafHyperGNN | 82.86 | 74.38 | 93.48 | 88.22 | 87.87 | 83.08 | 63.58 | 58.43 |
| TF-HNN | 81.91 | 75.11 | 93.78 | 87.45 | 90.88 | 83.97 | 65.87 | 57.80 |
| UniGNN | 81.15 | 74.17 | 93.58 | 89.48 | 90.45 | 79.05 | 66.01 | 57.63 |
| AllSetTransformer | 81.37 | 73.19 | 93.85 | 89.17 | 91.52 | 83.23 | 66.78 | 57.29 |
| ED-HNN | 81.02 | 73.21 | 93.88 | 88.89 | 91.71 | 82.99 | 65.77 | 57.18 |
| EHNN | 81.34 | 70.12 | 93.95 | 86.63 | 91.81 | 83.49 | 63.06 | 57.62 |

benchmark datasets. For each setting, we report the classification accuracy separately on the very high–degree and very low–degree subsets of the test nodes.

**Results and Analysis.** From the results summarized in Tables A27 and A28, we draw two key observations: (1) Across all datasets and all HNN architectures, we consistently observe a structural unfairness phenomenon: models achieve substantially higher accuracy on very high-degree nodes compared to very low-degree nodes. A plausible explanation is that high-degree nodes benefit more from message passing because they can aggregate richer and more reliable higher-order structural information, whereas low-degree nodes struggle to leverage structural signals and are more vulnerable to noise introduced by sparse or unreliable neighbors. (2) The performance disparity becomes more pronounced under stricter degree thresholds. When the threshold is reduced from 5% to 1%, the gap between very high-degree and very low-degree nodes typically increases substantially. This suggests that the most extreme-degree nodes exhibit the strongest disparity, further underscoring the critical role of degree heterogeneity in shaping HNN behavior.

These analyses provide a clearer understanding of how HNNs behave across extreme degree levels and reveal that improving the performance of low-degree nodes remains a key bottleneck in advancing HNN models. This highlights an important direction for future work: designing mechanisms that better enable low-degree nodes to exploit structural information.

# D  ADDITIONAL DISCUSSION AND ANALYSIS

## D.1  WHY DO HNNS PERFORM DIFFERENTLY ACROSS DATASETS

In this section, we systematically examine why HNN performance varies across datasets, as noted in the key insights for RQ1. Our analysis suggests that such variation may arise from both dataset characteristics and architectural design choices.

**Dataset-driven factors.** (1) Many advanced HNNs perform well on highly homophilous datasets but exhibit sharp degradation on heterophilous graphs, with performance frequently dropping below that of MLPs. This may be because heterophilous links mix features from different classes, leading to feature collapse and reduced class separability. (2) Performance for most HNN architectures consistently drops on large and structurally complex hypergraphs. For example, Trivago contains a large number of label categories, increasing classification difficulty, while Yelp exhibits extremely dense hyperedges that may over-mix signals during propagation. Interestingly, TF-HNN performs comparatively well on both datasets, suggesting that training-free hypergraph message passing may be more suitable for large, noisy, or highly complex real-world hypergraphs.

**Architecture-driven factors.** (1) Methods that involve explicit hypergraph expansion (e.g., HyperGCN, LEGCN, HJRL, DPHGNN) may unintentionally distort higher-order relationships by converting hyperedges into pairwise structures. This design often preserves performance on datasets dominated by isolated or pairwise interactions (e.g., Pubmed), but leads to noticeable degradation on datasets where many nodes participate in rich higher-order interactions (e.g., Cora, DBLP-CA, and NTU2012). (2) Spatial-based models (e.g., UniGNN, AllSetTransformer, ED-HNN) and TF-HNN generally provide more stable performance across homophilous and heterophilous datasets. Their skip-connection style message passing retains raw node information, helping mitigate feature dilution during higher-order propagation.

## D.2 Trade-offs among Performance, Scalability, and Data Characteristics

For spectral-based models, most advanced approaches (e.g., PhenomNN, SheafHyperGNN, HJRL) consistently outperform earlier variants such as HGNN, HyperGCN, and HCHA on homophilous datasets. However, this accuracy gain relies on more complex expansion mechanisms and Laplacian operators, which substantially reduce scalability. As shown in Table 1, they frequently encounter OOM issues on large or dense hypergraphs such as Trivago and Yelp. TF-HNN provides a lightweight alternative, achieving top-ranked performance on most datasets while maintaining strong scalability due to its training-free message-passing design. Spatial-based architectures generally offer a more favorable scalability–performance balance. Models such as UniGNN, AllSetTransformer, and ED-HNN deliver accuracy comparable to advanced spectral methods on homophilous data with substantially lower memory consumption. Tensor-based methods (e.g., EHNN and T-HyperGNN) perform worse on homophilous datasets, but relative to spectral- and spatial-based HNNs, they often achieve better performance on heterophilous benchmarks, particularly EHNN, which also demonstrates stronger scalability than T-HyperGNN. Although MLPs perform substantially worse than HNNs on homophilous datasets, they often excel on heterophilous benchmarks and outperform many HNN architectures. Furthermore, by removing high-order message passing, MLPs achieve markedly better scalability.

## E  Related Works

Hypergraph neural networks (HNNs) (Yadati et al., 2019; Prokopchik et al., 2022; Wang et al., 2023a; Xie et al., 2025) have been promising tools for handling learning tasks involving higher-order data, with notable applications in various fields, such as social network analysis (Sun et al., 2023; Yang et al., 2023; Tan et al., 2025c), bioinformatics (Luo et al., 2024; Li et al., 2025a; Xu et al., 2026), and recommender systems (Li et al., 2025b; Tan et al., 2025b; Zai et al., 2025). However, there exists no established benchmark specifically dedicated to comprehensively evaluating hypergraph neural networks. In this section, we introduce a broader range of related studies concerning the comparative evaluations of HNNs, providing sufficient context for our benchmark work.

Kim et al. (Kim et al., 2024b) recently presented the first survey dedicated to HNNs, with an in-depth and step-by-step guide. The survey comprehensively reviews existing HNN architectures, training strategies, and applications, establishing a foundational understanding crucial for advancing the field of HNNs. To further understand the expressive power of HNNs, Wang et al. (Wang et al., 2025) conduct the first theoretical analysis on the generalization performance of distinct HNN architectures, offering practical guidance for improving HNNs' effectiveness. Nevertheless, systematic empirical evaluations of different HNN algorithms remain scarce, leaving a limited understanding of their comparative performance in practice. To facilitate the reproducibility and empirical evaluation of HNN algorithms, several open-sourced libraries have been developed in recent years. HyFER (Hwang et al., 2021) is a well-modularized framework for implementing and evaluating HNNs, dividing the entire learning process into data, model, and task components. Moreover, to address the scalability problem that most existing implementations suffer from, HyFER is built on top of Deep Graph Library (DGL) (Wang et al., 2019), which is a highly efficient open-sourced library for GNNs. DHG (Gao et al., 2022) is an open-sourced PyTorch-based toolbox designed for general HNNs. It supports various hypergraph preprocessing methods (e.g., sampling, expansion) and convolution operators, facilitating the evaluation of HNNs. TopoX (Hajij et al., 2024) is a suite of Python packages for machine learning on topological domains. These packages enhance and generalize functionalities found in mainstream hypergraph computations and learning tools, enabling them on topological domains. TopoBench (Telyatnikov et al., 2024) is a modular Python library that standardizes benchmarking and accelerates research in Topological Deep Learning (TDL). It supports training and comparing Topological Neural Networks (TNNs) across diverse domains, including graphs, simplicial complexes, cellular complexes, and hypergraphs. However, these libraries do not fully cover the latest HNN algorithms, datasets, and evaluation tasks, and they provide only limited empirical results without offering an in-depth and comprehensive analysis of existing HNN methods.

To fill the gap, we develop DHG-Bench, the first comprehensive benchmark tailored explicitly for HNNs. Distinguished by its broad coverage, DHG-Bench spans a wide range of algorithms, datasets, and evaluation tasks, thereby establishing a standardized and versatile framework for deep hypergraph learning research. Moreover, it provides comprehensive and systematic empirical evaluations that

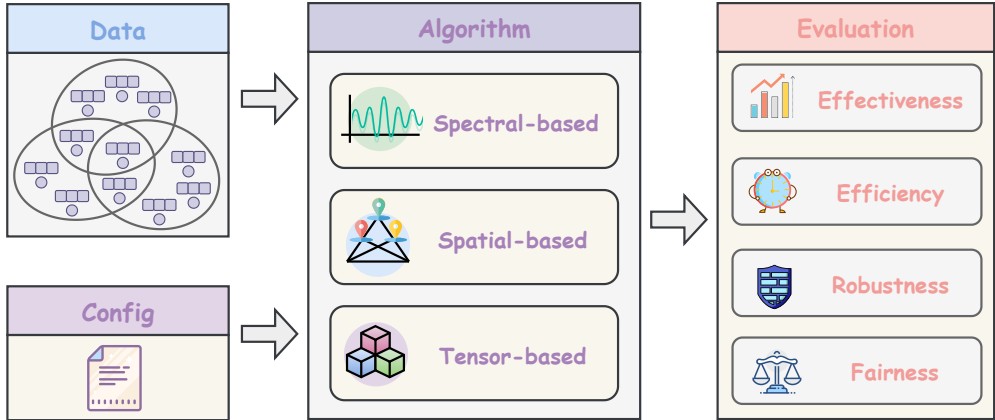

Figure A5: The package structure of DHG-Bench, which mainly consists of four modules.

uncover the strengths and limitations of different algorithms. By offering such in-depth quantitative analyses, our benchmark fosters deeper insights into the challenges and opportunities of HNNs, thereby advancing the state-of-the-art in this emerging field.

## F  PACKAGE

We have developed DHG-Bench [3], an open-sourced package that provides a comprehensive and unbiased platform for evaluating HNN algorithms and supporting future research in this domain.

As shown in Figure A5, the code structure is well-designed to ensure fair experimental setups across different algorithms, easy reproduction of the experimental results, and support for flexible assembly of models for experiments. The DHG-Bench consists of the following four key modules. ❶ The Config module includes the files that define the necessary hyperparameters and settings. ❷ The Data module is used to load and preprocess datasets. ❸ The Algorithm module has 17 built-in state-of-the-art algorithms, covering three representative categories: spectral-based, spatial-based, and tensor-based methods. ❹ The evaluation module supports multi-faceted testing of algorithmic performance, encompassing effectiveness, efficiency, robustness, and fairness.

## G  THE USE OF LLMS

We used large language models (LLMs) solely as a writing assistant to polish the paper, specifically for grammar checking and typo correction. In addition, LLMs were occasionally consulted to rephrase sentences for improved readability and to ensure a consistent academic tone. No part of the technical content, experimental design, or analysis was generated by LLMs. Their role was strictly limited to minor linguistic refinement.

---

[3]https://github.com/Coco-Hut/DHG-Bench

