# OpenReview forum: "DHG-Bench: A Comprehensive Benchmark for Deep Hypergraph Learning"
_ICLR.cc/2026/Conference — ICLR 2026 Poster_

### Official Review · Reviewer_FzKL · 2025-10-22

**Soundness:** 2
**Presentation:** 3
**Contribution:** 2
**Rating:** 4
**Confidence:** 4

**Summary:**

This paper introduces a large benchmark for hypergraph neural networks (HNNs), consisting of 17 HNN methods across 22 datasets. The benchmark standardizes training/evaluation spanning node-, hyperedge-, and graph-level tasks, then analyses four dimensions: effectiveness, efficiency, robustness, and fairness. Empirically, no model dominates and models resist structural noise but are sensitive to feature/label noise.

**Strengths:**

1. The work organizes experiments around RQ1–RQ4 (effectiveness, efficiency, robustness, fairness), with clear task coverage across node/edge/graph; complete leaderboards appear in the appendix with standardized splits and operators.
2. Findings such as no single HNN dominates, accuracy–efficiency trade-offs, and fairness brittleness of message passing are likely to steer algorithm development (e.g., toward robust/efficient or debiased HNNs).

**Weaknesses:**

1. Hyperedge prediction uses random splits with mixed heuristic negative sampling. This ignores temporal/inductive drift and can create unrealistic candidate sets. The paper can be strengthened by adding (a) temporal splits (train earlier hyperedges, predict future ones), (b) inductive splits with disjoint node/hyperedge partitions, and (c) open-world negatives drawn from realistic candidate generators (size- and motif-conditioned but respecting time).
2. All tasks are supervised. Many real deployments use self-supervised pretraining or contrastive objectives. The benchmark can be strengthened by adding self-supervised learning tracks (node/hyperedge masking, motif prediction, contrastive pretraining) and evaluate fine-tuning across tasks to reflect modern practice.
3. The paper successfully demonstrates that HNN performance varies significantly across datasets (a key insight for RQ1), but it falls short of explaining why. The analysis largely stops at reporting performance rankings.

**Questions:**

1. Is performance sensitive to datasets with a few very large hyperedges versus many small ones?
2. How do different models handle nodes with very high vs. low degrees?
3. Given the results, what is the recommended decision-making process for a practitioner when selecting an HNN model for a new task?
4. What are the key trade-offs, supported by the benchmark's findings, that one should consider between performance, scalability, and specific data characteristics (e.g., homophily)?

---

> ### Author Response · Authors · 2025-11-25
> **Overall Response to Reviewer FZKL**
>
> We sincerely thank the reviewer for recognizing the strengths of our work. We are glad that the comprehensive experimental design, task coverage, standardized splits, and complete leaderboards are found valuable. We also appreciate the acknowledgment of our key empirical insights, including the lack of a universally dominant HNN, accuracy–efficiency trade-offs, and fairness brittleness, which we hope will inform future HNN development.
>
> We respond to your concerns and questions as follows.
>
> **[Preview]**
>
> * **Weakness #1:** Random hyperedge splits with mixed heuristic negative sampling
> * **Weakness #2:** Lack of self-supervised learning tracks
> * **Weakness #3:** Lack of performance analysis
> * **Question #1:** Sensitivity to hyperedge size
> * **Question #2:** High- vs. low-degree node behavior
> * **Question #3:** Selection of HNN for a new task
> * **Question #4:** Trade-offs among performance, scalability, and data characteristics
> * **Conclusion.**

---

> ### Author Response · Authors · 2025-11-25
> **Response to Weakness #1 (Part I)**
>
> > **Weakness #1:** Hyperedge prediction uses random splits with mixed heuristic negative sampling. This ignores temporal/inductive drift and can create unrealistic candidate sets. The paper can be strengthened by adding (a) temporal splits (train earlier hyperedges, predict future ones), (b) inductive splits with disjoint node/hyperedge partitions, and (c) open-world negatives drawn from realistic candidate generators (size- and motif-conditioned but respecting time).
>
> **A:** Thank you for your insightful feedback. We first clarify that **we follow the standard hyperedge prediction evaluation protocol established in prior works [1,2,3]**, ensuring consistency with the existing literature. Specifically, we use random hyperedge splits with a 60%/20%/20% train/validation/test ratio, along with three heuristic-based negative sampling strategies (Size-NS, Motif-NS, and Clique-NS) to construct negative hyperedge sets.
>
> Regarding negative sampling, we would like to clarify that **Size-NS is a size-conditioned sampler that draws non-hyperedges whose size distribution matches that of real hyperedges**. **Motif-NS is a motif-conditioned strategy that performs clique expansion and samples connected components in the expanded graph**. By combining these strategies, we can produce negative hyperedges whose structural and size statistics resemble realistic candidate sets. We fully agree that more realistic candidate generators for negative samples could further strengthen the benchmark, and we consider this a promising direction for future work.
>
> In the revision, following your suggestions, **we have now provided temporal splits and inductive splits** to account for potential temporal and inductive drift, thereby enabling more realistic evaluation scenarios. The detailed settings and experimental results are provided below.
>
> ***1. Temporal Splits Evaluation***
>
> * **Settings.** Since the datasets in our current benchmark are static hypergraphs and therefore do not support temporal splits, we introduce two widely used temporal hypergraph datasets: the email network Email-Enron and the drug network NDC-Classes [4]. Their detailed statistics are reported in Table R1. Based on timestamp information, we sort all hyperedges in ascending temporal order and let $T$ denote the maximum timestamp. Hyperedges with timestamps $\le 0.6T$ are used for training, those within $(0.6T, 0.8T]$ form the validation set, and those with timestamps $> 0.8T$ constitute the test set, resulting in a 60\%/20\%/20\% temporal split.
>
> **Table R1:** Statistics of the two temporal hypergraphs.
>
> | Dataset | # Nodes | # Edges | # Timestamps |
> | :----: |:----: |:----: |:----: |
> | Email-Enron  | 1,161 | 49,724 | 5,891 |
> | NDC-Classes  |  143 | 10,883 | 10,788 |
>
> * **Results.**
>
> **Table R2:** Hyperedge prediction with temporal splits.
>
> | Dataset  | HGNN (AUROC \| AP) | SheafHyperGNN (AUROC \| AP) | TF-HNN (AUROC \| AP) | UniGNN (AUROC \| AP) | ED-HNN (AUROC \| AP) | EHNN (AUROC \| AP)|
> | :----:  | :----: | :----: | :----: | :----: | :----: |  :----: |
> | Email-Enron  | 87.30 $\pm$ 0.10 \| 86.42 $\pm$ 0.21 | 80.17 $\pm$ 0.52 \| 80.85 $\pm$ 0.85 | 78.87 $\pm$ 0.99 \| 79.07 $\pm$ 0.53 | 82.52 $\pm$ 0.59 \| 82.02 $\pm$ 0.92 | 76.97 $\pm$ 1.13 \| 76.29 $\pm$ 0.23 | 80.58 $\pm$ 0.63 \| 79.19 $\pm$ 0.10 |
> | NDC-Classes  | 94.22 $\pm$ 0.25 \| 93.75 $\pm$ 0.49 | 91.97 $\pm$ 0.17 \| 92.03 $\pm$ 0.06 | 87.28 $\pm$ 0.32 \| 88.77 $\pm$ 0.92 | 92.17 $\pm$ 0.05 \| 90.36 $\pm$ 0.46 | 75.93 $\pm$ 2.11 \| 76.04 $\pm$ 0.74 | 86.06 $\pm$ 6.05 \| 88.47 $\pm$ 3.84 |
>
> * **Findings.** As shown in Table R2, HGNN outperforms all other HNN architectures on both temporal hypergraphs, suggesting a stronger capability to capture group-level temporal interaction patterns, making it more suitable for real-world higher-order relational prediction. In contrast, ED-HNN consistently achieves substantially lower predictive performance across both datasets. Moreover, all HNN models exhibit noticeably lower accuracy on Email-Enron compared to NDC-Classes, which may be attributed to the increased temporal complexity introduced by its larger number of nodes and hyperedges, thereby making inductive prediction more challenging.

---

> ### Author Response · Authors · 2025-11-25
> **Response to Weakness #1 (Part II)**
>
> ***2. Inductive Splits Evaluation***
>
> * **Settings.** In the inductive setting, we divide the nodes of each dataset into three disjoint subsets for training, validation, and testing. Hyperedges in each split are constrained to include only nodes within the corresponding subset, ensuring a strictly disjoint node–hyperedge partition. In our experiments, we adopt a 40%/20%/40% split for the training, validation, and testing node sets, respectively.
>
> * **Results.**
>
> **Table R3:** Hyperedge prediction with inductive splits.
>
> | Dataset  | HGNN (AUROC \| AP) | SheafHyperGNN (AUROC \| AP)| TF-HNN (AUROC \| AP)| UniGNN (AUROC \| AP)| ED-HNN (AUROC \| AP)| EHNN (AUROC \| AP)|
> | :----:  | :----: | :----: | :----: | :----: | :----: |  :----: |
> | Cora  | 74.07 $\pm$ 8.50 \| 76.81 $\pm$ 9.75 | 60.13 $\pm$ 4.90 \| 65.06 $\pm$ 3.79 | 80.81 $\pm$ 4.68 \| 84.19 $\pm$ 5.95 | 67.63 $\pm$ 5.91 \| 72.69 $\pm$ 3.01 | 54.33 $\pm$ 3.93 \| 59.44 $\pm$ 1.23 | 68.85 $\pm$ 1.17 \| 71.02 $\pm$ 2.96 |
> | Pubmed  | 65.18 $\pm$ 9.41 \| 63.09 $\pm$ 9.67 | 65.59 $\pm$ 1.25 \| 66.84 $\pm$ 0.11 | 71.74 $\pm$ 0.67 \| 73.37 $\pm$ 1.09 | 59.28 $\pm$ 3.11 \| 61.62 $\pm$ 0.46 | 75.22 $\pm$ 1.89 \| 76.74 $\pm$ 2.02 | 64.66 $\pm$ 10.14 \| 63.92 $\pm$ 9.83 |
> | Actor  | 71.22 $\pm$ 5.37 \| 70.37 $\pm$ 4.71 | 67.04 $\pm$ 2.63 \| 71.44 $\pm$ 3.54 | 70.98 $\pm$ 1.91 \| 71.41 $\pm$ 2.17 | 57.26 $\pm$ 2.50 \| 60.60 $\pm$ 2.49 | 67.89 $\pm$ 3.64 \| 68.33 $\pm$ 3.65 | 64.73 $\pm$ 4.65 \| 63.60 $\pm$ 5.11 |
>
> * **Findings.** As shown in Table R3, TF-HNN typically ranks first or second across inductive hyperedge prediction datasets, indicating strong generalization to inductive distribution shift. In contrast, UniGNN performs noticeably worse in the inductive setting, particularly on Pubmed and Actor, suggesting that it is more sensitive to inductive drift. Moreover, our results suggest that inductive robustness may vary across datasets, as the same architecture does not always perform consistently on different hypergraphs. For example, ED-HNN achieves the best performance on Pubmed but the lowest on Cora. These observations collectively demonstrate that inductive hyperedge prediction remains a non-trivial challenge for current HNNs, and model behavior can vary substantially across datasets.
>
> **The above experimental results and analysis have been incorporated into Appendix C.7 in the revised manuscript.** We sincerely appreciate your constructive suggestion!

---

> ### Author Response · Authors · 2025-11-25
> **Response to Weakness #2 (Part I)**
>
> > **Weakness #2:** All tasks are supervised. Many real deployments use self-supervised pretraining or contrastive objectives. The benchmark can be strengthened by adding self-supervised learning tracks (node/hyperedge masking, motif prediction, contrastive pretraining) and evaluate fine-tuning across tasks to reflect modern practice.
>
> **A:** Thank you for your valuable suggestions. We would like to clarify that DHG-Bench currently focuses on supervised learning, as the vast majority of existing HNN literature adopts supervised prediction tasks [5,6,7], making it the dominant evaluation paradigm. **Building a standardized and comprehensive supervised benchmark is essential for two reasons**. (1) Prior HNN studies rely on different datasets, baselines, and experimental setups (e.g., splitting strategies and hyperparameter choices), which hinders fair and reproducible comparison. (2) Existing supervised evaluations mainly emphasize effectiveness, while offering limited empirical understanding of efficiency and trustworthiness (e.g., robustness and fairness). This gap prevents a holistic assessment of the strengths and limitations of different HNN architectures.
>
> We acknowledge that several recent works have begun exploring **self-supervised hypergraph learning** [8,9,10]. However, **these studies primarily focus on designing pretext signals and training objectives under label scarcity**, rather than comparing or benchmarking different HNN architectures, **making them orthogonal to the current scope of our study**. For this reason, self-supervised tracks were not included in the initial version.
>
> Following your suggestion, **we have now incorporated self-supervised learning tracks into the benchmark** to better reflect modern training practices and enable evaluation under pretraining–fine-tuning settings.
>
> * **Settings.** We adopt two recently proposed hypergraph self-supervised learning methods, **TriCL [8] and SE-HSSL [9]**, to pretrain different HNN architectures. The pretrained models are then fine-tuned on both **node classification** and **hyperedge prediction** tasks. For node classification, following [8, 9], we use a 10%/10%/80% split of labeled nodes for training, validation, and testing, and report accuracy. For hyperedge prediction, we follow [10] and adopt a 60%/20%/20% split of hyperedges, evaluating performance with AUROC and Average Precision (AP).
>
> * **Results.**
>
> **Table R4:** Node classification performance with TriCL pretraining.
>
> | Dataset  | HGNN | SheafHyperGNN | TF-HNN | UniGNN | ED-HNN | EHNN |
> | :----:  | :----: |  :----: | :----: | :----: |  :----: | :----: |
> | Cora  | 68.74 $\pm$ 2.42 | 62.13 $\pm$ 4.34 | 64.79 $\pm$ 2.33 | 67.55 $\pm$ 3.38 | 64.54 $\pm$ 3.20 | 62.37 $\pm$ 4.28 |
> | Pubmed  | 80.74 $\pm$ 1.02 | 77.14 $\pm$ 1.57 | 80.48 $\pm$ 1.23 | 81.48 $\pm$ 1.83 | 80.17 $\pm$ 0.78 | 80.37 $\pm$ 0.73 |
> | Actor  | 73.28 $\pm$ 2.13 | 81.17 $\pm$ 0.26 | 78.60 $\pm$ 1.46 | 78.92 $\pm$ 0.55 | 81.76 $\pm$ 0.92 | 78.03 $\pm$ 3.73 |
>
> **Table R5:** Hyperedge prediction performance with TriCL pretraining.
>
> | Dataset | HGNN (AUROC \| AP) | SheafHyperGNN (AUROC \| AP) | TF-HNN (AUROC \| AP) | UniGNN (AUROC \| AP) | ED-HNN (AUROC \| AP) | EHNN (AUROC \| AP) |
> | :----: | :----: | :----: | :----: | :----: | :----: | :----: |
> | Cora | 81.25 $\pm$ 6.13 \| 81.64 $\pm$ 6.17 | 69.87 $\pm$ 9.72 \| 70.96 $\pm$ 9.37 | 79.53 $\pm$ 6.74 \| 79.75 $\pm$ 6.77 | 77.50 $\pm$ 6.75 \| 77.41 $\pm$ 7.05 | 78.82 $\pm$ 6.78 \| 80.24 $\pm$ 6.63 | 81.25 $\pm$ 4.53 \| 81.28 $\pm$ 4.90 |
> | Pubmed | 66.80 $\pm$ 5.44 \| 65.45 $\pm$ 4.02 | 51.21 $\pm$ 4.47 \| 52.45 $\pm$ 3.87 | 71.42 $\pm$ 1.30 \| 72.14 $\pm$ 1.10 | 68.97 $\pm$ 0.59 \| 68.28 $\pm$ 0.26 | 67.74 $\pm$ 1.06 \| 68.89 $\pm$ 1.45 | 71.01 $\pm$ 1.95 \| 67.87 $\pm$ 3.16 |
> | Actor | 52.73 $\pm$ 5.37 \| 53.43 $\pm$ 5.31 | 50.45 $\pm$ 1.52 \| 50.55 $\pm$ 1.01 | 48.67 $\pm$ 2.33 \| 49.64 $\pm$ 1.34 | 45.43 $\pm$ 3.15 \| 48.74 $\pm$ 1.68 | 51.39 $\pm$ 3.60 \| 52.82 $\pm$ 2.73 | 53.27 $\pm$ 4.63 \| 51.99 $\pm$ 2.81 |

---

> ### Author Response · Authors · 2025-11-25
> **Response to Weakness #2 (Part II)**
>
> **Table R6:** Node classification performance with SE-HSSL pretraining.
>
> | Dataset  | HGNN | SheafHyperGNN | TF-HNN | UniGNN | ED-HNN | EHNN |
> | :----:  | :----: | :----: | :----: | :----: | :----: |  :----: |
> | Cora  | 72.79 $\pm$ 0.43 | 67.65 $\pm$ 1.57 | 68.00 $\pm$ 1.19 | 70.51 $\pm$ 0.75 | 70.95 $\pm$ 1.76 | 69.85 $\pm$ 2.72 |
> | Pubmed  | 82.67 $\pm$ 0.24 | 83.11 $\pm$ 1.11 | 81.81 $\pm$ 0.68 | 85.27 $\pm$ 0.10 | 83.71 $\pm$ 0.16 | 82.03 $\pm$ 1.73 |
> | Actor  | 81.12 $\pm$ 0.67 | 80.45 $\pm$ 1.04 | 79.88 $\pm$ 0.50 | 82.30 $\pm$ 0.79 | 83.01 $\pm$ 0.93 | 81.39 $\pm$ 1.24 |
>
> **Table R7:** Hyperedge prediction performance with SE-HSSL pretraining.
>
> | Dataset  | HGNN (AUROC \| AP) | SheafHyperGNN (AUROC \| AP) | TF-HNN (AUROC \| AP) | UniGNN (AUROC \| AP) | ED-HNN (AUROC \| AP) | EHNN (AUROC \| AP) |
> | :----:  | :----: | :----: | :----: | :----: | :----: | :----: |
> | Cora  | 85.31 $\pm$ 4.68 \| 85.22 $\pm$ 4.92 | 55.18 $\pm$ 5.24 \| 57.51 $\pm$ 3.75 | 84.74 $\pm$ 4.99 \| 84.29 $\pm$ 5.36 | 82.20 $\pm$ 5.62 \| 81.61 $\pm$ 5.96 | 78.33 $\pm$ 7.22 \| 76.64 $\pm$ 4.53 | 67.94 $\pm$ 6.35 \| 67.72 $\pm$ 6.27 |
> | Pubmed  | 73.18 $\pm$ 0.76 \| 70.07 $\pm$ 0.72 | 56.42 $\pm$ 3.69 \| 55.30 $\pm$ 2.71 | 72.24 $\pm$ 0.70 \| 73.83 $\pm$ 1.02 | 69.38 $\pm$ 3.58 \| 69.52 $\pm$ 2.97 | 68.52 $\pm$ 0.39 \| 69.74 $\pm$ 0.49 | 69.21 $\pm$ 4.05 \| 66.16 $\pm$ 5.33 |
> | Actor  | 62.20 $\pm$ 4.31 \| 60.29 $\pm$ 2.61 | 42.16 $\pm$ 3.67 \| 47.11 $\pm$ 1.59 | 50.97 $\pm$ 3.99 \| 54.05 $\pm$ 2.89 | 47.41 $\pm$ 0.67 \| 50.11 $\pm$ 0.39 | 52.97 $\pm$ 0.67 \| 52.19 $\pm$ 1.10 | 50.03 $\pm$ 0.04 \| 50.02 $\pm$ 0.03 |
>
> * **Findings.** Based on the results reported in Tables R4–R7, we observe that:
>
>     - Different self-supervised training frameworks lead to noticeable variations in HNN backbone performance. Overall, models pretrained with SE-HSSL and subsequently fine-tuned achieve stronger and more consistent downstream performance than those trained under TriCL in most cases.
>     - Even under the same SSL framework, HNNs may exhibit divergent performance across downstream tasks. For example, within TriCL, EHNN performs relatively worse on node classification but achieves top-ranked performance on hyperedge prediction.
>     - Across both SSL frameworks, HNN architectures obtain substantially lower hyperedge prediction accuracy on the heterophilous Actor dataset. This suggests that existing self-supervised objectives may struggle to effectively capture higher-order relationships in strongly heterophilous hypergraphs.
>
> **We have updated the manuscript and incorporated the above results and analysis into Appendix C.8.** Thank you for your constructive feedback!

---

> ### Author Response · Authors · 2025-11-25
> **Response to Weakness #3**
>
> > **Weakness #3:** The paper successfully demonstrates that HNN performance varies significantly across datasets (a key insight for RQ1), but it falls short of explaining why. The analysis largely stops at reporting performance rankings.
>
> **A:** Thank you for your constructive feedback. We agree that explaining why HNN performance varies across datasets is essential, and we have added a more systematic discussion accordingly. Our analysis suggests that such variation may stem from both **dataset characteristics** and **architectural design choices**:
>
> * **Dataset-driven factors**
>
>   - **Homophily vs. heterophily.** Many advanced HNNs perform well on highly homophilous datasets but exhibit sharp degradation on heterophilous graphs, with performance frequently dropping below that of MLPs. This may be because heterophilous links mix features from different classes, leading to feature collapse and reduced class separability.
>
>   - **Scale, density, and label complexity.** Performance for most HNN architectures consistently drops on large and structurally complex hypergraphs. For example, Trivago contains a large number of label categories, increasing classification difficulty, while Yelp exhibits extremely dense hyperedges that may over-mix signals during propagation. Interestingly, TF-HNN performs comparatively well on both datasets, suggesting that training-free hypergraph message passing may be more suitable for large, noisy, or highly complex real-world hypergraphs.
>
> * **Architecture-driven factors**
>
>   - **Hypergraph expansion mechanisms.** Methods that involve explicit hypergraph expansion (e.g., HyperGCN, LEGCN, HJRL, DPHGNN) may unintentionally distort higher-order relationships by converting hyperedges into pairwise structures. This design often preserves performance on datasets dominated by isolated or pairwise interactions (e.g., Pubmed), but leads to noticeable degradation on datasets where many nodes participate in rich higher-order interactions (e.g., Cora, DBLP-CA, and NTU2012).
>
>   - **Skip connections and message-passing robustness.** Spatial-based models (e.g., UniGNN, AllSetTransformer, ED-HNN) and TF-HNN generally provide more stable performance across homophilous and heterophilous datasets. Their skip-connection style message passing retains raw node information, helping mitigate feature dilution during higher-order propagation.
>
> **We have incorporated this discussion into Appendix D.1 in the revised manuscript**, which we believe further strengthens the contribution and offers useful insights for future research. Thank you again for the helpful suggestion.

---

> ### Author Response · Authors · 2025-11-25
> **Response to Question #1**
>
> > **Question #1:** Is performance sensitive to datasets with a few very large hyperedges versus many small ones?
>
> **A:** Thank you for your question! In response, we empirically analyze the sensitivity of different HNN models to datasets containing a few very large hyperedges versus many small ones.
>
> * **Settings.** We construct modified datasets to systematically evaluate model sensitivity. Specifically, we define **super-large hyperedges** as those containing **at least 10% of all nodes in the hypergraph**. We sort all hyperedges in descending order by size and iteratively merge them; once the merged hyperedge exceeds the super-large threshold, we restart the merging process for the next one. By controlling the number of constructed super-large hyperedges (0, 2, 4, 6, 8, and 10), where 0 corresponds to the original dataset, we obtain variants that introduce only a few extremely large hyperedges while keeping all remaining ones small.
>
> * **Results.**
>
> **Table R8:** Hyperedge size sensitivity analysis on Cora.
>
> | Method             | 0     | 2     | 4     | 6     | 8     | 10    |
> |-------------------|-------|-------|-------|-------|-------|-------|
> | HGNN              | 77.90 | 77.22 | 75.66 | 74.74 | 72.35 | 67.86 |
> | TF-HNN            | 79.47 | 79.20 | 78.49 | 77.93 | 76.63 | 76.04 |
> | AllSetTransformer | 78.02 | 77.87 | 77.34 | 76.45 | 76.10 | 75.24 |
> | ED-HNN            | 78.58 | 77.93 | 77.25 | 76.69 | 75.78 | 75.10 |
> | EHNN              | 76.51 | 76.01 | 75.98 | 75.98 | 76.13 | 76.04 |
>
> **Table R9:** Hyperedge size sensitivity analysis on DBLP-CA.
>
> | Method             | 0     | 2     | 4     | 6     | 8     | 10    |
> |-------------------|-------|-------|-------|-------|-------|-------|
> | HGNN              | 91.00 | 90.42 | 89.81 | 88.93 | 88.32 | 87.30 |
> | TF-HNN            | 91.38 | 90.28 | 89.96 | 89.44 | 89.03 | 88.57 |
> | AllSetTransformer | 91.51 | 90.95 | 90.34 | 89.48 | 88.31 | 87.29 |
> | ED-HNN            | 91.55 | 91.09 | 90.72 | 89.98 | 89.43 | 88.84 |
> | EHNN              | 90.47 | 90.47 | 90.44 | 90.48 | 90.50 | 90.51 |
>
> **Table R10:** Hyperedge size sensitivity analysis on Actor.
>
> | Method             | 0     | 2     | 4     | 6     | 8     | 10    |
> |-------------------|-------|-------|-------|-------|-------|-------|
> | HGNN              | 77.83 | 77.91 | 77.94 | 77.93 | 77.90 | 77.72 |
> | TF-HNN            | 85.96 | 85.96 | 85.68 | 85.22 | 85.57 | 85.61 |
> | AllSetTransformer | 85.66 | 85.69 | 85.68 | 85.63 | 85.84 | 85.70 |
> | ED-HNN            | 85.77 | 85.79 | 85.80 | 85.76 | 85.74 | 85.77 |
> | EHNN              | 86.21 | 86.05 | 86.19 | 86.18 | 86.07 | 85.93 |
>
>
> **Table R11:** Hyperedge size sensitivity analysis on Pokec.
>
> | Method             | 0     | 2     | 4     | 6     | 8     | 10    |
> |-------------------|-------|-------|-------|-------|-------|-------|
> | HGNN              | 57.87 | 58.98 | 58.57 | 57.87 | 57.11 | 57.78 |
> | TF-HNN            | 59.17 | 59.13 | 59.18 | 59.06 | 59.07 | 59.22 |
> | AllSetTransformer | 58.55 | 58.90 | 58.76 | 59.02 | 58.67 | 58.74 |
> | ED-HNN            | 58.68 | 58.71 | 59.05 | 58.82 | 58.91 | 58.98 |
> | EHNN              | 58.23 | 58.23 | 58.20 | 58.06 | 58.11 | 58.02 |
>
>
> * **Findings.** Based on the empirical results reported in Tables R8–R11, we derive the following key observations:
>
>     - On homophilic datasets, introducing only a few extremely large hyperedges while keeping the rest small consistently degrades model performance. As the proportion of these super-large hyperedges increases, performance generally continues to decline. This is likely because a small number of oversized hyperedges disrupt fine-grained local structure, causing the models to lose the class-consistent neighborhood signals that homophilic settings rely on.
>
>     - On heterophilic datasets, increasing the proportion of super-large hyperedges generally maintains stable performance and may even yield slight improvements. A plausible explanation is that, in heterophilic settings, the presence of a small number of oversized hyperedges further weakens the influence of the original heterophilic connections during message passing, thereby reducing the impact of noisy or label-inconsistent neighbors.
>
>     - Among all evaluated architectures, the tensor-based EHNN demonstrates the strongest robustness to extreme hyperedge-size skew: its performance remains stable across all constructed settings on both homophilic and heterophilic datasets.
>
> **We have updated the manuscript and incorporated the above results and analysis into Appendix C.9.** These findings offer meaningful guidance for understanding how HNNs behave under the long-tailed hyperedge size distributions commonly seen in real-world hypergraph data, thereby making our benchmark evaluation more comprehensive and insightful. Thank you for your valuable feedback!

---

> ### Author Response · Authors · 2025-11-25
> **Response to Question #2**
>
> > **Question #2:** How do different models handle nodes with very high vs. low degrees?
>
> **A:** Thank you for your insightful question! In response, we conduct experiments to compare the behavior of different HNNs on nodes with very high versus very low degrees.
>
> * **Settings.** To investigate this question, we design an experiment that explicitly contrasts model behavior on nodes with substantially different degrees. Specifically, we define **very high–degree** nodes as those whose degrees fall within the top-$p$% of the dataset, and **very low–degree** nodes as those in the bottom-$p$%. To ensure robustness, we consider two thresholds, $p=1$ and $p=5$. Our study evaluates 8 representative HNN architectures spanning three major categories across four benchmark datasets. For each setting, we report the classification accuracy separately on the very high–degree and very low–degree subsets of the test nodes.
>
> * **Results.**
>
> **Table R12:** Comparison of high-degree vs. low-degree node performance on Cora.
>
> | Degree  | HGNN | PhenomNN | SheafHyperGNN | TF-HNN | UniGNN | AllSetTransformer | ED-HNN | EHNN |
> | :----: |  :----: |  :----: |  :----: |  :----: |  :----: | :----: | :----: | :----: |
> | Very High (1%)| 85.58 | 85.59 | 85.48 | 85.55 | 86.22 | 85.36 |85.36|83.58|
> | Very Low (1%)| 72.05 | 73.47 | 74.38 | 75.11 | 74.12 | 73.25 |73.21|69.93|
> | Very High (5%) | 81.82 | 80.74 | 82.86 | 81.91 | 81.15 | 81.37 | 81.02 | 81.34|
> | Very Low (5%) | 72.05 | 73.47  |  74.38  | 75.11 | 74.17 | 73.19 | 73.21 |70.12 |
>
> **Table R13:** Comparison of high-degree vs. low-degree node performance on DBLP-CA.
>
> | Degree  | HGNN | PhenomNN | SheafHyperGNN | TF-HNN | UniGNN | AllSetTransformer | ED-HNN | EHNN |
> | :----: |  :----: |  :----: |  :----: |  :----: |  :----: | :----: | :----: | :----: |
> | Very High (1%)| 93.61 | 93.97 | 94.16 | 94.42 | 94.22 | 95.37 |94.73|95.24|
> | Very Low (1%)| 88.04 | 89.70 | 88.22 | 87.44  | 89.49 | 89.12 |88.88|86.73|
> | Very High (5%) | 93.38 |93.69  | 93.48 | 93.78 | 93.58| 93.85 |93.88|93.95|
> | Very Low (5%)| 88.13 |89.64 | 88.22 | 87.45 | 89.48 | 89.17 |88.89|86.63|
>
> **Table R14:** Comparison of high-degree vs. low-degree node performance on Actor.
>
> | Degree  | HGNN | PhenomNN | SheafHyperGNN | TF-HNN | UniGNN | AllSetTransformer | ED-HNN | EHNN |
> | :----: |  :----: |  :----: |  :----: |  :----: |  :----: | :----: | :----: | :----: |
> | Very High (1%) | 73.37 | 93.18 | 82.42 | 93.71 | 93.73 | 94.18 |95.42|95.84|
> | Very Low (1%) | 64.93 | 66.51 | 74.10 | 73.97 | 65.35 | 72.55 |70.80|75.66|
> | Very High (5%) | 76.93 | 90.24 | 87.87 | 90.88| 90.45 | 91.52 |91.71|91.81|
> | Very Low (5%) | 76.39 | 80.22 | 83.08 | 83.97 | 79.05 | 83.23 |82.99|83.49|
>
> **Table R15:** Comparison of high-degree vs. low-degree node performance on Pokec.
>
> | Degree  | HGNN | PhenomNN | SheafHyperGNN | TF-HNN | UniGNN | AllSetTransformer | ED-HNN | EHNN |
> | :----: |  :----: |  :----: |  :----: |  :----: |  :----: | :----: | :----: | :----: |
> | Very High (1%) | 59.96 | 64.60 | 64.29 | 67.03| 67.34 | 68.24 |67.63|63.99|
> | Very Low (1%) | 58.23 | 57.42 | 58.41 | 57.80 | 57.61 | 57.11 |57.22|57.62|
> | Very High (5%) | 58.39 | 64.74 | 63.58 | 65.87 | 66.01 | 66.78 | 65.77 | 63.06 |
> | Very Low (5%)| 57.80 | 57.98  | 58.43 | 57.80 | 57.63 | 57.29 | 57.18 | 57.62 |
>
> * **Findings.** From the results summarized in Tables R12–R15, we draw two key observations:
>
>     - Across all datasets and all HNN architectures, we consistently observe a structural unfairness phenomenon: **models achieve substantially higher accuracy on very high-degree nodes compared to very low-degree nodes.** A plausible explanation is that high-degree nodes benefit more from message passing because they can aggregate richer and more reliable higher-order structural information, whereas low-degree nodes struggle to leverage structural signals and are more vulnerable to noise introduced by sparse or unreliable neighbors.
>     - **The performance disparity becomes more pronounced under stricter degree thresholds.** When the threshold is reduced from 5% to 1%, the gap between very high-degree and very low-degree nodes typically increases substantially. This suggests that the most extreme-degree nodes exhibit the strongest disparity, further underscoring the critical role of degree heterogeneity in shaping HNN behavior.
>
> These analyses provide a clearer understanding of how HNNs behave across extreme degree levels and reveal that **improving the performance of low-degree nodes remains a key bottleneck in advancing HNN models**. This highlights an important direction for future work: designing mechanisms that better enable low-degree nodes to exploit structural information.
>
> **We have incorporated the above experimental results and analyses into Appendix C.10 of the revised paper.** Thank you again for your constructive feedback!

---

> ### Author Response · Authors · 2025-11-25
> **Response to Question #3**
>
> > **Question #3:** Given the results, what is the recommended decision-making process for a practitioner when selecting an HNN model for a new task?
>
> **A:** Thank you for your constructive question. Drawing on the comprehensive benchmarking results and analyses presented in this work, we offer practical guidance for selecting appropriate HNN models for new tasks. For clarity, we organize our recommendations by task type.
>
> * **Node-level prediction tasks.** We recommend **TF-HNN as the first-choice model**.
> Across a wide range of datasets, TF-HNN **consistently achieves top-ranked node classification performance**, demonstrating its strong ability to learn highly discriminative node representations. Moreover, its training-free message-passing architecture offers **substantial efficiency and scalability benefits**, making it well-suited for large-scale or resource-constrained applications.
> Importantly, our experiments show that, compared with other HNNs, **TF-HNN does not exhibit pronounced weaknesses in robustness or fairness**, making it a reliable choice for most node-level scenarios.
>
> * **Edge-level or higher-order relation prediction tasks (e.g., hyperlink prediction, hyperedge prediction).**
> **We suggest starting with EHNN, HGNN, and HyperGCN.** Together, these models account for most of the best and second-best results on hyperedge prediction benchmarks. Their performance, however, varies across homophilic and heterophilic settings: on homophilic datasets, **EHNN and HyperGCN** generally perform better; on heterophilic datasets, **HGNN and EHNN** tend to yield stronger results.
> Our robustness analysis further indicates that **HGNN is more sensitive to structural perturbations**, and may therefore be less dependable under distribution shifts or noisy hypergraph structures. As a result, **EHNN and HyperGCN are generally safer and more robust defaults**, while HGNN should be chosen with awareness of dataset stability.
>
> * **Graph-level prediction tasks.**  No single architecture consistently outperforms all others across datasets and evaluation metrics in hypergraph classification. Nonetheless, **HJRL, DPHGNN, and AllSetTransformer frequently appear among the top-performing models**, reflecting their strong ability to capture and discriminate global structural patterns that drive hypergraph-level prediction. However, our robustness experiments reveal that **DPHGNN can be sensitive to structural and feature perturbations**, and practitioners are therefore advised to carefully assess its stability before deployment.
> Among these models, **AllSetTransformer often provides a more favorable utility–efficiency trade-off**, making it particularly appealing in computationally constrained environments.
>
> **The above guidance for practitioners has been incorporated into Section 5 of the revised manuscript.** We believe that this discussion will meaningfully support real-world decision-making by helping practitioners identify the most suitable HNN models for their applications. We sincerely appreciate your thoughtful comments!

---

> ### Author Response · Authors · 2025-11-25
> **Response to Question #4**
>
> > **Question #4:** What are the key trade-offs, supported by the benchmark's findings, that one should consider between performance, scalability, and specific data characteristics (e.g., homophily)?
>
> **A:** Thank you for the insightful question. We fully recognize that discussing the trade-offs among performance, scalability, and data characteristics (e.g., homophily) is essential, as such analysis provides practical guidance for choosing appropriate HNN models under different tasks, datasets, and deployment constraints.
>
> **For spectral-based models**, most advanced approaches (e.g., PhenomNN, SheafHyperGNN, HJRL) consistently outperform earlier variants such as HGNN, HyperGCN, and HCHA on homophilous datasets. However, this accuracy gain relies on more complex expansion mechanisms and Laplacian operators, which substantially reduce scalability. As shown in Table 1, they frequently encounter OOM issues on large or dense hypergraphs such as Trivago and Yelp. TF-HNN provides a lightweight alternative, achieving top-ranked performance on most datasets while maintaining strong scalability due to its training-free message-passing design. **Spatial-based architectures** generally offer a more favorable scalability–performance balance. Models such as UniGNN, AllSetTransformer, and ED-HNN deliver accuracy comparable to advanced spectral methods on homophilous data with substantially lower memory consumption. **Tensor-based methods (e.g., EHNN and T-HyperGNN)** perform worse on homophilous datasets, but relative to spectral- and spatial-based HNNs, they often achieve better performance on heterophilous benchmarks, particularly EHNN, which also demonstrates stronger scalability than T-HyperGNN. Although **MLPs** perform substantially worse than HNNs on homophilous datasets, they often excel on heterophilous benchmarks and outperform many HNN architectures. Furthermore, by removing high-order message passing, MLPs achieve markedly better scalability.
>
> **We have incorporated the above discussion into Appendix D.2 of the revised manuscript.** We sincerely appreciate your thoughtful and constructive feedback.

---

> ### Author Response · Authors · 2025-11-25
> **Conclusion**
>
> Thank you again for your time and valuable insights, and hope our responses can address all your concerns! We are excited about the impact DHG-Bench may have on the community and are grateful for your support in advancing this work.
>
> **As the current ratings reveal somewhat negative evaluations of our work, we would greatly appreciate it if you could kindly consider raising the ratings to show your support for our work. Thank you!**
>
> ---
>
> **References**
>
> **[1]** Patil, P., Sharma, G., & Murty, M. N. (2020, May). Negative sampling for hyperlink prediction in networks. In Pacific-Asia conference on knowledge discovery and data mining (pp. 607-619). Cham: Springer International Publishing.
>
> **[2]** Hwang, H., Lee, S., Park, C., & Shin, K. (2022, July). Ahp: Learning to negative sample for hyperedge prediction. In Proceedings of the 45th international ACM SIGIR conference on research and development in information retrieval (pp. 2237-2242).
>
> **[3]** Ko, Y., Tong, H., & Kim, S. W. (2025). Enhancing hyperedge prediction with context-aware self-supervised learning. IEEE Transactions on Knowledge and Data Engineering.
>
> **[4]** Benson, A. R., Abebe, R., Schaub, M. T., Jadbabaie, A., & Kleinberg, J. (2018). Simplicial closure and higher-order link prediction. Proceedings of the National Academy of Sciences, 115(48), E11221-E11230.
>
> **[5]** Feng, Y., You, H., Zhang, Z., Ji, R., & Gao, Y. (2019, July). Hypergraph neural networks. In Proceedings of the AAAI conference on artificial intelligence (Vol. 33, No. 01, pp. 3558-3565).
>
> **[6]** Wang, P., Yang, S., Liu, Y., Wang, Z., & Li, P. Equivariant Hypergraph Diffusion Neural Operators. In The Eleventh International Conference on Learning Representations.
>
> **[7]** Tang, B., Liu, Z., Jiang, K., Chen, S., & Dong, X. Training-Free Message Passing for Learning on Hypergraphs. In The Thirteenth International Conference on Learning Representations.
>
> **[8]** Lee, D., & Shin, K. (2023, June). I’m me, we’re us, and i’m us: Tri-directional contrastive learning on hypergraphs. In Proceedings of the AAAI conference on artificial intelligence (Vol. 37, No. 7, pp. 8456-8464).
>
> **[9]** Li, F., Wang, X., Cheng, D., Zhang, W., Zhang, Y., & Lin, X. (2024, August). Hypergraph self-supervised learning with sampling-efficient signals. In Proceedings of the Thirty-Third International Joint Conference on Artificial Intelligence (pp. 4398-4406).
>
> **[10]** Kim, S., Kang, S., Bu, F., Lee, S. Y., Yoo, J., & Shin, K. HypeBoy: Generative Self-Supervised Representation Learning on Hypergraphs. In The Twelfth International Conference on Learning Representations.

---

### Official Review · Reviewer_CWSt · 2025-10-26

**Soundness:** 3
**Presentation:** 3
**Contribution:** 3
**Rating:** 6
**Confidence:** 4

**Summary:**

The authors propose a comprehensive hypergraph neural network (HNN) benchmark, which is called DHG-Bench.

The benchmark implementation incorporates most of the representative HNNs and datasets.

Moreover, the authors evaluate the HNNs under diverse scenarios (e.g., classification, hyperedge prediction, and noisy cases).

**Strengths:**

S1. The authors provided a timely benchmark for hypergraph neural networks.

S2. The benchmark is comprehensive, in terms of both HNNs and downstream tasks.

S3. Most of the benchmark hypergraph datasets have been covered.

**Weaknesses:**

I do not have major criticisms of this work, but I have several suggestions.

**W1. [pip installation]** For now, I think one needs to download the GitHub repo to run the code. I think authors can improve the code to be easier to use, as in PyG (https://pytorch-geometric.readthedocs.io/en/2.4.0/install/installation.html).

**W2. [Label split]** While many HNNs use a 50/25/25 split for node classification, I personally think this ratio contains too many training nodes, compared to the common graph evaluation settings. Can the authors analyze the HNNs in more label-scarce (i.e., less training nodes) scenarios?

**Questions:**

See Weakness

---

> ### Author Response · Authors · 2025-11-25
> **Overall Response to Reviewer CWSt**
>
> Thank you for your thoughtful and encouraging feedback on our paper. We sincerely appreciate your recognition of the timeliness of DHG-Bench, as well as its comprehensive coverage of both HNN models and downstream tasks. We are also glad that you acknowledge the diversity of hypergraph datasets included in our benchmark.
>
> We respond to your concerns and questions as follows.
>
> **[Preview]**
>
> * **Weakness #1:** Improve the installation of code for easier use.
> * **Weakness #2:** Lack label-scarce scenario.
> * **Conclusion.**

---

> ### Author Response · Authors · 2025-11-25
> **Response to Weakness #1**
>
> > **Weakness #1:** **[pip installation]** For now, I think one needs to download the GitHub repo to run the code. I think authors can improve the code to be easier to use, as in PyG
>
> **A:** Thank you very much for this helpful suggestion. **We have improved the entire codebase and now fully support pip installation for the package.** Users no longer need to download the GitHub repository manually, and the code library can be directly installed via:
> ```nginx
> pip install dhg-bench
> ```
> **We have also updated the README.md file in the GitHub repository referenced in the revised paper.** to provide clear instructions on how to quickly run experiments and tune hyperparameters using the imported library. In addition, all model-specific hyperparameter configuration files are included in the repository to further ensure reproducibility.
>
> We sincerely appreciate your suggestion, which will significantly improve the usability of our work and further broaden its impact within the HNN research community.

---

> ### Author Response · Authors · 2025-11-25
> **Response to Weakness #2**
>
> > **Weakness #2:** [Label split] While many HNNs use a 50/25/25 split for node classification, I personally think this ratio contains too many training nodes, compared to the common graph evaluation settings. Can the authors analyze the HNNs in more label-scarce (i.e., less training nodes) scenarios?
>
> **A:** Thank you for your valuable suggestions. We agree that analyzing the HNNs in more label-scarce scenarios could provide additional insights into the effectiveness of the HNN algorithms, particularly in understanding their applicability in real-world settings where labeled data is limited. In response, **we conducted additional experiments under more label-scarce settings.**
>
> * **Settings.** We first split the node labels into 20%/20%/60% for the train/validation/test sets. The validation and test sets are then kept fixed, and different levels of label scarcity are simulated by masking a portion of the training labels. Specifically, we adjust the masking ratio so that the visible training labels constitute 20%, 15%, 10%, 5%, and 1% of all nodes. This design allows us to systematically examine how HNNs behave as labeled data becomes increasingly limited. We evaluate 8 representative HNN algorithms spanning three major categories (spectral-based, spatial-based, and tensor-based) on the Cora and Actor datasets, and report model performance in terms of accuracy.
>
> * **Results.**
>
> **Table A9 (New)**: Label-scarce node classification on Cora.
>
> | Method        | 20%   | 15%   | 10%   | 5%    | 1%    |
> | :-----------: | :---: | :---: | :---: | :---: | :---: |
> | HGNN          | 74.84 | 73.24 | 70.09 | 64.75 | 42.30 |
> | PhenomNN      | 75.35 | 74.07 | 71.96 | 67.55 | 44.96 |
> | SheafHyperGCN | 76.06 | 74.66 | 71.29 | 66.37 | 43.67 |
> | TF-HNN         | 76.31 | 75.07 | 71.77 | 64.48 | 39.29 |
> | UniGNN        | 76.08 | 74.04 | 70.89 | 64.85 | 43.18 |
> | AllSetTransformer  | 73.48 | 72.33 | 68.46 | 61.70 | 40.44 |
> | ED-HNN         | 74.20 | 72.41 | 69.93 | 63.65 | 42.79 |
> | T-HyperGNN    | 69.02 | 66.99 | 62.50 | 52.89 | 36.60 |
>
>
> **Table A10 (New)**: Label-scarce node classification on Actor.
>
> | Method        | 20%   | 15%   | 10%   | 5%    | 1%    |
> | :-----------: | :---: | :---: | :---: | :---: | :---: |
> | HGNN          | 77.90 | 77.79 | 77.52 | 76.39 | 70.12 |
> | PhenomNN      | 82.99 | 82.89 | 82.56 | 81.77 | 76.89 |
> | SheafHyperGCN | 84.16 | 83.67 | 82.95 | 80.48 | 72.27 |
> | TF-HNN         | 85.34 | 84.74 | 83.95 | 81.74 | 74.88 |
> | UniGNN        | 82.85 | 82.65 | 82.64 | 81.69 | 76.34 |
> | AllSetTransformer  | 84.06 | 83.46 | 82.25 | 79.55 | 75.47 |
> | ED-HNN         | 84.74 | 84.30 | 83.38 | 81.14 | 75.53 |
> | T-HyperGNN    | 84.87 | 84.39 | 83.48 | 81.29 | 73.63 |
>
> * **Findings.** Based on results in Table A9 and A10, we have the following key observations:
>
>   - As label scarcity increases, all HNN models exhibit a clear degradation in performance, with the decline becoming more significant under extremely low-label settings; notably, all methods experience substantial drops when the labeled ratio decreases from 5% to 1%.
>   - Across both datasets, PhenomNN consistently shows the strongest robustness under highly label-scarce conditions (1% and 5%). In contrast, TF-HNN, although it achieves SOTA performance in label-abundant scenarios (see Table 1 in the original manuscript), suffers a severe accuracy collapse when supervision is limited and ranks as the second-worst method on Cora at the 1% label ratio.
>   - The performance degradation on the homophilous Cora dataset is more pronounced than on the heterophilous Actor dataset. This may be because heterophilous links introduce misleading feature mixing, which reduces the usefulness of label information during training and makes Actor less sensitive to label scarcity.
>
> **We have included the above results and analysis into Appendix C.4 in the revised manuscript.**
> We believe this additional analysis enriches our study and offers a clearer picture of HNN algorithm's effectiveness. Thank you again for your constructive suggestion!

---

> ### Author Response · Authors · 2025-11-25
> **Conclusion**
>
> Thank you again for your time and valuable insights, and hope our responses can address all your concerns! We are excited about the impact DHG-Bench may have on the community and are grateful for your support in advancing this work.
>
> **We would be even encouraging if you could kindly consider raising the ratings!**

---

> ### Comment · Reviewer_CWSt · 2025-11-27
>
> Thank you for the detailed responses.
>
> My primary concern was about the inconvenience of installation, and since this has been resolved by pip-based approach, I would like to raise my score from 6 to 8.

---

> > ### Author Response · Authors · 2025-11-27
> >
> > Thank you very much for your positive feedback! We sincerely appreciate your suggestions to improve installation convenience and to include empirical analysis under label-scarce scenarios. We are pleased that our responses have addressed your concerns, and your support and encouraging evaluation mean a great deal to us.

---

> ### Comment · Reviewer_CWSt · 2025-11-27
>
> Dear Authors,
>
> I have an additional question regarding the broader impact of DHG-Bench beyond benchmarking common hypergraph downstream tasks. Do you have any plans to extend its application to other domains, such as computer vision or bioinformatics?
>
> To elaborate, I have often heard that practitioners hesitate to adopt hypergraphs due to implementation complexity and challenges in integrating them with other ML models. I believe that DHG-Bench has the potential to help address these obstacles.
>
> In this context, I am curious whether you have any plans or ideas for further improving its applicability to such domains. You do not need to provide implementation details; a high-level plan or discussion would be greatly appreciated.
>
> Best regards,

---

> > ### Author Response · Authors · 2025-11-27
> >
> > Thank you very much for your insightful question. **Expanding the broader impact of DHG-Bench and making hypergraph learning more accessible across different application domains is indeed one of our long-term goals.** We fully agree that lowering the barriers to adopting hypergraphs can greatly benefit practitioners in fields such as computer vision and bioinformatics.
> >
> > We do have plans to extend DHG-Bench to broader domains, and **our current roadmap primarily includes three directions**:
> >
> > * **Incorporating domain-adapted hypergraph construction methods.**
> > For many real-world datasets, the initial hypergraph modeling step is crucial for uncovering meaningful higher-order interactions. We plan to introduce widely used construction strategies tailored to specific domains. For example, Fuzzy C-Means–based hypergraph construction methods [1] and dynamic hypergraph construction approaches [2] for images in computer vision, as well as sample–feature incidence modeling [3] for genomic data in bioinformatics. These tools will allow researchers to build hypergraphs more easily and reduce modeling complexity.
> >
> > * **Integrating domain-specific hypergraph learning algorithms.**
> > Algorithms designed with domain priors often outperform general-purpose HNNs on specialized downstream tasks. We therefore plan to include models developed for specific domains, such as Vision HGNN [1] and HgVT [2] for image classification and retrieval, as well as DISHyper [4] and DGHNN [5] for gene-related prediction tasks.
> >
> > * **Supporting training pipelines for domain-specific downstream tasks.**
> > Applications in other domains may require training objectives beyond the three mainstream tasks currently evaluated in DHG-Bench. As such, we aim to support task-specific loss functions and training paradigms. Examples include list-wise loss [6] and triplet-based ranking objectives [7], which are frequently used in image retrieval.
> >
> > We sincerely appreciate your thoughtful question and your commitment to advancing the research community. We will continue to extend DHG-Bench to a broader range of application domains moving forward. If you have any further questions, please feel free to contact us. Your suggestions have been extremely valuable in helping us improve our work.
> >
> > ---
> >
> > **References**
> >
> > **[1]** Han, Y., Wang, P., Kundu, S., Ding, Y., & Wang, Z. (2023). Vision hgnn: An image is more than a graph of nodes. In Proceedings of the IEEE/CVF International Conference on Computer Vision (pp. 19878-19888).
> >
> > **[2]** Fixelle, J. (2025). Hypergraph Vision Transformers: Images are More than Nodes, More than Edges. In Proceedings of the Computer Vision and Pattern Recognition Conference (pp. 9751-9761).
> >
> > **[3]** Tian, Z., Hwang, T., & Kuang, R. (2009). A hypergraph-based learning algorithm for classifying gene expression and arrayCGH data with prior knowledge. Bioinformatics, 25(21), 2831-2838.
> >
> > **[4]** Deng, C., Li, H. D., Zhang, L. S., Liu, Y., Li, Y., & Wang, J. (2024). Identifying new cancer genes based on the integration of annotated gene sets via hypergraph neural networks. Bioinformatics, 40(Supplement_1), i511-i520.
> >
> > **[5]** Li, B., Xiao, X., Zhang, C., Xiao, M., & Zhang, L. (2025). DGHNN: A deep graph and hypergraph neural network for pan-cancer related gene prediction. Bioinformatics, btaf379.
> >
> > **[6]** Revaud, J., Almazán, J., Rezende, R. S., & Souza, C. R. D. (2019). Learning with average precision: Training image retrieval with a listwise loss. In Proceedings of the IEEE/CVF International Conference on Computer Vision (pp. 5107-5116).
> >
> > **[7]** Gordo, A., Almazan, J., Revaud, J., & Larlus, D. (2017). End-to-end learning of deep visual representations for image retrieval. International Journal of Computer Vision, 124(2), 237-254.

---

> > > ### Comment · Reviewer_CWSt · 2025-11-27
> > >
> > > Thank you for your detailed responses!
> > >
> > > It seems the proposed bench would be useful for applying hypergraph to a wide range of applications, which I think is a significant contribution in the field.

---

### Official Review · Reviewer_KSgV · 2025-10-29

**Soundness:** 2
**Presentation:** 3
**Contribution:** 2
**Rating:** 4
**Confidence:** 4

**Summary:**

This paper presents DHG-Bench, a large-scale benchmark for Hypergraph Neural Networks (HNNs). The authors implement 17 existing methods and evaluate them on 22 datasets across node-, hyperedge-, and graph-level classification tasks. The evaluation covers four axes: effectiveness (accuracy), efficiency (runtime and memory), robustness (to structural, feature, and supervision perturbations), and fairness ($\Delta$DP and $\Delta$EO metrics). The authors also release an open-source library to ensure reproducibility. Overall, this is an ambitious and useful engineering effort that aims to unify evaluation practices in hypergraph learning.

**Strengths:**

1. **Comprehensive benchmark coverage.** DHG-Bench implements 17 HNN models and evaluates them on 22 datasets spanning node-, hyperedge-, and graph-level tasks.
2. **Multi-dimensional evaluation.** The benchmark goes beyond accuracy to assess efficiency, robustness, and fairness, providing a more holistic view of model behavior.
3. **Reproducibility focus.** The open-source code and datasets enable other researchers to replicate results and extend the benchmark.
4. **Insightful findings.** The experiments reveal important phenomena such as scalability bottlenecks, fairness gaps, and underperformance of HNNs on heterophilic datasets.

**Weaknesses:**

1. **Limited conceptual novelty.** The benchmark aggregates existing models but does not introduce new methodologies or theoretical advances.
2. **Insufficient graph-based baselines.** Only two GNNs are included, and simple but strong baselines (e.g., direction-aware GNNs) are missing, making it difficult to quantify the advantage of hypergraphs.
3. **Directed hypergraphs ignored.** The benchmark only evaluates undirected hypergraphs, omitting variants that model asymmetric or causal relations, i.e. directed hypergraphs.
4. **Superficial scalability analysis.** Many methods fail with OOM errors, but mitigation strategies (e.g., mini-batching, sparse operations) are not reported.
5. **Shallow heterophily treatment.** HNNs underperform MLPs on heterophilic datasets, but causes such as oversmoothing or feature mixing are not analyzed in detail.

**Questions:**

1. **Baseline selection.** What criteria determined the included baselines? Why were simple but strong baselines (MLPs, direction-aware GNNs) not systematically included?
2. **Directed hypergraphs and variants.** Do you plan to extend DHG-Bench to support directed hypergraphs, heterogeneous hyperedges, or temporal hypergraphs? If not, please justify.
3. **OOM diagnostics.** For models that ran out of memory, which mitigation strategies were attempted (e.g., mini-batching, sparse matrices, mixed precision)? Can you report peak memory usage?
4. **Heterophily analysis.** Can you quantify heterophily per dataset, test hypotheses (e.g., oversmoothing, feature collapse), and relate results to prior literature on heterophilic GNN behavior?
5. **Fairness metrics.** For datasets lacking explicit sensitive attributes, how were $\Delta$DP/ta$EO computed? Were proxy attributes used, and how were they validated?

---

> ### Author Response · Authors · 2025-11-25
> **Overall Response to Reviewer KSgV**
>
> We sincerely thank the reviewers for recognizing the strengths of our work. We are delighted that you found DHG-Bench to be a well-constructed and valuable benchmark with comprehensive coverage across 17 HNN models, 22 datasets, and multiple task levels. Your acknowledgment of our efforts to establish a consistent and multi-dimensional evaluation framework is greatly appreciated. We are also grateful for the positive comments on the reproducibility and the insights offered by our benchmark.
>
> We respond to your concerns and questions as follows.
>
> **[Preview]**
>
> * **Weakness #1:** Limited conceptual novelty.
> * **Weakness #2 & Question #1:** Insufficient graph-based baselines.
> * **Weakness #3:** Directed hypergraphs ignored.
> * **Question #2:** Future plan for directed hypergraphs and variants.
> * **Weakness #4 & Question #3:** Superficial scalability analysis.
> * **Weakness #5 & Question #4:** Shallow heterophily treatment.
> * **Question #5:** Fairness metrics.
> * **Conclusion.**
> ---

---

> ### Author Response · Authors · 2025-11-25
> **Response to Weakness #1**
>
> > **Weakness #1:** Limited conceptual novelty. The benchmark aggregates existing models but does not introduce new methodologies or theoretical advances.
>
> **A:** We appreciate the reviewer's comment. The primary area of our paper is "datasets and benchmarks", where the main objective is to establish standardized and reproducible evaluation foundations for the field and to provide a comprehensive empirical understanding and analysis.
> The core contributions of our work are proposing **the first comprehensive benchmark for deep hypergraph learning**, which does not necessarily require proposing new algorithms or theoretical frameworks. Specifically, our contributions are as follows:
>
> * **Comprehensive Benchmark for Deep Hypergraph Learning.**
> We present the first comprehensive benchmark for deep hypergraph learning, which integrates **17 state-of-the-art HNN methods** and **22 widely used hypergraph datasets**, covering **node-level, edge-level, and graph-level tasks**. This unified and extensive setup enables a thorough assessment of algorithmic performance across diverse learning scenarios.
>
> * **Standardized Experiment Conduction Protocol.** Another key contribution of our work is the establishment of a consistent experimental protocol. The lack of **standardization in experimental setups**, including parameter configurations, initialization procedures, and dataset splitting strategies, has long been a major barrier to reproducibility in HNN studies. Our benchmark provides a comprehensive and unified protocol that standardizes these components, enabling more reliable and comparable results across different methods.
>
> * **Multi-dimensional Empirical Evaluation and Analysis.** While **effectiveness** is often the primary focus in existing HNN studies, we argue that understanding the **efficiency, robustness, and fairness** of HNN algorithms is equally crucial for guiding their real-world deployment. Our work introduces a more comprehensive evaluation standard that incorporates these dimensions, enabling HNN methods to be assessed not only in terms of predictive performance but also in terms of their practical feasibility and reliability.
>
> * **Open-sourced and Extensible Benchmark Library.** We release DHG-Bench as an easy-to-use and extensible library to support and accelerate future HNN research. The toolkit provides **standardized data loaders and algorithm implementations, unified training pipelines, and fully reproducible experimental configurations**. This practical contribution offers long-term value to the community and lays a solid foundation for subsequent methodological advances.
>
> We hope this clarifies the scope and importance of our contributions, and we will make sure to emphasize these points more clearly in the revised manuscript. Thank you again for your feedback!

---

> ### Author Response · Authors · 2025-11-25
> **Response to Weakness #2 & Question #1 (Part I)**
>
> > **Weakness #2 & Question #1:** What criteria determined the included baselines? Why were simple but strong baselines (MLPs, direction-aware GNNs) not systematically included?
>
> **A:** Thank you for your questions. We would like to address these concerns in detail.
>
> We want to clarify that **our selection of baselines follows prior HNN studies [1, 2, 3]**, which commonly include MLP and two representative clique-expansion-based GNNs, CEGCN and CEGAT, as comparison models.
>
> We also note that **MLP is already included in our experiments, as mentioned in Appendix A.2 of the original manuscript**. Regarding direction-aware GNNs, we do not include them because our benchmark focuses on undirected hypergraphs. After clique expansion, **the resulting graph remains undirected**, leading to symmetric message propagation. Under this setting, **direction-aware GNNs cannot fully demonstrate their strengths in modeling directional information**, and therefore they are not well aligned with the scope of our study.
>
> Following your suggestions, **we include two widely used direction-aware GNNs, MagNet [4] and DirGNN [5]**, as additional baselines and compare it with representative HNNs. We evaluate both models on node-level, edge-level, and graph-level tasks, **with the corresponding results reported in Table R1, Table R2, and Table R3, respectively**.
>
> **Table R1**: Node classification performance with two newly added direction-aware GNNs.
>
> | Method  | Cora | Pubmed | DBLP-CA | Warmart | Actor | Pokec|
> | :----: |  :----: |  :----: |  :----: |  :----: |  :----: | :----: |
> | MLP | 75.33 $\pm$ 0.88 | 86.62 $\pm$ 0.26 | 85.54 $\pm$ 0.15 | 63.21 $\pm$ 0.12 | 86.06 $\pm$ 0.36 | 59.64 $\pm$ 0.48 |
> | **MagNet** | **77.10 $\pm$ 1.35** | **86.12 $\pm$ 0.16** | **89.99 $\pm$ 0.31** | **71.81 $\pm$ 0.27** | **67.62 $\pm$ 0.56**|**57.01 $\pm$ 0.69**|
> | **DirGNN** | **78.17 $\pm$ 0.81** | **86.50 $\pm$ 0.46** | **90.75 $\pm$ 0.28** | **73.78 $\pm$ 0.09** | **84.92 $\pm$ 0.49** |**58.47 $\pm$ 0.87**|
> | HGNN | 77.90 $\pm$ 1.17 | 86.17 $\pm$ 0.52 | 91.00 $\pm$ 0.27 | 77.12 $\pm$ 0.12 | 77.83 $\pm$ 0.37 | 57.87 $\pm$ 0.76 |
> | HyperGCN | 78.38 $\pm$ 1.63 | 87.42 $\pm$ 0.42 | 89.51 $\pm$ 0.18 | 68.75 $\pm$ 0.56 | 81.82 $\pm$ 0.39 | 57.51 $\pm$ 0.54 |
> | PhenomNN | 78.97 $\pm$ 1.41 | 87.81 $\pm$ 0.12 | 91.83 $\pm$ 0.25 | OOM | 83.14 $\pm$ 0.49 | 58.43 $\pm$ 0.92 |
> | SheafHyperGNN | 79.03 $\pm$ 0.90 | 87.10 $\pm$ 0.47 | 91.09 $\pm$ 0.31 | OOM | 85.00 $\pm$ 0.32 | 59.06 $\pm$ 0.37 |
> | TF-HNN | 79.47 $\pm$ 1.31 | 87.90 $\pm$ 0.37 | 91.38 $\pm$ 0.24 | 77.04 $\pm$ 0.12 | 85.96 $\pm$ 0.41 | 59.17 $\pm$ 0.52 |
> | UniGNN | 79.41 $\pm$ 1.24 | 87.57 $\pm$ 0.54 | 91.71 $\pm$ 0.20 | 76.26 $\pm$ 0.58 | 84.61 $\pm$ 0.46 | 58.56 $\pm$ 0.73 |
> | AllSetTransformer | 78.02 $\pm$ 1.43 | 87.79 $\pm$ 0.30 | 91.51 $\pm$ 0.22 | 78.61 $\pm$ 0.13 | 85.66 $\pm$ 0.41 | 58.55 $\pm$ 0.56 |
> | ED-HNN | 78.58 $\pm$ 0.52 | 87.65 $\pm$ 0.23 | 91.55 $\pm$ 0.19 | 77.90 $\pm$ 0.21 | 85.77 $\pm$ 0.46 | 58.68 $\pm$ 0.40 |
> | EHNN | 76.51 $\pm$ 1.52 | 87.12 $\pm$ 0.31 | 90.47 $\pm$ 0.43 | 77.95 $\pm$ 0.14 | 86.21 $\pm$ 0.49 | 58.23 $\pm$ 1.07 |
> | T-HyperGNN | 74.20 $\pm$ 1.37 | 86.28 $\pm$ 0.62 | 85.44 $\pm$ 0.14 | 73.48 $\pm$ 0.33 | 85.32 $\pm$ 0.48 | 58.82 $\pm$ 0.49 |

---

> ### Author Response · Authors · 2025-11-25
> **Response to Weakness #2 & Question #1 (Part II)**
>
> **Table R2**: Hyperedge prediction performance with two newly added direction-aware GNNs.
>
> | Method  | Cora (AUROC \| AP) | Pubmed  (AUROC \| AP) | Actor  (AUROC \| AP) | Pokec  (AUROC \| AP)|
> | :----: |  :----: |  :----: |  :----: |  :----: |
> | MLP | 68.01 $\pm$ 1.23 \| 71.32 $\pm$ 1.13 | 66.00 $\pm$ 0.44 \| 69.21 $\pm$ 0.61 | 54.75 $\pm$ 2.29 \| 53.63 $\pm$ 1.56 | 69.69 $\pm$ 2.56 \| 69.07 $\pm$ 3.03 |
> | **MagNet** | **56.45 $\pm$ 0.02 \| 55.18 $\pm$ 0.01**  | **53.64 $\pm$ 0.02** \| **54.79 $\pm$ 0.01** | **50.76 $\pm$ 0.02** \| **50.21 $\pm$ 0.02** | **79.95 $\pm$ 0.01** \| **80.78 $\pm$ 0.01** |
> | **DirGNN** | **63.02 $\pm$ 0.02** \| **61.38 $\pm$ 0.03** | **55.03 $\pm$ 0.02** \| **55.28 $\pm$ 0.02** | **51.72 $\pm$ 0.02** \| **51.33 $\pm$ 0.02** | **80.14 $\pm$ 0.01** \| **79.65 $\pm$ 0.01** |
> | HGNN | 73.70 $\pm$ 1.19 \| 71.13 $\pm$ 1.57 | 66.08 $\pm$ 9.84 \| 63.67 $\pm$ 9.02 | 72.42 $\pm$ 1.96 \| 68.79 $\pm$ 1.83 | 86.09 $\pm$ 0.92 \| 84.32 $\pm$ 0.95 |
> | HyperGCN | 77.34 $\pm$ 1.30 \| 77.15 $\pm$ 0.33 | 66.46 $\pm$ 8.87 \| 64.84 $\pm$ 7.98 | 55.01 $\pm$ 8.76 \| 56.29 $\pm$ 7.44 | 91.45 $\pm$ 0.70 \| 90.76 $\pm$ 0.70 |
> | PhenomNN | 75.71 $\pm$ 0.91 \| 75.22 $\pm$ 1.42 | 74.29 $\pm$ 0.85 \| 72.93 $\pm$ 1.27 | 56.65 $\pm$ 3.04 \| 55.75 $\pm$ 2.87 | 70.83 $\pm$ 2.52 \| 70.17 $\pm$ 2.36 |
> | SheafHyperGNN | 70.53 $\pm$ 5.28 \| 70.93 $\pm$ 4.04 | 68.26 $\pm$ 1.92 \| 68.07 $\pm$ 1.18 | 59.83 $\pm$ 6.77 \| 59.84 $\pm$ 5.73 | 83.44 $\pm$ 2.49 \| 85.11 $\pm$ 1.80 |
> | TF-HNN | 76.94 $\pm$ 0.86 \| 76.57 $\pm$ 0.71 | 73.75 $\pm$ 0.73 \| 75.54 $\pm$ 0.72 | 54.03 $\pm$ 1.71 \| 54.06 $\pm$ 1.57 | 68.00 $\pm$ 0.97 \| 67.41 $\pm$ 1.20 |
> | UniGNN | 73.51 $\pm$ 0.87 \| 75.23 $\pm$ 1.51 | 74.20 $\pm$ 0.82 \| 71.76 $\pm$ 1.16 | 50.24 $\pm$ 1.26 \| 50.01 $\pm$ 0.56 | 85.64 $\pm$ 1.20 \| 84.36 $\pm$ 1.48 |
> | AllSetTransformer | 72.55 $\pm$ 2.95 \| 74.86 $\pm$ 1.85 | 71.09 $\pm$ 2.99 \| 73.15 $\pm$ 2.49 | 55.84 $\pm$ 5.99 \| 58.73 $\pm$ 4.39 | 83.65 $\pm$ 4.34 \| 83.36 $\pm$ 4.72 |
> | ED-HNN | 67.24 $\pm$ 1.91 \| 69.89 $\pm$ 2.24 | 70.09 $\pm$ 0.43 \| 72.61 $\pm$ 0.48 | 51.74 $\pm$ 2.79 \| 52.27 $\pm$ 2.54 | 85.27 $\pm$ 1.48 \| 84.95 $\pm$ 1.43 |
> | EHNN | 78.99 $\pm$ 0.99 \| 79.54 $\pm$ 0.93 | 76.50 $\pm$ 0.62 \| 75.94 $\pm$ 0.70 | 65.69 $\pm$ 0.46 \| 65.37 $\pm$ 0.35 | 88.63 $\pm$ 1.58 \| 91.31 $\pm$ 0.88 |
> | T-HyperGNN | 58.91 $\pm$ 1.23 \| 62.17 $\pm$ 1.58 | 58.35 $\pm$ 4.43 \| 55.81 $\pm$ 3.71 | 49.16 $\pm$ 0.22 \| 50.20 $\pm$ 0.41 | 65.21 $\pm$ 1.21 \| 66.90 $\pm$ 1.56 |
>
> **Table R3**: Hypergraph classification performance with two newly added direction-aware GNNs.
>
> | Method  | RHG-10 (Acc \| F1\_ma)|  IMDB-Dir-Genre (Acc \| F1\_ma)| Steam-Player (Acc \| F1\_ma)| Twitter-Friend (Acc \| F1\_ma)|
> | :----: |  :----: |  :----: |  :----: |  :----: |
> | MLP | 91.70 $\pm$ 1.02 \| 91.43 $\pm$ 1.09 | 75.12 $\pm$ 0.70 \| 71.10 $\pm$ 0.74 | 52.34 $\pm$ 0.55 \| 51.60 $\pm$ 0.68 | 57.25 $\pm$ 1.81 \| 52.88 $\pm$ 4.57 |
> | **MagNet** | **93.20 $\pm$ 0.02** \| **92.95 $\pm$ 0.02** | **75.94 $\pm$ 0.01** \| **71.45 $\pm$ 0.00** | **51.75 $\pm$ 0.02** \| **51.12 $\pm$ 0.03** | **55.11 $\pm$ 0.02** \| **46.64 $\pm$ 0.03** |
> | **DirGNN** | **94.80 $\pm$ 0.01**\| **94.68 $\pm$ 0.01**  | **76.53 $\pm$ 0.02** \| **72.59 $\pm$ 0.02** | **52.33 $\pm$ 0.01** \| **52.24 $\pm$ 0.01** | **54.81 $\pm$ 0.04** \| **46.02 $\pm$ 0.03** |
> | HGNN | 94.60 $\pm$ 1.66 \| 94.47 $\pm$ 1.84 | 76.76 $\pm$ 2.66 \| 72.02 $\pm$ 4.37 | 51.65 $\pm$ 2.51 \| 50.91 $\pm$ 2.92 | 55.42 $\pm$ 2.03 \| 46.81 $\pm$ 4.27 |
> | HyperGCN | 85.50 $\pm$ 1.10 \| 95.42 $\pm$ 1.09 | 77.53 $\pm$ 0.99 \| 72.97 $\pm$ 1.08 | 51.17 $\pm$ 3.32 \| 50.48 $\pm$ 3.12 | 56.95 $\pm$ 4.17 \| 50.12 $\pm$ 5.88 |
> | PhenomNN | 91.10 $\pm$ 0.73 \| 90.77 $\pm$ 0.77 | 74.59 $\pm$ 0.61 \| 70.15 $\pm$ 0.88 | 51.65 $\pm$ 3.06 \| 48.94 $\pm$ 4.55 | 57.40 $\pm$ 3.84 \| 48.26 $\pm$ 4.66 |
> | SheafHyperGNN | 96.00 $\pm$ 1.38 \| 95.96 $\pm$ 1.32 | 77.00 $\pm$ 1.14 \| 72.78 $\pm$ 1.17 | 53.11 $\pm$ 2.39 \| 52.56 $\pm$ 2.74 | 56.49 $\pm$ 2.51 \| 51.43 $\pm$ 4.42 |
> | TF-HNN | 95.90 $\pm$ 0.80 \| 95.88 $\pm$ 0.78 | 76.41 $\pm$ 1.31 \| 71.89 $\pm$ 1.45 | 54.85 $\pm$ 1.82 \| 52.72 $\pm$ 2.54 | 56.18 $\pm$ 3.53 \| 44.17 $\pm$ 8.95 |
> | UniGNN | 95.50 $\pm$ 1.38 \| 95.40 $\pm$ 1.44 | 77.12 $\pm$ 0.88 \| 72.93 $\pm$ 1.43 | 51.46 $\pm$ 2.48 \| 48.85 $\pm$ 2.59 | 55.88 $\pm$ 4.14 \| 46.48 $\pm$ 4.90 |
> | AllSetTransformer | 97.30 $\pm$ 0.98 \| 97.26 $\pm$ 1.04 | 76.47 $\pm$ 1.38 \| 72.26 $\pm$ 1.12 | 53.01 $\pm$ 2.77 \| 48.21 $\pm$ 7.20 | 60.15 $\pm$ 1.70 \| 51.52 $\pm$ 7.00 |
> | ED-HNN | 96.50 $\pm$ 0.77 \| 96.41 $\pm$ 0.78 | 77.12 $\pm$ 1.11 \| 72.87 $\pm$ 0.44 | 52.82 $\pm$ 2.65 \| 48.73 $\pm$ 2.36 | 57.40 $\pm$ 2.66 \| 42.57 $\pm$ 5.09 |

---

> ### Author Response · Authors · 2025-11-25
> **Response to Weakness #2 & Question #1 (Part III)**
>
> * **Findings.** From the results shown in the tables above, we derive the following key findings:
>
>   - In node classification, the two newly added direction-aware GNNs generally fall short of most HNN methods across the six datasets, reflecting the advantage of HNN architectures in modeling higher-order structures. We also observe that DirGNN achieves competitive performance on heterophilous datasets such as Actor and Pokec, likely because its separation mechanism in neighbor aggregation helps mitigate the adverse feature mixing effects induced by heterophily.
>   - In hyperedge prediction, direction-aware GNNs perform notably worse than HNNs and, in many cases, even underperform traditional MLPs. A key reason is that their directional aggregation mechanism, which separates incoming and outgoing neighbors, reinforces a pairwise and asymmetric view of interactions. This asymmetry limits the model’s ability to form coherent representations of multi-node groups and makes it difficult to capture the joint, order-invariant dependencies required for accurate hyperedge prediction.
>   - In hypergraph classification, direction-aware GNNs remain less competitive than state-of-the-art HNNs, which benefit from explicit modeling of higher-order interactions that are crucial for capturing complex hypergraph structures.
>
> These findings highlight the importance of explicit hypergraph modeling, as HNNs can capture higher-order, group-level interactions that pairwise GNN architectures inherently miss, leading to stronger performance on tasks that rely on true hypergraph structure. **We have updated the manuscript to include the new results and analysis in Appendix C.5**. Thank you for your valuable suggestion!

---

> ### Author Response · Authors · 2025-11-25
> **Response to Weakness #3**
>
> > **Weakness #3:** Directed hypergraphs ignored. The benchmark only evaluates undirected hypergraphs, omitting variants that model asymmetric or causal relations, i.e. directed hypergraphs.
>
> **A:** We thank the reviewer for this insightful comment. We fully agree that directed hypergraphs are an important and emerging research direction. Our decision to focus on undirected hypergraphs in the current DHG-Bench is mainly motivated by the current landscape of HNN research, and we clarify this choice as follows:
>
> * **Directed HNNs remain largely under-explored.** Most existing HNN methods are designed for undirected hypergraphs, which remain the dominant setting in current hypergraph learning research. The widely used HNN algorithms integrated into DHG-Bench, such as HGNN, TF-HNN, AllSetTransformer, and ED-HNN, cannot be directly extended to model asymmetric group interactions because they rely on symmetric hyperedges and neglect orientation during message passing. In contrast, only a few recent studies have attempted to develop general HNNs for directed hypergraphs (e.g., GeDi-HNN [6]). As a result, unified and systematic evaluation of directed hypergraph models is still not well supported by the existing methodological landscape, given the limited availability of general and broadly adopted directed HNN architectures.
>
> * **A standardized benchmark for undirected hypergraphs is urgently needed.** As discussed in our introduction, within the mainstream undirected setting, existing HNN studies rely on inconsistent datasets, evaluation protocols, and baseline selections, which makes it difficult to achieve fair and comparable evaluation. In addition, systematic multi-dimensional analysis across effectiveness, efficiency, robustness, and fairness remains largely absent, limiting our ability to gain a clear and reliable understanding of the behavior, strengths, and limitations of current HNN models. Establishing a comprehensive and unified benchmark for undirected hypergraph learning is, therefore, a necessary and foundational step to consolidate the field and to provide a solid basis before advancing to more complex directed hypergraph scenarios.
>
> We appreciate the reviewer’s insightful suggestion regarding the extension to directed hypergraph evaluation. This is indeed an important direction, and we will leave it as a key component of our future work. Our plans for incorporating directed hypergraphs are further detailed in our response to **Question #2**.

---

> ### Author Response · Authors · 2025-11-25
> **Response to Question #2**
>
> > **Question #2:**  Directed hypergraphs and variants. Do you plan to extend DHG-Bench to support directed hypergraphs, heterogeneous hyperedges, or temporal hypergraphs? If not, please justify.
>
> **A:** Thank you for the insightful question. We recognize that extending DHG-Bench to support deep hypergraph learning on more complex hypergraphs (e.g., directed, heterogeneous, and temporal hypergraphs) is essential, as real-world hypergraphs often exhibit temporal, directional, and heterogeneous properties. However, developing HNNs capable of modeling such complex higher-order interactions remains in its early stage [7]. To further enhance HNN research, **we have plans to extend DHG-Bench in the future, aiming to make it capable of benchmarking algorithms on complex hypergraphs.**
>
> * **Future Integration of Complex Hypergraph Datasets.** We are actively exploring the integration of more complex hypergraph datasets that reflect real-world scenarios. This includes directed hypergraphs (e.g., citation networks modeling asymmetric group citation behaviors [6]), heterogeneous hypergraphs (e.g., e-commerce or multimodal recommendation data involving multiple entity types [8]), and temporal hypergraphs (e.g., dynamic user–item interaction logs reflecting evolving higher-order relations [9]). These extensions will further enrich the benchmark and provide broader coverage of real-world hypergraph learning scenarios.
>
> * **Implementation of Specialized HNN Algorithms.** We also plan to provide standardized implementations of HNN algorithms specifically designed for different types of complex hypergraphs and integrate them into DHG-Bench. For example, we intend to incorporate directed HNNs (e.g., GeDi-HNN [6], DHMConv [10]), heterogeneous HNNs (e.g., HWNN [11]), and temporal HNNs (e.g., THINK [12]). These additions will further expand the applicability of DHG-Bench and support more realistic and diverse hypergraph learning tasks.
>
> * **Developing more benchmark tasks for complex hypergraphs.** In addition to expanding our datasets and algorithms, we plan to design additional benchmark tasks tailored for HNNs on complex hypergraph data, enabling more comprehensive evaluation across node-level, edge-level, and graph-level learning scenarios. By providing tasks that are aligned with the structural characteristics of directed, heterogeneous, and temporal hypergraphs, we aim to catalyze the development of HNNs that more effectively exploit the intricate nature of higher-order interactions.
>
> In summary, we are committed to extending DHG-Bench's evaluation capabilities on complex hypergraphs, ensuring that it remains a comprehensive and practical resource for the research community as the field of deep hypergraph learning continues to evolve.

---

> ### Author Response · Authors · 2025-11-25
> **Response to Weakness #4 & Question #3**
>
> > **Weakness #4 & Question #3:** For models that ran out of memory, which mitigation strategies were attempted (e.g., mini-batching, sparse matrices, mixed precision)? Can you report peak memory usage?
>
> **A:** Thank you for your questions. In DHG-Bench, our primary mitigation strategy for handling memory-intensive settings is the unified support for sparse-matrix storage and training. Sparse operations are broadly compatible with all HNN models and effectively reduce memory overhead without affecting training dynamics, making them a practical and reliable choice. Below, we detail this strategy and explain why certain other techniques were not adopted.
>
> * **Support for Sparse Matrix.** DHG-Bench implements full sparse support for all HNNs, including sparse incidence matrices and sparse matrix computations during message passing. Representing the incidence matrix in a sparse format substantially reduces memory consumption, particularly for large-scale datasets. Sparse tensor operations also eliminate the need to materialize dense intermediate matrices during aggregation, which lowers peak memory usage in both the forward and backward passes. This design allows DHG-Bench to scale to larger hypergraphs than would be feasible with dense representations and serves as our main approach to preventing the OOM issue.
>
> * **Why Mini-batching is not Used.** Following the standard practice in most related HNN studies, DHG-Bench employs full-batch training for all models. Hypergraphs differ fundamentally from graphs because hyperedges connect multiple nodes simultaneously. However, there is currently no widely adopted, hypergraph-specific mini-batch sampling strategy that preserves hyperedge integrity or provides unbiased training signals. Existing sampling methods designed for graphs do not directly transfer to hypergraphs, as they often break hyperedge structures or distort higher-order relationships. DHG-Bench therefore follows the full-batch protocol to ensure comparability with prior works. Developing principled mini-batch sampling strategies for hypergraphs is an important direction, and we plan to explore this in future extensions of DHG-Bench.
>
> * **Why Mixed-Precision is not Used.** Mixed-precision training can reduce memory usage in some deep learning models. However, many HNNs rely on sparse operations and irregular message-passing kernels, and while PyTorch technically allows FP16 sparse tensors, most sparse operators either lack full FP16 support or exhibit numerical instability in half-precision settings. To keep the evaluation fair and consistent across all models, we choose not to include the mixed precision strategy.
>
> Regarding peak memory usage, we would like to clarify that **the memory consumption reported in our paper already corresponds to the peak GPU memory observed during model training, as stated in Section. 3.3 (RQ2)**. Therefore, the reported results accurately reflect the maximum memory footprint encountered by each model during the full training process.
>
> **We have added the above discussion of the memory mitigation strategies used during training to Appendix B.7 in the revised manuscript**, and we believe that this clarification enhances the transparency and clarity of our experimental analysis. Thank you again for your constructive suggestion!

---

> ### Author Response · Authors · 2025-11-25
> **Response to Weakness #5 & Question #4 (Part I)**
>
> > **Weakness #5 & Question #4:** Can you quantify heterophily per dataset, test hypotheses (e.g., oversmoothing, feature collapse), and relate results to prior literature on heterophilic GNN behavior?
>
> **A:** Thank you for your insightful question! We would like to first clarify that **we have already reported the hyperedge homophily ratios of all datasets in Table A1 in the original manuscript**, following the metric introduced in [13] for quantifying heterophily in hypergraphs. This ratio is defined as the average proportion of node pairs within each hyperedge that belong to the same class. A lower value therefore indicates higher heterophily, as it reflects fewer same-class node pairs co-occurring within hyperedges. **The detailed values are reproduced below for convenience**.
>
> **Table R4:** Hyperedge homophily ratio for different datasets.
>
> | Cora | Pubmed | Cora-CA | DBLP-CA | NTU2012 | ModelNet40 | Walmart | Trivago | Actor | Ratings | Gamers | Pokec | Yelp |
> | :--: | :----: | :-----: | :------: | :------: | :---------: | :-----: | :------: | :---: | :-----: | :----: | :---: | :---: |
> | 0.75 | 0.78 | 0.78 | 0.87 | 0.79 | 0.87 | 0.60 | 0.98 | 0.46 | 0.37 | 0.49 | 0.45 | 0.29 |
>
> To analyze the underlying causes of performance degradation on heterophilous hypergraphs, we follow the reviewer’s suggestions and test the two key hypotheses: oversmoothing and feature collapse.
>
> At the first step, **we evaluate how the accuracy of four representative HNN architectures changes as the number of layers increases on two heterophilous datasets**, Actor and Pokec, with the goal of examining whether oversmoothing occurs.
>
> **Table R5:** Node classification on Actor with varying layer depths.
>
> | Method             | 1                 | 2                 | 3                 | 4                 | 5                 |
> |-------------------|-------------------|-------------------|-------------------|-------------------|-------------------|
> | MLP               | 81.23 $\pm$ 0.39  | **86.06 $\pm$ 0.36**  | 84.90 $\pm$ 0.41  | 84.55 $\pm$ 0.54  | 84.28 $\pm$ 0.69  |
> | HGNN              | 77.63 $\pm$ 0.74  | 73.84 $\pm$ 0.37  | 70.82 $\pm$ 0.70  | 68.59 $\pm$ 0.68  | 67.33 $\pm$ 0.45  |
> | SheafHyperGNN     | 85.00 $\pm$ 0.32  | 84.71 $\pm$ 0.43  | 83.61 $\pm$ 0.48  | 82.88 $\pm$ 0.41  | 82.15 $\pm$ 0.63  |
> | AllSetTransformer | 85.79 $\pm$ 0.77  | 85.63 $\pm$ 0.35  | 85.68 $\pm$ 0.55  | 85.38 $\pm$ 0.35  | 85.49 $\pm$ 0.21  |
> | ED-HNN            | 85.69 $\pm$ 0.45  | 85.82 $\pm$ 0.28  | 85.53 $\pm$ 0.37  | 84.93 $\pm$ 0.47  | 82.60 $\pm$ 9.96  |
>
> **Table R6:** Node classification on Pokec with varying layer depths.
>
> |  Method                | 1                  | 2                  | 3                  | 4                  | 5                  |
> |-------------------|--------------------|--------------------|--------------------|--------------------|--------------------|
> | MLP               | 57.91 $\pm$ 0.61   | **59.64 $\pm$ 0.48**   | 58.81 $\pm$ 0.58   | 58.52 $\pm$ 0.85   | 58.94 $\pm$ 0.87   |
> | HGNN              | 57.43 $\pm$ 0.67   | 57.48 $\pm$ 0.82   | 57.26 $\pm$ 0.78   | 56.88 $\pm$ 1.24   | 56.79 $\pm$ 0.68   |
> | SheafHyperGNN     | 59.02 $\pm$ 0.42   | 58.94 $\pm$ 0.67   | 58.26 $\pm$ 0.61   | 58.03 $\pm$ 0.83   | 57.93 $\pm$ 0.73   |
> | AllSetTransformer | 58.75 $\pm$ 0.48   | 58.58 $\pm$ 0.55   | 58.50 $\pm$ 0.85   | 58.54 $\pm$ 0.58   | 58.35 $\pm$ 0.34   |
> | ED-HNN            | 58.52 $\pm$ 0.32   | 58.71 $\pm$ 0.30   | 58.74 $\pm$ 0.50   | 58.24 $\pm$ 0.50   | 58.11 $\pm$ 0.58   |
>
> According to Tables R5 and R6, although increasing depth generally causes a gradual performance decline in HNNs (i.e., oversmoothing), all HNN variants already underperform the MLP baseline under the 1-layer message passing. This suggests that **depth is not the primary factor behind the performance gap.**

---

> ### Author Response · Authors · 2025-11-25
> **Response to Weakness #5 & Question #4 (Part II)**
>
> To further examine the underlying factors, we first compute the Mean Average Distance (MAD) [14], a widely adopted metric for measuring the smoothness (i.e., similarity) of graph representations. A lower MAD value indicates that connected nodes have more similar embeddings, reflecting a smoother representation space. Specifically, we report the MAD values for both the raw input features and the representations obtained after the first layer (Tables R7 and R8). Next, to assess the extent of feature mixing under heterophily, we measure the similarity between each node and its heterophilous neighbors using the cosine distance (Tables R9 and R10). Formally, the heterophilous similarity is defined as:
>
> $$
> \mathrm{Sim}^{\text{diff}}
> = \mathrm{avg}_{(i,j):\, j\in\mathcal{N}^{\text{diff}}(i)}
> \cos\left( h_i^{(l)},\; h_j^{(l)} \right)
> $$
> where $\mathcal{N}^{\text{diff}}(i) = \{ j \in \mathcal{N}(i) \mid y_j \neq y_i \}$ denotes the set of heterophilous neighbors whose labels differ from that of node $i$.
>
> **Table R7:** MAD values of raw inputs and 1-layer embeddings on Actor.
> |   Layer   |   HGNN   | SheafHyperGNN | AllSetTransformer |  ED-HNN  |   MLP   |
> | :--: | :-------: | :------------: | :----------------: | :-------: | :------: |
> |  0   | 0.8114 |    0.8114    |      0.8114       |  0.8114  | 0.8114 |
> |  1   | 0.4700 |    0.3976    |      0.2379       |  0.5540  | 0.7456 |
>
> **Table R8:** MAD values of raw inputs and 1-layer embeddings on Pokec.
>
> |  Layer    |   HGNN   | SheafHyperGNN | AllSetTransformer |  ED-HNN  |   MLP   |
> | :--: | :-------: | :------------: | :----------------: | :-------: | :------: |
> |  0   | 0.2697 | 0.2697 | 0.2697 | 0.2697 | 0.2697 |
> |  1   | 0.1054 | 0.1490 | 0.0058 | 0.0592 | 0.2034 |
>
> **Table R9:** $\mathrm{Sim}^{\text{diff}}$ of raw inputs and 1-layer embeddings on Actor.
>
> | Layer     |   HGNN   | SheafHyperGNN | AllSetTransformer |  ED-HNN  |    MLP    |
> | :--: | :-------: | :------------: | :----------------: | :-------: | :--------: |
> |  0   | 0.0584 | 0.0584 | 0.0584 | 0.0584 | 0.0584 |
> |  1   | 0.4013  | 0.4274  | 0.4892  | 0.0515 | -0.0829 |
>
> **Table R10:** $\mathrm{Sim}^{\text{diff}}$ of raw inputs and 1-layer embeddings on Pokec.
> |  Layer    |   HGNN   | SheafHyperGNN | AllSetTransformer |  ED-HNN  |   MLP   |
> | :--: | :-------: | :------------: | :----------------: | :-------: | :------: |
> |  0   | 0.0775 | 0.0775 | 0.0775 | 0.0775 | 0.0775 |
> |  1   | 0.6364  | 0.7573  | 0.9796  | 0.7818  | 0.2990  |
>
> Our empirical analysis reveals **two key observations:** (1) After only 1-layer hypergraph message passing, MAD decreases sharply compared to the raw input features, indicating that node representations rapidly become more homogeneous. This demonstrates that HNN message passing introduces representation smoothness at a very early stage. (2) The similarity between nodes and their heterophilous neighbors increases substantially, suggesting that heterophilous links cause strong cross-class feature mixing and pull representations of different classes closer together. Such mixing reduces class separability and ultimately impairs the effectiveness of HNN-based classifiers in heterophilous settings.
>
> These observations align closely with prior theoretical and empirical findings on heterophilic GNNs. Existing studies (e.g., [15,16,17]) suggest that **heterophily may negatively affect message-passing architectures, because features of nodes from different classes are falsely mixed, leading to feature collapse and making nodes increasingly indistinguishable**. Our results directly validate this hypothesis in the hypergraph setting: the sharp MAD reduction and pronounced cross-class similarity we observe mirror the failure patterns reported in these works.
>
> We thank the reviewer for emphasizing the importance of heterophily analysis. We believe that such an investigation provides deeper insights into the behavior of HNNs and can inform the development of more effective architectures tailored to heterophilic hypergraphs. **We have included the above results and analysis in Appendix C.6 in the revised manuscript.**

---

> ### Author Response · Authors · 2025-11-25
> **Response to Question #5**
>
> > **Question #5:** **Fairness metrics.** For datasets lacking explicit sensitive attributes, how were $\Delta_{DP}/\Delta_{EO}$ computed? Were proxy attributes used, and how were they validated.
>
> **A:** Thank you for your questions. We would like to clarify that our benchmark evaluates fairness only on datasets that contain explicit demographic-sensitive attributes, including
> German, Bail, and Credit. In these datasets, the sensitive attribute (e.g., gender or race) is directly provided in the raw data, allowing us to compute both $\Delta_{DP}$ and $\Delta_{EO}$ in a standard manner. For all other datasets in our benchmark, we do not perform any fairness evaluation and do not report fairness metrics, as these datasets do not include explicit sensitive attribute information. Notably, we do not create or use any proxy attributes for fairness evaluation, as this could introduce uncontrolled biases and undermine the reliability of the fairness analysis.

---

> ### Author Response · Authors · 2025-11-25
> **Conclusion**
>
> Thank you again for your time and valuable insights. We hope our responses have fully addressed your concerns. We are excited about the potential impact of DHG-Bench on the community and truly appreciate your support in advancing this work.
>
> **As the current ratings reveal somewhat negative evaluations of our work, we are wondering if you could kindly consider raising the ratings to show your support for our work. Thank you!**
>
> ---
>
> **References**
>
> **[1]** Chien, E., Pan, C., Peng, J., & Milenkovic, O. You are AllSet: A Multiset Function Framework for Hypergraph Neural Networks. In International Conference on Learning Representations.
>
> **[2]** Wang, Y., Gan, Q., Qiu, X., Huang, X., & Wipf, D. (2023, July). From hypergraph energy functions to hypergraph neural networks. In International Conference on Machine Learning (pp. 35605-35623). PMLR.
>
> **[3]** Bai, S., Zhang, F., & Torr, P. H. (2021). Hypergraph convolution and hypergraph attention. Pattern Recognition, 110, 107637.
>
> **[4]** Zhang, X., He, Y., Brugnone, N., Perlmutter, M., & Hirn, M. (2021). Magnet: A neural network for directed graphs. Advances in neural information processing systems, 34, 27003-27015.
>
> **[5]** Rossi, E., Charpentier, B., Di Giovanni, F., Frasca, F., Günnemann, S., & Bronstein, M. M. (2024, April). Edge directionality improves learning on heterophilic graphs. In Learning on graphs conference (pp. 25-1). PMLR.
>
> **[6]** Fiorini, S., Coniglio, S., Ciavotta, M., & Del Bue, A. (2024). Let there be direction in hypergraph neural networks. Transactions on Machine Learning Research.
>
> **[7]** Kim, S., Lee, S. Y., Gao, Y., Antelmi, A., Polato, M., & Shin, K. (2024, August). A survey on hypergraph neural networks: an in-depth and step-by-step guide. In Proceedings of the 30th ACM SIGKDD Conference on Knowledge Discovery and Data Mining (pp. 6534-6544).
>
> **[8]** Khan, B., Wu, J., Yang, J., & Ma, X. (2025). Heterogeneous hypergraph neural network for social recommendation using attention network. ACM Transactions on Recommender Systems, 3(3), 1-22.
>
> **[9]** Wang, J., Ding, K., Hong, L., Liu, H., & Caverlee, J. (2020, July). Next-item recommendation with sequential hypergraphs. In Proceedings of the 43rd international ACM SIGIR conference on research and development in information retrieval (pp. 1101-1110).
>
> **[10]** Zhao, W., Ma, Z., & Yang, Z. (2024, April). Dhmconv: Directed hypergraph momentum convolution framework. In International Conference on Artificial Intelligence and Statistics (pp. 3385-3393). PMLR.
>
> **[11]** Sun, X., Yin, H., Liu, B., Chen, H., Cao, J., Shao, Y., & Viet Hung, N. Q. (2021, March). Heterogeneous hypergraph embedding for graph classification. In Proceedings of the 14th ACM international conference on web search and data mining (pp. 725-733).
>
> **[12]** Agarwal, S., Sawhney, R., Thakkar, M., Nakov, P., Han, J., & Derr, T. (2022, November). Think: Temporal hypergraph hyperbolic network. In 2022 IEEE International Conference on Data Mining (ICDM) (pp. 849-854). IEEE.
>
> **[13]** Li, M., Gu, Y., Wang, Y., Fang, Y., Bai, L., Zhuang, X., & Lio, P. (2025, April). When hypergraph meets heterophily: New benchmark datasets and baseline. In Proceedings of the AAAI Conference on Artificial Intelligence (Vol. 39, No. 17, pp. 18377-18384).
>
> **[14]** Chen, D., Lin, Y., Li, W., Li, P., Zhou, J., & Sun, X. (2020, April). Measuring and relieving the over-smoothing problem for graph neural networks from the topological view. In Proceedings of the AAAI conference on artificial intelligence (Vol. 34, No. 04, pp. 3438-3445).
>
> **[15]** Zhu, J., Yan, Y., Zhao, L., Heimann, M., Akoglu, L., & Koutra, D. (2020). Beyond homophily in graph neural networks: Current limitations and effective designs. Advances in neural information processing systems, 33, 7793-7804.
>
> **[16]** Luan, S., Hua, C., Lu, Q., Zhu, J., Zhao, M., Zhang, S., ... & Precup, D. (2022). Revisiting heterophily for graph neural networks. Advances in neural information processing systems, 35, 1362-1375.
>
> **[17]** Yan, Y., Hashemi, M., Swersky, K., Yang, Y., & Koutra, D. (2022, November). Two sides of the same coin: Heterophily and oversmoothing in graph convolutional neural networks. In 2022 IEEE International Conference on Data Mining (ICDM) (pp. 1287-1292). IEEE.

---

### Official Review · Reviewer_uqj6 · 2025-10-30

**Soundness:** 3
**Presentation:** 4
**Contribution:** 3
**Rating:** 8
**Confidence:** 4

**Summary:**

The paper introduces a benchmark for hypergraph representation learning, by evaluating 17 popular hypergraph networks on 22 datasets containing node-level, edge-level and hypergraph-level tasks. The evaluation is performed in terms of accuracy, efficiency, robustness and fairness. The paper also provide the code for the benchmark, representing a useful resource for evaluating future models.

**Strengths:**

- Hypergraph models suffer from inadequate evaluation, which slows down the advancement of the field. Moreover, most existing setups contain only node-level classification tasks. This paper represents a significant step forward in understanding the limitations of current models and provides a consistent, uniform framework for evaluating new models in a fair way.
- The paper points out a couple of limitations exhibited by current models, which represent good areas for future research. In particular, it shows how little progress has been made in hyperedge-level prediction, highlighting the need for more focused efforts in this area.
- Exploring additional metrics such as robustness and fairness is an important represents an important direction for advancing the field.

**Weaknesses:**

- The node classification setup used in the paper appears similar to that of ED-HNN. However, the reported results are noticeably lower. Is there any major difference in the training setup?
- I agree that structural robustness is an important metric. However, for models that explicitly take connectivity into account, not observing a drop in performance at a 90% perturbation ratio seems more like a negative result than a positive one. The paper presents robust performance across different perturbation levels as a desirable outcome, but to me, this suggests either that the dataset does not require higher-order processing or that the model does not properly incorporate structural information. I suggest that once the new metric is introduced, the authors also include a discussion on how a good hypergraph model is expected to behave under such conditions. ****
- It would be interesting to see more statistics on the datasets used in the experiments to highlight the extent to which they are representative of higher-order interactions. For example, it is known that PubMed has around 80% of its nodes isolated. Reporting statistics such as the number of isolated nodes, the number of nodes involved only in pairwise interactions, and the number of nodes participating in higher-order interactions would be particularly useful for node-level classification tasks, where the performance on isolated nodes would depend solely on the MLP component, regardless of the model used.

**Questions:**

Please see the Weaknesses section

---

> ### Author Response · Authors · 2025-11-25
> **Overall Response to Reviewer uqj6**
>
> Thank you for your thoughtful and encouraging feedback. We are glad that you recognize the importance of establishing a comprehensive and unified evaluation framework for hypergraph learning. Your acknowledgment of our efforts to identify limitations in existing models, especially the limited progress in hyperedge-level prediction, is greatly appreciated. We are also grateful for your positive remarks on the inclusion of robustness and fairness as new evaluation dimensions, which we believe are essential for advancing the field.
>
> We respond to your concerns and questions as follows.
>
> **[Preview]**
>
> * **Weakness #1:** Lower ED-HNN performance.
> * **Weakness #2:** Insufficient interpretation under varying metrics.
> * **Weakness #3:** More statistics on the datasets.
> * **Conclusion.**

---

> > ### Author Response · Authors · 2025-11-25
> > **Response to Weakness #1**
> >
> > > **Weakness #1**: The node classification setup used in the paper appears similar to that of ED-HNN. However, the reported results are noticeably lower. Is there any major difference in the training setup?
> >
> > **A:** Thank you for your question. We would like to clarify that our experiments strictly follow the official ED-HNN implementation, using the same data split ratios, preprocessing procedures, and evaluation metrics. Therefore, **no intentional differences were introduced into the experimental setup**. The observed performance gap may stem from several factors known to affect reproducibility in hypergraph learning models:
> >
> > * **Randomness and hardware-dependent non-determinism.** Although we used the same random seed as in the ED-HNN codebase, differences in PyTorch versions, CUDA/cuDNN kernels, and GPU architectures can cause seeds to produce slightly different parameter initializations, floating-point behavior, and non-deterministic kernel execution. These factors are known to introduce noticeable variability in HNN performance across machines. Specifically, ED-HNN contains multiple MLP-based components with relatively large parameterization, which may increase its sensitivity to weight initialization. Small differences in the initial parameters or numerical precision can potentially lead to variations in training trajectories and final performance.
> >
> > * **The optimal hyperparameter configuration for ED-HNN is not fully specified.** Although the original paper specifies ranges for some parameters, several key settings remain unclear. For instance, in the core "EquivSetConv" module, the parameter "alpha" controls the balance between propagated features and initial node representations, yet its recommended value for different datasets is not explicitly stated. Likewise, the choices of activation functions for various MLP components across datasets are not fully documented. Although we performed a careful search over reasonable ranges for these unspecified parameters, the resulting configuration may still differ from the optimal settings used in the original implementation.
> >
> > We sincerely thank the reviewer for your careful evaluation. **All experiments in DHG-Bench were conducted under a fully controlled and identical experimental environment, following the same training setup and evaluation protocols across all models**. This design ensures a fair comparison among different HNN architectures. For better reproducibility, we have also provided detailed descriptions of the experimental environment in Appendix B.3, as well as the hyperparameter configuration files in our GitHub repository.

---

> > > ### Author Response · Authors · 2025-11-25
> > > **Response to Weakness #2**
> > >
> > > > **Weakness #2**: I agree that structural robustness is an important metric. However, for models that explicitly take connectivity into account, not observing a drop in performance at a 90% perturbation ratio seems more like a negative result than a positive one. The paper presents robust performance across different perturbation levels as a desirable outcome, but to me, this suggests either that the dataset does not require higher-order processing or that the model does not properly incorporate structural information. I suggest that once the new metric is introduced, the authors also include a discussion on how a good hypergraph model is expected to behave under such conditions.
> > >
> > > **A:** Thank you for your valuable suggestions. We fully agree that, for the newly introduced **robustness** and **fairness** metrics in our work, it is important to clearly discuss how an ideal HNN model is expected to behave during evaluation. **In response, we have added a detailed discussion as follows:**
> > >
> > > * **Discussion on Robustness Metrics**
> > >
> > >     - **Structure Robustness:** (1) **In homophilous settings**, meaningful higher-order relations benefit classification. Under drop perturbations, a desirable HNN should maintain accuracy that is no lower than that of structure-agnostic baselines (e.g., MLPs), and ideally remain as stable as possible. This indicates that when higher-order structure exists, the model is indeed able to effectively leverage it. Under addition perturbations, which introduce noisy or spurious links, an ideal HNN is expected to identify and down-weight these noisy edges during message passing. Consequently, the model should also maintain stable performance and stay close to the clean-hypergraph accuracy, demonstrating resilience to the adverse effects of structural noise. (2) **In heterophilous settings**, many higher-order connections are not helpful and may even be harmful. In this case, as the perturbation ratio increases, a robust HNN is expected to show a performance trend that remains stable or even improves. Such a trend indicates that disrupting harmful heterophilous links enables the model to better capture the remaining homophilous patterns, reflecting stronger robustness to misleading structural signals.
> > >
> > >     - **Feature Robustness:** For feature robustness evaluation, an ideal HNN is one whose predictive performance degrades slowly as feature noise increases or feature sparsity becomes more severe. Under our benchmark setting, we expect a good HNN to maintain an average performance clearly above the baseline obtained when all features are replaced with random noise, indicating that the model can effectively exploit meaningful feature signals. Likewise, as the feature sparsity ratio increases, the model’s performance should decline gradually while remaining above the extreme case where only a single feature dimension is preserved and, within this feasible range, stay as close as possible to the clean-hypergraph performance. Such behavior reflects the model’s ability to utilize informative features even under highly degraded or partially missing feature conditions.
> > >
> > >     - **Label Robustness:** For label robustness evaluation, we regard an ideal HNN as one whose predictive performance remains insensitive to different levels of label noise and label sparsity. Under our benchmark setting, a strong HNN should retain test accuracy close to its clean-data performance, showing either minimal degradation or no noticeable drop as the proportion of noisy labels increases or as the fraction of labeled training nodes decreases.
> > >
> > > * **Discussion on Fairness Metrics**
> > >
> > >     For fairness evaluation, an ideal HNN maintains strong predictive performance while exhibiting no algorithmic bias across different sensitive demographic groups. Specifically, under our benchmark setting, a good HNN should achieve high node classification accuracy while simultaneously attaining low values on the fairness metrics demographic parity ($\Delta_{DP}$) and equalized odds ($\Delta_{EO}$).
> > >
> > > **We have incorporated the above discussion into Appendix B.6 in the revised manuscript.** We believe that adding this discussion provides clearer guidance for interpreting the strengths and weaknesses of different models in our benchmark, and also helps identify promising directions for improving future HNN designs. Thank you for your suggestion.

---

> ### Author Response · Authors · 2025-11-25
> **Response to Weakness #3**
>
> > **Weakness #3**: It would be interesting to see more statistics on the datasets used in the experiments to highlight the extent to which they are representative of higher-order interactions. For example, it is known that PubMed has around 80% of its nodes isolated. Reporting statistics such as the number of isolated nodes, the number of nodes involved only in pairwise interactions, and the number of nodes participating in higher-order interactions would be particularly useful for node-level classification tasks, where the performance on isolated nodes would depend solely on the MLP component, regardless of the model used.
>
> **A:** Thank you for your insightful suggestions. We have now provided the statistics on **the number of isolated nodes ($I_{node}$)**, **the number of nodes involved only in pairwise interactions ($P_{node}$)**, and **the number of nodes participating in higher-order interactions ($H_{node}$)** for all datasets in Table A1.
>
> * **Table A1 (Revised)**: Statistics of the standard node-level datasets: $|e|$ denotes the hyperedge size, while $\mathcal{H_{edge}}$ indicates the hyperedge homophily ratio introduced in [1]. $I_{node}$, $P_{node}$, and $H_{node}$ indicate the isolated nodes, the nodes involved only in pairwise interactions, and the nodes participating in higher-order interactions.
>
>     | Dataset     | # Nodes | # Edges | # Features | Avg. \|e\| | $\mathcal{H}_{edge}$ | \# $I_{node}$ | \# $P_{node}$ | \# $H_{node}$ | # Classes |
>     | :---------: | :------: | :------: | :---------: | :-------: | :------------------: | :------------: | :------------: | :------------: | :--------: |
>     | Cora        | 2,708   | 1,579   | 1,433       | 3.03      | 0.75   | 1,274 | 205  | 1,229  | 7   |
>     | Pubmed      | 19,717  | 7,963   | 500         | 4.35      | 0.78   | 15,877 | 201   | 3,639   | 3   |
>     | Cora-CA     | 2,708   | 1,072   | 1,433       | 4.28      | 0.78   | 320   | 278  | 2,110  | 7   |
>     | DBLP-CA     | 41,302  | 22,363  | 1,425       | 4.45      | 0.87   | 0     | 3,876 | 37,426 | 6   |
>     | NTU2012     | 2,012   | 2,012   | 100         | 5.00      | 0.79   | 0     | 0    | 2,012  | 67  |
>     | ModelNet40  | 12,311  | 12,311  | 100         | 5.00      | 0.87   | 0     | 0    | 12,311 | 40  |
>     | Walmart     | 88,860  | 69,906  | 100         | 6.59      | 0.60   | 0     | 3,295 | 85,565 | 11  |
>     | Trivago     | 172,738 | 233,202 | 300         | 3.12      | 0.98   | 0     | 25,532| 147,206| 160 |
>     | Actor       | 16,255  | 10,164  | 50          | 5.25      | 0.46   | 563   | 600  | 15,092 | 3   |
>     | Ratings     | 22,299  | 2,090   | 111         | 3.10      | 0.37   | 19,175 | 176 | 2,948   | 5   |
>     | Gamers      | 16,812  | 2,627   | 7           | 6.23      | 0.49   | 456   | 624  | 15,732 | 2   |
>     | Pokec       | 14,998  | 2,406   | 65          | 2.29      | 0.45   | 11,798  | 1,948  | 1,252   | 2   |
>     | Yelp        | 50,758  | 679,302 | 1,862       | 6.66      | 0.29   | 0     | 19   | 50,739 | 9   |
>
> * **Findings**. As shown in the revised Table A1, we find several distinct patterns across datasets regarding their degree of higher-order connectivity. **Pubmed, Ratings, and Pokec exhibit extremely high proportions of isolated nodes (over 78%)**, indicating that the majority of their nodes do not participate in any higher-order interactions and therefore cannot benefit from higher-order message passing. In contrast, datasets such as **Cora-CA, Actor, and Gamers contain substantially more nodes engaged in higher-order hyperedges than nodes that are isolated or involved only in pairwise interactions**, suggesting that higher-order information plays a more prominent role in these graphs. Moreover, **DBLP-CA, NTU2012, ModelNet40, Walmart, Trivago, and Yelp contain no isolated nodes at all, and the number of nodes participating in higher-order interactions overwhelmingly dominates**, reflecting rich and strong higher-order structural patterns in these datasets.
>
> **We have updated Table A1 in Appendix A.1 in the revised manuscript.** We believe these statistics are highly valuable for understanding the extent to which each dataset exhibits higher-order interactions, and they provide a clearer context for interpreting node-level classification performance across different HNN models. Thank you for this constructive suggestion!

---

> ### Author Response · Authors · 2025-11-25
> **Conclusion**
>
> Thank you again for your time and valuable insights, and hope our responses can address all your concerns! We are excited about the impact DHG-Bench may have on the community and are grateful for your support in advancing this work.
>
> ---
>
> **References**
>
> **[1]** Li, M., Gu, Y., Wang, Y., Fang, Y., Bai, L., Zhuang, X., & Lio, P. (2025, April). When hypergraph meets heterophily: New benchmark datasets and baseline. In Proceedings of the AAAI Conference on Artificial Intelligence (Vol. 39, No. 17, pp. 18377-18384).

---

### Author Response · Authors · 2025-11-25
**Global Response**

We sincerely thank all reviewers for their thoughtful and constructive feedback on our paper. We are delighted that the reviewers recognize the significance of our benchmark to the Deep Hypergraph Learning (DHGL) research community. Specifically, reviewers acknowledged:

* **Timely and Impactful Benchmark.** Reviewers agreed that DHG-Bench provides a timely and much-needed benchmark that addresses the long-standing inadequacy of systematic evaluation in hypergraph learning. The benchmark fills a critical gap and is expected to accelerate progress in the field. (Reviewer **uqj6**, Reviewer **CWSt**)

* **Comprehensive Coverage of Models, Datasets, and Tasks.** The benchmark is praised for its comprehensive coverage, implementing 17 HNN models, spanning 22 datasets, and supporting node-, hyperedge-, and graph-level tasks. Reviewers noted that most benchmark datasets are included and that task coverage is complete. (Reviewer **KSgV**, Reviewer **CWSt**, Reviewer **FzKL**)


* **Multi-Dimensional and High-Quality Experimental Design.** Reviewers highlighted that DHG-Bench moves beyond accuracy-focused evaluation by incorporating efficiency, robustness, and fairness, yielding a holistic and rigorous experimental design. The RQ1–RQ4 structure was regarded as clear, comprehensive, and highly informative. (Reviewer **uqj6**, Reviewer **KSgV**, Reviewer **FzKL**)

* **Insightful Findings and Guidance for Future Research.** Reviewers appreciated the benchmark’s insights, including the stagnation in hyperedge prediction, the fairness brittleness exhibited by HNNs, the scalability bottlenecks, and the absence of a universally dominant model. Together, these findings reveal important limitations in current approaches and point toward promising directions for future research. (Reviewer **uqj6**, Reviewer **KSgV**, Reviewer **FzKL**)

---

### **Summary of Changes**

We have provided detailed responses to all reviewer comments and incorporated additional experiments and clarifications in the revised manuscript. All modifications in the manuscript are highlighted in **blue**.

The changes are summarized as follows:

1. We have added experiments in label-scarce scenarios (**Appendix C.4**; Reviewer CWSt).
2. We have discussed the newly introduced evaluation metrics in detail (**Appendix B.6**) and revised **Table A1** to include additional statistics (Reviewer uqj6).
3. We have added experiments comparing direction-aware GNNs (**Appendix C.5**; Reviewer KSgV).
4. We have elaborated on our memory mitigation strategy (**Appendix B.7**; Reviewer KSgV).
5. We have added experiments analyzing the influence of heterophilic hypergraphs (**Appendix C.6**; Reviewer KSgV).
6. We have added experiments on hyperedge prediction under temporal and inductive split settings (**Appendix C.7**; Reviewer FzKL).
7. We have added experiments evaluating HNNs in self-supervised settings (**Appendix C.8**; Reviewer FzKL).
8. We have analyzed model sensitivity to the hyperedge size distribution (**Appendix C.9**) and model behavior on extreme-degree nodes (**Appendix C.10**) (Reviewer FzKL).
9. We have provided practical guidance on selecting appropriate HNNs for different tasks (**Section 5**; Reviewer FzKL).
10. We have explained performance variations across datasets (**Appendix D.1**) and discussed key trade-offs among performance, scalability, and data characteristics (**Appendix D.2**) (Reviewer FzKL).

Once again, we are grateful for the reviewers’ valuable insights, which have greatly helped us improve our work. We have made every effort to address all concerns thoroughly. Thank you for considering our rebuttal, and we kindly request that you reconsider the ratings of our paper, if possible.

---

### Comment · Reviewer_CWSt · 2025-11-27

Dear Reviewers,

I am Reviewer CWst for this manuscript. First of all, thank you very much for your continued academic service. Since the discussion has been relatively quiet so far, I would like to briefly share my current thoughts on the manuscript.

In my view, this work offers two notable contributions:

**1. A common benchmark for reproducible research**

In contrast to graph representation learning, where libraries such as PyG are widely used, the hypergraph learning community still lacks a commonly adopted benchmark. Existing resources either cover only a limited set of methods or miss many recent advances. DHG-Bench, by including a broad and up-to-date collection of hypergraph methods, has strong potential to serve as a “PyG for hypergraphs,” thereby improving reproducibility and facilitating future research in hypergraph learning.

**2. Potential impact on real-world applications**

Although hypergraphs have been frequently highlighted for their applicability across diverse domains, implementing and integrating hypergraph neural networks in practice remains challenging, especially for researchers and practitioners outside this area. A well-structured benchmark such as DHG-Bench can lower this barrier, making it easier for other communities to adopt hypergraph neural networks and explore new applications.

For these reasons, although it may be somewhat early, I am inclined to champion the acceptance of this paper, and I respectfully encourage the reviewers to consider these points when reading the authors’ responses.

Once again, thank you very much for your time and valuable service.

Best regards,

Reviewer CWst

---

> ### Author Response · Authors · 2025-11-27
>
> We sincerely thank Reviewer CWSt for the thoughtful and encouraging comments. Your recognition of DHG-Bench’s reproducibility value and its potential real-world impact is deeply motivating for us. We truly appreciate your support and will continue to maintain and extend DHG-Bench, further broadening its practical value and strengthening its impact within the research community.

---

> ### Comment · Reviewer_KSgV · 2025-11-27
>
> Dear Authors and Reviewers,
>
> I have carefully read the paper and the accompanying discussion, and I would like to offer my opinion on this work.
>
> My primary concern relates to the novelty of the proposed approach, particularly given the substantial number of existing surveys in the current state of the art.
>
> While the work shows potential, in my view it is not yet ready for publication for two main reasons:
>
> 1. **Missing Comparison to Directed Hypergraph Methods.**
>    The paper proposes a new benchmarking framework; however, as highlighted in the author response, a significant portion of the datasets employed in the study are known to perform better with directed GNNs. For this reason, I believe the work lacks an essential comparison with the emerging and rapidly growing area of *directed hypergraph models* [1, 2]. Including such baselines is crucial to demonstrate that the proposed benchmark contributes meaningfully beyond existing surveys.
>
> 2. **Lack of Novel Dataset Contributions.**
>    Rather than relying solely on reintroducing existing state-of-the-art datasets (as stated among the contributions), the paper would benefit substantially from proposing *new* datasets specifically designed to showcase the advantages of HGNNs over standard graph-based approaches. Such datasets would better support the claimed contributions and offer concrete evidence of when and why hypergraph structures are beneficial. I recommend considering the methodology illustrated in paper [1] as an example of an impactful approach.
>
> Sincerely,
>
> Reviewer KSgV
>
> [1] Fiorini, Stefano, et al. *Let there be direction in hypergraph neural networks.* Transactions on Machine Learning Research (2024).
> [2] Ma, Zitong, Wenbo Zhao, and Zhe Yang. *Directed hypergraph representation learning for link prediction.* International Conference on Artificial Intelligence and Statistics. PMLR, 2024.
> [3] Platonov, Oleg, et al. *A critical look at the evaluation of GNNs under heterophily: Are we really making progress?* The Eleventh International Conference on Learning Representations.

---

> > ### Author Response · Authors · 2025-12-02
> > **Follow-Up Response to Reviewer KSgV (Part I)**
> >
> > We thank the reviewer `KSgV` for the time and effort spent reading our paper and response. To avoid any misunderstanding, we would like to clarify several points related to the interpretation of our results and the intended scope of our contribution.
> >
> > ### 1. Response to Novelty Concern (Comparison with Surveys)
> >
> > > **Reviewer's Comment:** "My primary concern relates to the novelty of the proposed approach, particularly given the substantial number of existing surveys in the current state of the art."
> >
> > The reviewer questions the novelty of our work by comparing it to existing HNN surveys. We respectfully clarify that surveys and benchmarks serve fundamentally different purposes. While prior surveys (e.g., [1, 2]) focus on conceptual categorization and taxonomy, our work introduces the first standardized and reproducible empirical benchmark for HNNs. These contributions are complementary, not conflicting.
> >
> > * **Filling the Evaluation Gap (vs. Literature Summary):** Currently, the hypergraph learning community lacks a commonly adopted benchmark. While surveys summarize and discuss existing methods, they do not offer a unified evaluation framework. Our work fills this gap by integrating a broad and up-to-date collection of hypergraph methods into a standardized protocol and codebase, functioning as a **“PyG for hypergraphs,”** which no existing survey attempts to deliver.
> >
> > * **Empirical Insights (vs. Narrative Descriptions):** Our large-scale and multi-dimensional empirical analysis uncovers the potential strengths and limitations of existing HNN algorithms, offering valuable insights that can inform and inspire future research in this field. These empirical insights cannot be obtained through literature summaries or taxonomic surveys alone.
> >
> > * **Lowering the Entry Barrier (vs. Theoretical Review):** By releasing an easy-to-use library and unified pipeline, we significantly lower the barrier for practitioners from diverse domains (e.g., computer vision, bioinformatics, recommender systems) to integrate HNNs into their applications. This practical impact extends beyond the scope of surveys, which typically remain theoretical.
> >
> > **Conclusion:** Our contribution is not "another survey," but the first systematic and comprehensive benchmark that enables reproducible, apples-to-apples comparisons across HNN methods.

---

> ### Author Response · Authors · 2025-12-02
> **Follow-Up Response to Reviewer KSgV (Part II)**
>
> ### 2. Response to Directed Hypergraph Methods
>
> > **Reviewer's Comment:** "...as highlighted in the author response... perform better with directed GNNs. For this reason, I believe the work lacks an essential comparison with... directed hypergraph models..."
>
> We would like to clarify the presentation of the results in our previous response, as the reviewer’s current interpretation appears to stem from a misunderstanding of the formatting:
>
> * **Clarification on "Highlighted" Entries:** The bolded numbers in Tables R1–R3 were used solely to indicate the newly added results for clarity, rather than to denote best performance.
> * **Performance Reality:** A direct inspection of the values confirms that Directed GNNs generally underperform state-of-the-art HNNs in our evaluation, particularly in hyperedge prediction. This contradicts the premise that directed methods are superior on these datasets.
>
> Furthermore, regarding the inclusion of Directed Hypergraphs Models:
> While we agree that directed HNNs represent an emerging and promising research direction, the vast majority of mainstream HNN literature focuses on undirected hypergraphs. The community currently lacks a fair and comprehensive benchmark even for this dominant setting.
>
> * **Scope Justification:** Our goal is to establish a solid foundation for the most widely used (undirected) setting first. Mixing early-stage directed methods into a predominantly undirected benchmark would introduce variable settings that complicate fair comparison.
> * **Future Roadmap:** We fully intend to extend DHG-Bench to complex variations, including directed hypergraphs, once the foundational benchmark is established.
>
> **Conclusion:** Since the premise of superior performance does not hold, and the suggested direction falls outside the mainstream undirected scope, we believe the current benchmark provides a sufficient, fair, and comprehensive evaluation without these additional baselines.
>
> ### 3. Response to Dataset Contributions
>
> > **Reviewer's Comment:** "...proposing new datasets specifically designed to showcase the advantages of HGNNs... I recommend considering the methodology illustrated in paper [3]..."
>
> The reviewer suggested constructing new datasets (referencing the methodology in [3], which was cited as [1] in the reviewer’s comments) to demonstrate the advantages of HNNs. We address this concern from two perspectives:
>
> * **Sufficiency of Real-World Data:** Our benchmark has integrated 22 widely-used real-world datasets across diverse domains. Our experimental results on these standard benchmarks already demonstrate the superior performance of HNNs over graph-based baselines in many settings. We believe that evaluations on established real-world datasets provide more practical evidence than experiments on synthetic data.
>
> * **Incompatibility of Suggested Method [3]:** We appreciate the reference to [3]. However, the data construction method in [3] is specifically designed for directed hypergraphs. Given that the mainstream HNN models operate on undirected structures (which defines the scope of our benchmark), applying a directed generation method would create a fundamental structural mismatch.
>
> **Conclusion:** To ensure a fair and appropriate comparison with the extensive existing literature, we prioritize standard real-world datasets over synthetic ones generated by recent methods that differ in structural assumptions.
>
> ---
>
> **References**
>
> **[1]** Kim, S., Lee, S. Y., Gao, Y., Antelmi, A., Polato, M., & Shin, K. (2024, August). A survey on hypergraph neural networks: an in-depth and step-by-step guide. In Proceedings of the 30th ACM SIGKDD Conference on Knowledge Discovery and Data Mining (pp. 6534-6544).
>
> **[2]** Antelmi, A., Cordasco, G., Polato, M., Scarano, V., Spagnuolo, C., & Yang, D. (2023). A survey on hypergraph representation learning. ACM Computing Surveys, 56(1), 1-38.
>
> **[3]** Fiorini, S., Coniglio, S., Ciavotta, M., & Del Bue, A. (2024). Let there be direction in hypergraph neural networks. Transactions on Machine Learning Research.

---

### Author Response · Authors · 2025-12-02
**Rebuttal Summary for Area Chair**

Dear Area Chair,

Thank you so much for reviewing our paper and the discussion process of our submission.

Here, we would like to briefly summarize the key progress we made during the discussion period.

Our paper initially received four reviews with scores 8 (uqj6), 4 (KSgV), 6 (CWSt), and 4 (FzKL). During the discussion period, we carefully addressed every concern and question raised by all reviewers and updated the manuscript accordingly. Two of the reviewers provided feedback on our rebuttal and one of them had already updated the score **before the bug was discovered**, as follows:

- Reviewer **CWSt** not only increased score **6 → 8 (26 Nov, 2025 17:15 AOE Time)**, moving from borderline accept to clear accept, but also **explicitly championed** our work **(26 Nov, 2025 22:29 AOE Time)** in newly added comments, highlighting its strong potential to serve as a "PyG for hypergraphs" that ensures reproducibility and lowers the barrier for real-world applications.
- Reviewer **uqj6** recommended a clear acceptance with score **8** before the discussion period and has not responded to our rebuttal.
- Reviewer **FzKL** gave us an initial score of 4 and has not responded to our rebuttal.
- Reviewer **KSgV** gave us an initial score of 4 and replied shortly before the discussion was paused.

**Regarding Reviewer KSgV specifically**:
They initially raised 10 questions (e.g., heterophily analysis and metric clarification). Our detailed rebuttal effectively addressed the majority of these technical concerns, as evidenced by the reviewer’s follow-up comments, which significantly narrowed the scope to three remaining conceptual points. Although the discussion period was paused, preventing further feedback, we provided a comprehensive response to these comments: (1) we clarified the misconception equating our empirical benchmark with conceptual surveys; (2) we corrected the reviewer’s factual misunderstanding of the results (they may have mistaken bold text for indicating the best performance), thereby justifying our baseline choices; and (3) we explained the structural incompatibility of the reviewer's suggested directed dataset for our undirected scope.

---

> ### Author Response · Authors · 2025-12-02
> **Summary of Responses for each Reviewer (Part I)**
>
> To facilitate your review process, we also provide a concise summary of the reviewers’ main concerns and our corresponding responses below. For a detailed summary of the manuscript revisions, please refer to our **Global Response**.
>
> ## Reviewer CWSt (Score 6 → 8)
>
> ### Responses to Initial Concerns
>
> * **Usability & Installation (W1)**
>
>     We supported pip installation for the package, as suggested. We also updated the README.md file in the GitHub repository referenced in the revised paper to provide clear instructions for quickly running experiments.
>
> * **Analyze HNNs in Label-scarce Scenarios (W2)**
>
>     We conducted additional experiments under more label-scarce settings.
>
> >The reviewer confirmed that the concerns were addressed and replied: **"My primary concern was about the inconvenience of installation, and since this has been resolved by pip-based approach, I would like to raise my score from 6 to 8."**
>
> ### Additional Comments in the Discussion Phase
>
> * **Future Extension Plan for Applications in Other Domains**
>
>     We outlined our roadmap for future extensions of DHG-Bench, which includes three planned directions for broadening its applicability to other domains.
>
> > The reviewer provided two key follow-up assessments:
> > * **Regarding the roadmap:** They acknowledged that **"the proposed benchmark would be useful for applying hypergraph to a wide range of applications, which is a significant contribution in the field."**
> > * **General endorsement:** In a separate comment block, the reviewer highlighted two notable contributions and explicitly expressed an inclination to **champion the acceptance of the paper**.
>
> ## Reviewer uqj6 (Score 8)
>
> * **Explanation of ED-HNN Performance (W1)**
>
>     We provided a detailed analysis of why ED-HNN’s performance in our benchmark does not fully match that reported in the original paper.
>
> * **More Discussion on Newly Introduced Metrics (W2)**
>
>     We added a detailed discussion on how an ideal hypergraph model is expected to behave under the newly introduced robustness and fairness metrics.
>
> * **More Statistics on the Datasets (W3)**
>
>     As suggested, we provided additional dataset statistics to highlight the extent of higher-order interactions in the hypergraph data.
>
> ## Reviewer FzKL (Score 4)
>
> * **Random Hyperedge Splits with Mixed Heuristic Negative Sampling (W1)**
>
>     We first clarified that our hyperedge prediction experiments follow the standard and widely adopted evaluation protocols used in prior studies. Nevertheless, following the reviewer’s suggestion, we provided additional temporal and inductive hyperedge splits to better account for potential temporal and inductive drift.
>
> * **All Tasks are Supervised (W2)**
>
>     We clarified that the current HNN literature primarily focuses on supervised tasks, and establishing a benchmark centered on these tasks is essential for ensuring fair comparisons. However, to further strengthen our benchmark as suggested, we introduced additional self-supervised learning tracks.
>
> * **Insufficient Explanation of Performance Variation across Datasets (W3)**
>
>     To better explain the observed performance variation, we provided a more systematic discussion from two perspectives: dataset characteristics and architectural design choices.
>
> * **In-depth Model Behavior Analysis (Q1 and Q2)**
>
>     We conducted additional experiments to evaluate model sensitivity to datasets containing very large hyperedges versus many small ones. We also performed analyses comparing how different HNNs behave on nodes with very high versus very low degrees. These results offer deeper insights into HNN model behavior under diverse structural conditions.
>
> * **A Guide for Practitioners (Q3 and Q4)**
>
>     We strengthened the paper by providing practical guidance on selecting appropriate HNN models for different types of tasks. In addition, we added a detailed discussion on the key trade-offs among model performance, scalability, and dataset characteristics.

---

> ### Author Response · Authors · 2025-12-03
> **Summary of Responses for each Reviewer (Part II)**
>
> ## Reviewer KSgV (Score 4)
>
> ### Responses to Initial Concerns
>
> * **Limited Conceptual Novelty (W1)**
>
>     We emphasized that the core contribution of our work is to introduce a fair, comprehensive, and reproducible benchmark foundation for the field, rather than proposing new methodologies or theoretical frameworks. We further detailed the specific contributions to demonstrate the work's value in bridging the evaluation gap.
>
> * **Baseline Selection Concerns (W2 and Q1)**
>
>     We clarified that our baseline selection follows standard practices. While direction-aware GNNs align less with our undirected focus, we incorporated two such baselines as suggested. The results further confirmed that HNNs outperform graph-based approaches in hypergraph learning.
>
> * **Directed Hypergraphs Ignored and Future Extension Plan (W3 and Q2)**
>
>     We justified our focus on standard undirected hypergraphs, highlighting that (1) they represent the mainstream literature, and (2) a solid undirected benchmark serves as a necessary foundation for complex variants. As suggested, we also outlined a roadmap to extend DHG-Bench to complex hypergraphs (e.g., directed hypergraphs) in future work.
>
> * **Explanation on OOM Mitigation Strategies (W4 and Q3)**
>
>     We clarified that our benchmark mitigates OOM issues through sparse matrix operations and explained why certain reviewer-suggested strategies were unsuitable. We also pointed out that peak GPU memory usage had already been reported in the original paper.
>
> * **Heterophily Analysis (W5 and Q4)**
>
>     We highlighted that heterophily was already quantified in Table A1. To investigate performance degradation, we conducted additional experiments testing the reviewer-suggested hypotheses (oversmoothing and feature collapse). Our results identified feature collapse as the primary factor, aligning with prior literature.
>
> * **Explanation of Fairness Metric Computation for Datasets without Sensitive Attributes (Q5)**
>
>     We clarified that fairness metrics are computed only on datasets that contain existing demographic-sensitive attributes, and we do not generate proxy attributes for any other dataset.
>
> ### Additional Comments in the Discussion Phase
>
> While the initial technical concerns were largely resolved, we addressed three remaining conceptual points raised shortly before the discussion paused.
>
> * **Novelty Concerns Arising from the Presence of Existing Surveys**
>
>     We addressed this concern by clarifying that, unlike prior surveys focused on conceptual categorization and taxonomy, our work provides the first standardized and reproducible empirical benchmark for HNNs. We explained its broader impact in bridging evaluation gaps and improving practical usability for practitioners.
>
> * **Missing Comparison to Directed Hypergraph Methods**
>
>     We corrected a factual misunderstanding regarding our results: the reviewer's expectation for directed baselines appeared to stem from misinterpreting the result formatting (mistaking bold text for best performance). We clarified that our experiments show that advanced HNNs outperform directed GNNs across all evaluated tasks. Furthermore, we restated why our benchmark focuses on mainstream undirected HNNs.
>
> * **Lack of Novel Dataset to Support the Advantage of HNNs**
>
>     We emphasized that our inclusion of diverse real-world hypergraphs provides sufficient empirical evidence to validate HNN advantages. We explained the structural incompatibility of the suggested dataset construction approach (which is tailored for directed hypergraphs) with our mainstream undirected scope.
>
> Finally, we would like to thank the AC again for your time and efforts in reviewing our submission.
>
> Best regards,
>
> Authors

---

### Meta-Review · Area_Chair_VGHC · 2026-01-06

**Summary:**

This paper introduces a comprehensive benchmark for hypergraph representation learning that evaluates 17 architectures across 22 datasets. The benchmark spans node-level, hyperedge-level, and hypergraph-level tasks, and evaluates models along multiple dimensions, including accuracy, efficiency, robustness, and fairness. The code and datasets have been released.

Most reviewers agree on the following strengths:
- The work addresses an important gap that has hindered progress in hypergraph learning research.
- The benchmark is comprehensive in terms of evaluated models, datasets, and metrics.
- The paper provides several meaningful insights that can inform future research.
- The code and datasets are released, supporting reproducibility.

The reviewers suggested additional experimental settings, and the authors have addressed most of these requests, making the benchmark more extensive and robust. The most critical concern raised by one reviewer is the limited conceptual novelty.

The paper’s strengths clearly outweigh its limitations, and acceptance is therefore recommended.

**Reviewer Concerns:**

The reviewers suggested additional experimental settings, and during the rebuttal phase, the authors addressed most of them, with only a few minor exceptions, such as directed hypergraph neural networks. However, the concern regarding limited conceptual novelty remains.

**Reviewer Scores:**

Considering the rebuttal, it is likely that most reviewers adjusted their scores positively, except for Reviewer KSgV, who provided additional critical comments. Moreover, Reviewer CWSt explicitly championed this work, although my decision was not solely based on that support.

---

### Decision · Program_Chairs · 2026-01-26

Accept (Poster)